# Testing the effects of topography, geometry and kinematics on modeled thermochronometer cooling ages in the eastern Bhutan Himalaya

Michelle E. Gilmore[1], Nadine McQuarrie [1], Paul R. Eizenhöfer [1], Todd A. Ehlers [2]

[1] Department of Geology and Environmental Science, University of Pittsburgh, Pittsburgh, 15260, USA
[2] Department of Geoscience, University of Tübingen, Tübingen, D-72074, Germany

*Correspondence to*: Nadine McQuarrie (nmcq@pitt.edu)

**Abstract.** In this study, reconstructions of a balanced geologic cross section in the Himalayan fold-thrust belt of eastern Bhutan are used in flexural-kinematic and thermokinematic models to understand the sensitivity of predicted cooling ages to changes in fault kinematics, geometry, topography, and radiogenic heat production. The kinematics for each scenario are
created by sequentially deforming the cross section with ~10-km deformation steps while applying flexural loading and erosional unloading at each step to develop a high-resolution evolution of deformation, erosion, and burial over time. By assigning ages to each increment of displacement, we create a suite of modeled scenarios that are input into a 2-D thermokinematic model to predict cooling ages. Comparison of model-predicted cooling ages to published thermochronometer data reveals that cooling ages are most sensitive to (1) location and size of fault ramps, (2) variable
shortening rates between 68-6.4 mm/yr, and (3) timing and magnitude of out-of-sequence faulting. The predicted ages are less sensitive to (4) radiogenic heat production, and (5) estimates of topographic evolution. We used the observed misfit of predicted to measured cooling ages to revise the cross section geometry and separate one large ramp previously proposed for the modern décollement into two smaller ramps. The revised geometry results in an improved fit to observed ages, particularly young AFT ages (2-6 Ma) located north of the Main Central Thrust. This study presents a successful approach
for using thermochronometer data to test the viability of a proposed cross section geometry and kinematics, and describes a viable approach to estimating the first-order topographic evolution of a compressional orogen.

## 1 Introduction

Cooling ages recorded by thermochronometers are a direct function of the timing, magnitude, and rate of exhumation in fold-thrust belts (e.g., Ehlers and Farley 2003; Shi and Wang 1987; Huerta and Rodgers, 2006; Rahn and Grasemann, 1999;
McQuarrie and Ehlers, 2017). However, the rate and magnitude of exhumation may be strongly controlled by the geometry and rate of deformation (Lock and Willett 2008, McQuarrie and Ehlers, 2015). Previous studies have shown that thermochronometers are most sensitive to the vertical motion of material, such as fault motion over a fault ramp, which focuses exhumation at that location (Whipp et al., 2007; Herman et al., 2010; Robert et al., 2011; Coutand et al., 2014; McQuarrie and Ehlers, 2015). Because of this, several hundred kilometers of horizontal shortening along a flat décollement,

a phenomenon commonly observed in fold-thrust belts, may occur without a significant thermal cooling signal (e.g., Batt and Brandon, 2002; Huntington et al., 2007; Whipp et al., 2007; Coutand et al., 2014). Thus, potential variations in cross section geometry such as the spatial distribution of ramps, the order of faulting, and how fault and ramp positions change with time are predicted to have a significant impact on the exhumation history of fold-thrust belts.

The shape of subsurface isotherms and the cooling history of minerals are also controlled by the evolution of topography, something that is largely unknown and often modeled either as a steady state topography that matches modern topography (e.g. Coutand et al., 2014; Erdös et al., 2014; Herman et al., 2010; Whipp et al., 2007) or as an evolving topography where relief increases or decreases with time as indicated by geologic datasets (e.g., Erdös et al., 2014). The spatial and temporal changes in cooling rate due to topographic relief depend on topographic wavelength and amplitude, exhumation rate and

duration, and the thermochronometer systems recording the change (Ehlers and Farley, 2003; Braun et al., 2002; Mancktelow and Grasemann, 1997; Stüwe et al., 1994). Attempts to predict past relief from thermochronometer ages using a non-linear inversion method (Valla et al., 2010) determined that relief development must be 2–3 times faster than the background exhumation/erosion rate to be recorded in the measured ages — a criterion that is hard to achieve in actively deforming and exhuming regions. These studies highlight yet unresolved issues regarding the best approach in deciphering

the topographic evolution of an actively deforming orogen. In this study, we evaluate the sensitivity of predicted cooling ages to the different parameters that control exhumation magnitude, rate and location in fold-thrust belts. We expand on the approach taken by McQuarrie and Ehlers (2015) and assess the control that cross section geometry, kinematics, shortening rates, and topographic assumptions have on modeled cooling ages by systematically changing these features. These parameters are evaluated using a balanced geologic cross section and associated thermochronometer data from the Bhutan

Himalaya on a section line that is adjacent (30 km east) to the one examined in McQuarrie and Ehlers (2015). This approach not only requires kinematic compatibility and the ability to match predicted cooling ages for each cross section, but also allows us to evaluate compatibility in geometry, age, and rate of deformation between the two adjacent sections.

## 2.1 Tectonostratigraphy

The Himalayan orogen initiated with collision of the Indian Plate with the Asian Plate c. 50-55 Ma (e.g., Patriat and

Achache, 1984; Klootwijk et al., 1992; Leech et al., 2005; Najman et al., 2010) and is divided into four geomorphic and tectonostratigraphic zones that span much of the east-west extent of the orogen. From south to north, these are the Subhimalaya, Lesser Himalaya, Greater Himalaya, and Tethyan Himalaya (Fig. 1). All of these units were derived from sediments originally deposited on the Indian Plate (Heim and Gansser, 1939; Gansser, 1964). In the following section, we describe the tectonostratigraphy and intervening structures expressed along a section line near Trashigang in the Bhutan

Himalaya (Figs. 1, 2) (Long et al., 2011a; 2011b).

The Subhimalaya zone is located north of the Main Frontal Thrust (MFT) and composed of synorogenic sedimentary deposits from the Himalayan foreland basin. In Bhutan the MFT emplaces a single thrust sheet of Miocene-Pliocene

Subhimalayan units referred to as the Siwalik Group over modern foreland basin deposits (Gansser, 1983; Long et al., 2011b; Coutand et al., 2016).

The Lesser Himalayan zone consists of Neoproterozoic to Permian strata, collectively grouped as the upper Lesser Himalaya, and Paleoproterozoic strata comprising the lower Lesser Himalaya (Gansser, 1983; Bhargava 1995; Long et al., 2011a). The youngest unit of the upper Lesser Himalaya, the Permian Gondwana succession, is exposed north of the Subhimalayan zone in the hanging wall of the Main Boundary Thrust (MBT) and in the immediate footwall of the thrust sheet carrying the stratigraphically older Permian Diuri Formation. North of these units, multiple fault-bound packages of the Neoproterozoic-Cambrian Baxa Group are repeated in the upper Lesser Himalayan duplex. The Shumar Thrust (ST) exposed immediately to the north is interpreted as the roof thrust of the system (McQuarrie et al., 2008; Long et al., 2011b). In the hanging wall of the Shumar Thrust, the Paleoproterozoic Daling-Shumar Group is overlain by the stratigraphically unconformable Neoproterozoic-Ordovician Jashidanda Formation. These strata are repeated multiple times to form the lower Lesser Himalayan duplex with the Main Central Thrust (MCT) as the roof thrust (McQuarrie et al., 2008; Long et al., 2011a; 2011b).

The MCT separates the Lesser Himalayan zone from the Greater Himalayan zone to the north (Heim and Gansser, 1939; Gansser, 1964). The Greater Himalaya is divided into two structural levels: the lower unit is above the MCT but below the out-of-sequence Kakhtang Thrust (KT), while the higher unit is in the hanging wall of the KT (Grujic et al., 2002). Estimates for the initiation of motion on the Main Central Thrust (MCT) range from ~25 to 20 Ma (e.g., Hodges et al., 1996; Daniel et al., 2003; Tobgay et al., 2012), with continued shearing in the Bhutan Himalaya through 18–16 Ma (Grujic et al., 2002; Daniel et al., 2003; Kellett et al., 2009). The age of motion on the KT is notably younger (14-8 Ma; Daniel et al., 2003; Grujic et al., 2002, 2011; Coutand et al., 2014). Regional-scale upright, non-cylindrical antiforms and synforms mapped throughout the Greater Himalaya are interpreted to be a result of underlying Lesser Himalayan duplex formation (Long et al., 2011b). Both Greater and Lesser Himalayan rocks preserve pervasive ductile deformation above and below the MCT (Grujic et al., 1996; Long et al., 2011c; 2016) that cannot be replicated with kinematics that only account for fault displacement. However, during initial emplacement of the MCT and active displacement on the MHT, ductile processes at depth transition to brittle processes as thrust and shear systems approach the surface, with a transition temperature of ~350°C (Avouac, 2007). These cooler processes, friction on brittle faults, and erosional exhumation control modeled fault rates (Beaumont et al., 2001; Jamieson et al., 2004; Avouac, 2007). Although our approach does not capture ductile deformation at depth, it does capture the displacement and cooling below 350–400°C. Even at temperatures below 350–400°C, almost all of the rocks in Bhutan have undergone some component of granular-scale strain (Grujic et al., 1996; Long et al., 2011c; 2016). In the models we present here, all thrust sheets are treated as rigid bodies that were translated by discrete structures.

### 2.2 Thermochronologic Data

We limit the data used to test the cross section by Long et al. (2011b) to those within 15 km of the line of section (Supplemental Table 1). Cooling ages are shown in map view (Fig. 1) and plotted versus distance from the MFT along the

Trashigang cross section (Fig. 2). In order to maintain structural context along the cross section, sample locations are projected onto the cross section along structure (i.e. in the direction of the fault trend, maintaining distance from structures). [40]Ar/[39]Ar of muscovite (MAr) and apatite fission track (AFT) data used are from previous studies and presented with 2σ analytical error (Supplementary Table 1) (Stüwe and Foster, 2001; Grujic et al., 2006; Long et al., 2012; Coutand et al.,
2014). Previously published zircon (U-Th)/He (ZHe) data are determined from the mean age of replicates (typically 3 grains) and presented in this study with a 2σ error that encompasses the range in measured ages (Long et al., 2012). Thirty km west of the cross section modeled in this manuscript is the Kuru Chu cross section (Fig. 1), which has an accompanying suite of cooling ages (Long et al., 2011b, 2012). The Kuru Chu cross section and accompanying data were forward modeled by McQuarrie and Ehlers (2015) using the approach presented here.

MAr data are published only for Greater Himalayan rocks in the immediate hanging wall of the MCT and range from 14.1±0.4 to 11.0±0.4 Ma (Stüwe and Foster, 2001). The spatial extent of this dataset is limited to a 9-km span in the structurally lower Greater Himalaya, including two cooling ages of 14.1 and 11.1 Ma from samples less than 0.5 km apart in the immediate hanging wall of the MCT. The range in ages could be a function of residence time at the highest temperatures reached (650-700°C) or residence time near the closure temperatures of the minerals, or how rapidly the minerals cooled
(e.g. Mottram et al., 2016). In a study that examined white mica ages immediately above and below the MCT in Sikkim, Mottram et al. (2015) showed that muscovite single-grain ages had a significantly larger age spread (2-5 Ma) that was not seen in MAr plateau ages. In addition to residence time and thermal conditions experienced by the rocks affecting argon loss, they suggested that a ± 2 Ma age dispersion would be expected due to diffusive differences caused by grain size variations. MAr ages from eastern Bhutan post-date the age of south-directed shear on the MCT in this region (Stüwe and Foster, 2001;
Grujic et al., 2002; Daniel et al., 2003; Kellett et al., 2009; Long et al., 2012). Thus, we interpret the age range of these four MAr samples as the window of permissible exhumation-induced cooling through the modeled closure temperatures of white mica (Ehlers et al., 2005; Braun, 2003).

The eight ZHe samples from Lesser Himalayan rocks that we use have cooling ages ranging from 11.6±0.1 to 7.3±0.8 Ma along a 40-km across-strike distance (~20-50 km north of the MFT). These ages were interpreted by Long et al. (2012) to
indicate structural uplift, exhumation, and cooling of Lesser Himalayan rocks through the zircon (U-Th)/He closure temperature at ~11.5-9.5 Ma. Measured ZHe ages in the Kuru Chu are 1-2 Myr younger than the ZHe ages along the Trashigang section between 15 and 30 km from the MFT (Long et al., 2012) (Figs. 1, 2). The predicted ZHe ages in this study do not account for the effects of radiation damage on the closure temperature (e.g. Guenthner et al., 2013), which could lead to potentially underestimating the ZHe closure temperature. However, the effects of radiation damage on ZHe (or
AHe) closure temperatures are most pronounced for long durations at relatively low (~220°C) temperatures (Guenthner et al., 2013). The Lesser Himalayan samples evaluated here experienced temperatures greater than 300-350°C (Long et al., 2011c, 2012), have young ages (typically ~7-11 Ma) and underwent extremely rapid cooling (e.g. 16.3-22.5°C /Myr since closure at ~180°C), thereby leading us to infer that radiation damage effects are minimal. North of the MCT, ZHe cooling ages are limited to two samples, one from the structurally higher Greater Himalaya and one from Tethyan rocks at the

western edge of the Sakteng Klippe. These samples recorded cooling ages of 7.4±1.6 Ma and 7.1±0.3 Ma, respectively (Coutand et al., 2014; Long et al., 2012).

AFT cooling ages from the Lesser Himalaya are limited to four samples that range between 6.3±2.3 to 4.2±1.0 Ma (Long et al., 2012; Grujic et al., 2006). The youngest age is from the Jashidanda unit in the immediate footwall of the MCT (Grujic et al., 2006). The three older ages from 6.3±2.3 to 5.7±1.0 Ma are from Diuri and Baxa units ~25-35 km farther south. In the structurally lower Greater Himalaya, AFT cooling ages progressively decrease from south to north from 7.8±2.8 to 3.7±0.6 Ma ~70-90 km north of the MFT (Grujic et al., 2006). One young AFT age of 3.1±1.2 Ma is immediately north of the MCT (Stüwe and Foster, 2001) and two similar ages of 3.0±1.4 Ma and 3.6±1.0 Ma are also found 10 km farther north (Grujic et al., 2006). The range in ages of six AFT samples from the structurally higher Greater Himalaya is 2.5±0.4 to 4.2±0.8 Ma (Coutand et al., 2014). In order to avoid skewing the overall fit of models based on fit or misfit to these six cooling ages from the higher Greater Himalaya (>25% of AFT data included in this study), we discuss these AFT data sampled north of the line of section as one collective sample point that includes the spatial and temporal variability of the entire cluster when comparing the data to model results in the following text sections. We apply the same approach for the cluster of three AFT data from the immediate hanging wall of the MCT, where ages range from 7.8±2.8 to 3.1±1.2 Ma in a span of less than 0.5 km along the line of section (Figs. 1, 2) (Stüwe and Foster, 2001; Grujic et al., 2006). However, to allow for visual comparison of individual cooling ages, all 22 AFT ages are shown in the figures when plotting predicted versus measured thermochronometer ages

Few of the cooling age data along the Trashigang section display age-elevation relationships, similar to that shown and discussed by McQuarrie and Ehlers (2015) for the Kuru Chu section. AFT ages from the structurally higher Greater Himalaya exhibit a modest age elevation relationship, which suggests exhumation rates of 0.4 mm/yr. Examining both Trashigang and Kuru Chu datasets, an age-elevation relationship may be present in the ZHe data with younger ages (8.5 to 10 Ma) at lower elevations (0.5 to 1 km) in the Kuru Chu and older ages (11 to 11.6 Ma) at higher elevations (1.6 to 2.4 km) along the Trashigang transect. If so, the data suggest differential exhumation of 0.7 mm/yr.

## 3 Methods

### 3.1 Flexural and Kinematic Model

Long et al. (2011b) published a balanced cross section in the Trashigang region of Bhutan (Fig. 2). We used the structural modeling software *Move* (Midland Valley) to sequentially deform (forward model) the Trashigang section using fault-slip amounts determined from the cross section. It is important to note that the models created in *Move* (and Pecube) do not attribute any mechanical behavior to the rocks; they only describe kinematics, or the motion of material. The cross section was deformed in ~10-km increments and included isostatic loading due to fault displacement and unloading due to erosion in each increment. The magnitude of isostatic load was determined from the difference between each increment of deformed topography and the topography of the previous step (McQuarrie and Ehlers, 2015). Erosional offloading was based on the

difference between the deformed, isostatically loaded profile and a new topographic profile generated at each deformation step (McQuarrie and Ehlers, 2015). The methods used to estimate the new topographic profiles are discussed in section 3.1.3. Including isostatic response in the model produces a record of syn-deformational exhumation and deposition, facilitates the steepening of the décollement over time, and develops a foreland basin (McQuarrie and Ehlers, 2017).

The process of linking kinematic models of deformation derived from balanced cross sections to advection-diffusion thermal models in order to calculate the evolving subsurface temperatures and predict cooling ages has been explored recently by several research groups (Almendral et al., 2014; Erdös et al., 2014; Mora et al., 2015; McQuarrie and Ehlers, 2015; Castelluccio et al., 2016; Rak et al., 2017). The level of kinematic detail modeled in each of these examples varies greatly, as well as how depths of measured samples were projected backwards in time. Each kinematic step can range from 5 to 30 km

over estimated time steps of 0.25-15 Ma. The flexural response of deformation has either been calculated explicitly in the reconstruction software (McQuarrie and Ehlers, 2015; Rak et al., 2017) or estimated based on reconstructed paleodepths, foreland basin history and/or perceived flexural response by using the flexural-slip unfolding algorithm in *Move* (Erdös et al., 2014; Mora et al., 2015; Castelluccio et al., 2016). Due to this growing method of linking cross section kinematics to thermal models, it is critical to examine how sensitive the predicted ages are to how flexural isostacy and topography are

calculated, because both control the depth and thus thermal history of rocks through time (McQuarrie and Ehlers, 2017).

### 3.1.1 Model Parameters

During the flexural-kinematic modeling process, effective elastic thickness (EET), crustal density, and initial décollement dip were systematically varied to optimize the fit of the final modeled cross section to the observed geology at the surface, foreland basin thickness (~6 km), and décollement dip (~4°) (Long et al., 2011b). We placed highest priority on matching

surface geology. Over 50 different flexural-kinematic models were created in which topography, EET, density, kinematics, or geometry were varied. Out of these models, nine presented in this study produced a foreland basin, dip of the décollement, and surface geology that were all considered acceptable: within 1 km of modern thickness; +1.0/-0.5° of modern dip; and 1 km of modern surface geology (Supplementary Fig. 1).

Young's modulus and mantle density were held constant at 70 GPa and 3.3 g/cm$^3$ respectively. Best-fitting flexural models

in this study used values of 65-70 km EET. These values correlate well with regional estimates for the Himalaya but are high compared to eastern Himalayan estimates (Jordan and Watts, 2005; Hammer et al., 2013) that are strongly dependent on the width of the modern foreland basin and the location of the Shillong Plateau. The EET values in our best fit models are based on reproducing the depth of the foreland basin preserved in the Siwaliks (5.5-6 km) and the dip of the modern décollement (similar to the ~5° dip of the Moho (Mitra et al., 2005; Singer et al., 2017)), and thus take into account the estimated strength

of the lithosphere over a much longer window of time. We emphasize that the particular values for EET and density are not unique, but represent a combination that are able to reproduce the surface geology. We evaluated EET values as low as 40-60 km. However, these were unable to match foreland basin thickness, geology exposed at the surface, or décollement dip. Keeping all other parameters the same, a change in EET at the last model step to reflect modern conditions established

between ~2 Ma to present would increase the depth of the décollement and decrease the resulting magnitude of erosion at the surface by ~250 m. Flexural-kinematic model parameters are presented in table 1 along with the kinematic and topographic variations used in each flexural model. A two-dimensional grid of points spaced 0.5 km apart was distributed across the section and sequentially deformed with the cross section to generate high-resolution displacement vectors describing how the kinematics of the system evolve in ~10-km increments. By assigning an age to each step, the displacement field is converted into a velocity field that is used in the thermal and cooling-age prediction model Pecube (Erdös et al., 2014; McQuarrie and Ehlers, 2015; Rak et al., 2017). Each model presented in this study was run using four to seven different suites of velocities to 1) see predominant trends on the predicted cooling ages and 2) determine which combination of velocities resulted in predicted cooling ages that best matched the measured data.

**3.1.2 Kinematic Variations Considered**

Out-of-sequence thrusting along the KT occurred sometime between 14 Ma and 8 Ma, significantly more recently than motion on the MCT (Davidson et al., 1997; Grujic et al., 2002; Daniel et al., 2003, Hollister and Grujic, 2006). However, uncertainty remains regarding the magnitude and age of slip along the KT. Long et al. (2011b) argued for 31-53 km of minimum KT displacement. We tested three kinematic scenarios in the flexural models by varying the relative timing of KT motion, called the Early KT, Split KT, and Late KT models (Fig. 3). Early KT is modeled with 45 km of motion along the KT immediately following motion on the Shumar Thrust (Fig. 3c.1). In Split KT, out-of-sequence thrusting is modeled in two separate stages with 25 km of motion applied after deformation along the Shumar Thrust, followed by 20 km of motion after upper Lesser Himalayan duplexing (Fig. 3c.2). Late KT is modeled with 45 km of out-of-sequence thrusting after development of the upper Lesser Himalayan duplex, similar to the proposed model of sequential deformation by Long et al. (2012) (Fig. 3c.3).

An enigmatic low-relief surface is preserved in the Bhutan Himalaya. In eastern Bhutan, this surface is located in the immediate footwall of the KT (Duncan et al., 2003; Grujic et al., 2006; Adams et al., 2013). The low-relief landscape contains hundreds of meters of sediment infilling of paleo-relief and is now out of equilibrium with respect to where it formed (Adams et al., 2016). In eastern Bhutan, the infilled sediment is derived from the structurally higher Greater Himalaya; conglomerate is common, thus making it easy to associate the clasts with rocks carried by the KT. Previous studies have highlighted the ubiquitous response of footwall subsidence and the development of low relief in the footwall region of out-of-sequence faults and thrusts (e.g. McQuarrie and Ehlers, 2015, 2017; Rak et al., 2017). Thus the low relief surface in eastern Bhutan may be a potential relict of KT motion. Given the uncertainty of magnitude and timing of KT motion, we tested multiple kinematic scenarios of out-of-sequence thrusting in this study. We hypothesized that changing the relative timing of out-of-sequence KT motion in relation to the evolution of the décollement would alter the topographic evolution and isostatic history of the modeled cross section, and associated thermochronometer ages predicted along it.

### 3.1.3 Topographic History Estimation

To model the isostatic response to deformation and erosion, we tested three different methods of estimating the topographic evolution during forward modelling of the cross section in *Move*. Each method was variable in topographic detail and in its ability to account for common factors of fold-thrust belt development such as deformation front migration, localized topographic uplift, and structural subsidence. The three topographic models were evaluated in the thermokinematic model to determine the sensitivity of the predicted thermochronometric data to each topographic scenario. The "no topography" scenario is the simplest of the three estimations with a topographic profile that remained at sea level throughout the entire section reconstruction. We also tested a "static topography" scenario with a topographic profile broadly similar to the modern topographic gradient of Bhutan (Duncan et al., 2003) that maintains a steep gradient in the first ~25 km behind the active deformation front, followed by shallower gradient with elevation increasing along a two-degree slope to a maximum of ~5 km (Fig. 4). This shape of the topographic profile remains identical or static throughout the kinematic evolution. The static topographic profile is spatially translated as the location of the deformation front is adjusted progressively southward throughout the sequential development of the fold-thrust belt. A critical caveat to the static topography method is that topographic elevations are not perturbed by isostatic loading. Thus, the grid points in the model subside due to deformation-induced loading, but the topography does not. The third topographic model is a "responsive topography" that estimates a topographic profile for each flexurally loaded ~10-km deformational step using a Python-based computer script (McQuarrie and Ehlers, 2015). New topography is defined by a northward increasing slope (similar to modern topography) in regions of active structural and topographic uplift, while in areas without active uplift, the program follows existing, isostatically loaded topography. This approach allows topography to respond to deformational loading and erosional unloading. For models using the static and responsive topographies, the initial topography assigned to the restored section simulates a pre-existing fold-thrust belt in the Tethyan sequence before the initiation of the MCT (Ratschbacher et al., 1994; Murphy and Yin, 2003; Webb et al., 2011). This topography maintains 0-km elevation from the southern end of the restored cross section to the lower Lesser Himalaya. Across the Lesser Himalaya, topographic elevation increases across a distance of 140 km and reaches a maximum elevation of 5 km above the lower Greater Himalaya, which at its southernmost extent is buried at a depth of 16 km below sea level.

### 3.2 Thermal and Cooling Age Prediction Model

The velocity field and topography for each increment of deformation after displacement, isostasic response, and erosion have been applied are used as input into a University of Tübingen modified version of Pecube (Braun, 2003; Whipp et al., 2009; McQuarrie and Ehlers, 2015). The thermokinematic model Pecube functions as: (1) a kinematic model that uses fault geometries and high-resolution point tracking inputs from *Move* to calculate rock transport velocities; (2) a transient thermal model that calculates the thermal field using fault motion, erosion above the topographic surface, rock thermophysical properties, and thermal boundary conditions; and (3) a set of age prediction algorithms (Ehlers et al., 2005) that calculate a

suite of thermochronometer ages for material at the topographic surface for each deformation step using the thermal histories of particles as they are exhumed and cooled from depth to the model surface (e.g. Coutand et al., 2014; McQuarrie and Ehlers, 2015). Modeled results highlight that increased rates of thrusting and exhumation advect isotherms upward while basin subsidence in the foreland locally depresses isotherms. Motion on the MCT and KT produce the same inverted thermal

gradients that have been both observed and reproduced in previous modeling studies (Henry et al., 1997; Bollinger et al., 2006; Hollister and Grujic, 2006; Herman et al., 2010).

### 3.2.1 Radiogenic Heat Production

The thermal state of the crust depends on the basal heat flow from the mantle and the material properties of the crust (e.g., thermal conductivity, density, heat capacity, and radiogenic heat production). Following the approach and rationale

summarized in McQuarrie and Ehlers (2017), we prescribe an exponential decrease in heat production with depth, as opposed to assuming a constant crustal heat production. An exponential decrease in heat production with depth requires definition of a surface heat production ($A_0$) and an "e-folding depth". One caveat of this approach is that material properties are not exhumed during the simulations to modify the surface heat production value. However, an exponential decrease in heat production with depth has the advantage of honoring observations that heat production diminishes with depth through

the crust and that this decline is not monotonic (Chapman, 1986; Ketcham, 1996; Brady et al., 2006). This approach not only allows matching measured surface values of heat production in the Himalaya (e.g., Whipp et al., 2007), but also produces reasonable mid and lower crustal temperatures that would not produce partial melts. We varied $A_0$ to test the sensitivity of predicted cooling ages to variations in rock thermophysical properties. Calculated values of radiogenic heat production in the Himalaya are highly variable. A low radiogenic heat production estimate of 0.8 $\mu W/m^3$ for the entire Indian Shield was

calculated based on observed low-in-heat-flow by Ray and Rao (2000), but other measurements have been estimated as high as 1.5-5.5 $\mu W/m^3$ due to the abundance of potassium, uranium, and thorium in granitic and gneissic rocks (Menon et al., 2003). Similar ranges of radiogenic heat production values from 1.5 to 6.0 $\mu W/m^3$, with clustering around 4 $\mu W/m^3$, have also been found for Greater Himalayan rocks (e.g., England et al., 1992; Whipp et al., 2007). Herman et al. (2010) concluded a best-fitting constant radiogenic heat production value of 2.2 $\mu W/m^3$ in their own thermokinematic model using a constant

basal temperature of 750°C at 80 km depth. In this study we tested models of the Long et al. (2011b) cross section geometry using $A_0$ values ranging from 4.0 to 1.0 $\mu W/m^3$ in 0.25-0.5 $\mu W/m^3$ increments and a heat production that decreases exponentially with an e-folding depth of 20 km. Thermal conductivity and heat capacity were held constant at values of 2.5 W/m K and 800 J/kg K, respectively, based on observed thermophysical properties for the lithologies present in the Himalaya (Whipp et al. 2007, and Ehlers, 2005). Although thermophysical properties such as thermal conductivity, heat

capacity, and density vary between different lithologies, the implementation of variable material properties in areas of large deformation is not possible in Pecube, which solves the advection-diffusion equation on an Eulerian grid. Thus, we address this potential issue by using the best available average measurements of thermophysical properties for the lithologies in this region. All thermal rock property parameters used in our Pecube simulations are listed in table 2.

### 3.2.2 Variable Deformation Age and Rate

To compare the effects of differing time and rate of fault motion on predicted cooling ages, several deformation ages and velocities were tested. The combinations of velocities, radiogenic heat production values, and flexural models tested are in table 3.

A constant velocity of 17.3 mm/yr using a MCT initiation age of 23 Ma was tested to determine if a generalized long-term rate of shortening can adequately reproduce published cooling ages. This rate is comparable to the ~15-25 mm/yr estimates of modern convergence (Bilham et al., 1997; Larson et al., 1999; Banerjee and Burgmann, 2002; Zhang et al., 2004; Bettinelli et al., 2006; Banerjee et al., 2008) and long-term rates of shortening for the Himalaya (DeCelles et al., 2001; Lavé and Avouac, 2000; Long et al., 2011b).

In Bhutan, variable rates of shortening have been proposed based on the integration of shortening estimates from balanced cross sections with thermochronometer data. These rates range from 4 to 60 mm/yr (Long et al., 2012). Modeling rates of shortening along the Kuru Chu section using cross section kinematics in a thermal model, McQuarrie and Ehlers (2015) found the best match to measured cooling ages with rates that varied from 7 mm/yr to as high as 75 mm/yr. We evaluate a suite of velocities starting with these two published variable deformation rate scenarios. Velocity model A is based on rates

proposed by Long et al. (2012) along the Trashigang section with pulses of rapid deformation during MCT motion (32 mm/yr) and the formation of the upper Lesser Himalayan duplex (37-41 mm/yr), separated by slower periods of deformation during lower Lesser Himalayan duplexing (15 mm/yr) and motion along the MBT and MFT (4-6 mm/yr). Velocity model B is broadly based on rates proposed by McQuarrie and Ehlers (2015), namely fast velocities (55-75 mm/yr) during formation of the upper Lesser Himalayan duplex. In velocity model B, MCT motion initiates at 20 Ma with a slower velocity (21

mm/yr) and duplexing of the lower Lesser Himalaya at similar rates (22-25 mm/yr), while the upper Lesser Himalayan duplex deforms at a rate of 69-75 mm/yr. Other rates of motion in this scenario are comparable to velocity model A. In addition to these rates, we varied rates of shortening for the formation of the lower Lesser Himalayan duplex (16-25 mm/yr), formation of the upper Lesser Himalayan duplex (45-75 mm/yr), and emplacement of the MBT and MFT (4-10 mm/yr). Inherent in testing these suites of velocities is testing the sensitivity of cooling ages to the start and end date of these different

structural systems.

### 4 Results

### 4.1 Flexural-kinematic Model

Summaries of the final output of all seven flexural-kinematic models to the published Trashigang cross section (Long et al., 2011b) are presented in table 1; supplemental figure 1 contains images of the results of each model. Because the flexural-

kinematic models control locations and magnitudes of erosion and burial that are input into the thermal model, we evaluate the effects of estimated topographic evolution, different proposed kinematics, and amount of subsidence (illustrated by the final shape of the décollement), on exhumation magnitudes and the geology exposed at the present-day surface.

The difference between model results are subtle but show local variations in total erosion of 0.5-4 km that is reflected in the final geology exposed at the surface of the model and depth to stratigraphic markers within the model. All models produced foreland basin depths within 2 km of the 5.5-6 km thick Siwalik section exposed in eastern Bhutan (Long et al. 2011a; 2011b). Average décollement dips varied from 3.75° to 5.4°. Six out of seven models are within the 4-7° décollement angle estimated for the Main Himalayan Thrust (Ni and Barazangi, 1984; Mitra et al., 2005; Schulte-Pelkum et al., 2005; Singer et al., 2017).

Each of the kinematic scenarios produced different flexural responses (Table 1). Models using Late KT deformation produced the deepest foreland basins and steepest décollement dips, along with under-eroded geology at the surface compared to the published section. These results are a function of the different kinematic scenarios imposing variations in the distribution of uplifted topography and consequently different flexural loading profiles over the evolution of the cross section. Early KT and Split KT scenarios have décollement dips shallower than Late KT models and result in a better match to the surface geology data, except when using No Topography. Differences among Early KT and Split KT décollement dips and surface geology are not systematic, indicating these differences are less driven by kinematics and appear to be more sensitive to slight variations in flexural isostasy parameters and the profile of the topographic load. The poorest fit to surface geology was produced by the model combining Split KT with No Topography (Supplemental Fig. 1). In all other model combinations, exposed geology is within ~1 km of the modern geology observed at the surface, with particularly good fits combining Early KT deformation with the Static Topography, and Split KT with Responsive Topography.

Topographic profiles from the final deformation step of each model vary in fit and misfit to observed topography along the Trashigang section (Fig. 4). The sea-level No Topography profile is the worst fit of the three estimations. Static Topography fits the steep topographic rise from the MFT to the southern trace of the Shumar thrust; however, to the north of the Shumar Thrust, estimated elevations are ~1 km greater than observed. Responsive Topography provides a better fit for the northern half of the section, including a local drop in elevation from 77-90 km along the section north of the MFT. However, the average 2° slope assigned to the topographic profile resulted in under-predicted elevations from 13-55 km, where the average observed topographic slope of the range is steeper (4.5°). Overall, Responsive Topography best reproduces the observed topography along the cross section.

For models using responsive and static topographies, we attribute the differences in décollement dip to the lower topographic relief produced using a 2° angle with Responsive Topography, compared to the steeper topographic angle near the deformation front and overall higher elevations with Static Topography (Fig. 4). The shallower topography from the Responsive Topography requires a steeper décollement to accommodate the same amount of material between the surface and the décollement (i.e. broadly maintaining the same taper angle). The most significant result of flexural modeling was identifying the relationship between uplift or subsidence of rock (as represented by the two-dimensional grid of points) and the uplift, subsidence or static position of topography. The static profile used when modeling with Static Topography or No Topography can result in regions of non-erosion and burial (with respect to the topographic surface). When the deformation front shifts toward the foreland, higher topography is translated southward with no direct relationship for where structural

uplift is occurring. Additionally, material will subside in areas responding to flexural loading while topography does not. This latter example is especially relevant south of the Kakhtang Thrust during out-of-sequence thrusting. While using Responsive Topography, both points and topography subside in front of the Kakhtang Thrust, which allows for minor amounts of erosion to occur across the entire section during fault motion. Using the Static Topography, points subside due to

the imposed load but topography does not which simulates burial in this region. Thus, the Static Topography disconnects the topographic evolution from the kinematic and flexural evolution by not accounting for structural uplift and subsidence. The thermal consequences of the different flexural-kinematic models are explored in Section 4.2.

### 4.2 Predicted Cooling Ages Across the Cross section

### 4.2.1 Effect of Radiogenic Heat Production and Constant Shortening Velocity on Predicted Ages

By holding velocity constant and testing multiple values of radiogenic heat production in Pecube, we can discern the effect that adjusting radiogenic heat production may have on the output of predicted cooling ages, as well as the viability of a constant rate of shortening with time. We compare predicted cooling ages for AFT, ZHe, and MAr systems to published ages using a range of surface radiogenic heat production ($A_0$) values from 1.0 to 3.0 $\mu W/m^3$ (Fig. 5). The kinematic input is from the flexural-kinematic model combining Split KT and Responsive Topography, coupled with a constant velocity of 17.3

mm/yr from 23 Ma to the present.
     The most apparent trend among all three thermochronometer systems is that predicted cooling ages become younger as $A_0$ increases from 1.0 to 3.0 $\mu W/m^3$ due to the higher temperatures throughout the model. In addition, for the MAr system, the modeled heat production values determine if the subsurface temperatures are hot enough in the crust to yield reset ages at the modeled present-day surface. When $A_0$ is low (1.0 $\mu W/m^3$), the only reset MAr ages are north of the Kakhtang Thrust, while

output with higher $A_0$ (3.0 $\mu W/m^3$) include MAr ages as young as 5.8 Ma in the upper Lesser Himalaya (Fig. 5a). Changes in $A_0$ have the smallest effect on predicted AFT cooling ages for a range of $A_0$ values tested. Predicted AFT ages are controlled by the motion of rocks over the active ramp located ~65-75 km from the MFT. The relatively rapid shortening rate produces a very shallow-dipping predicted age curve from ~30-65 km from the MFT, with the youngest ages focused at the active ramp. Changes in radiogenic surface heat production slightly change the predicted age by 1-3 Myr, with larger predicted age

differences (~5 Ma) for Greater Himalayan rocks that have not been transported over the ramp. These ages are significantly older (5-15 Myr) until 90 km from the MFT, the location of the Kakhtang Thrust (Fig. 5). For the ZHe system, changing $A_0$ values notably changes the pattern of predicted cooling ages. The hottest surface radiogenic heat production value ($A_0 = 3.0$ $\mu W/m^3$) produced a ZHe signal identical to the AFT, but slightly older. However, the coolest value ($A_0 = 1.0$ $\mu W/m^3$) generated a markedly different trend from ~25-65 km along section, where the youngest predicted ages are at the southern

limit of the upper Lesser Himalayan duplex and become gradually older to the north. This north-to-south younging of the predicted cooling ages is the expected signal for a hinterland-dipping duplex (Lock and Willett, 2008; McQuarrie and Ehlers 2017). The predicted age range, 5-10 Ma, closely matches the ages of upper Lesser Himalayan duplex formation (3.5-12.8

Ma) in the constant velocity model (Table 2 or 3). In AFT and ZHe systems, the trend of older predicted cooling ages 65-85 km north of the MFT forms an upside-down U-shape in the Greater Himalaya section between the MCT and KT regardless of surface radiogenic heat production value (Fig. 5).

Evaluating the fit between measured and modeled ages predicted by the different thermal models using this constant velocity, we observe that all three models reproduce less than half of all measured cooling ages. The best match to published AFT ages out of these three models is $A_0 = 1.0 \ \mu W/m^3$; however, this is still a rather poor fit. Even with a cool crust, we find that predicted ages are too young to fit most published AFT and ZHe ages but significantly too old to match published MAr data. While MAr prediction improves slightly with high surface radiogenic heat production ($A_0 = 3.0 \ \mu W/m^3$), even younger modeled AFT and ZHe ages poorly fit most data. These simultaneous over- and under-estimations of published ages require models with more complex rates of deformation and exhumation to match the measured ages.

### 4.2.2 Effect of Shortening Rate Variations on Predicted Ages

A constant rate of deformation described in the previous section does not produce cooling ages that match all three thermochronometer systems (Fig. 5). In this section we present modeled cooling ages from two variable velocity schemes that are compared to published cooling ages: velocity model A (Long et al., 2012) and velocity model B (McQuarrie and Ehlers, 2015) (Table 3). All variable velocity models presented in this section used a surface radiogenic heat production value of 2.5 $\mu W/m^3$ with the flexural-kinematic model combining Split KT and Responsive Topography as input. These parameters produced the best fit of modeled to measured ages for the original cross section geometry.

Using velocity model A in Pecube results in a visibly improved fit compared to the constant deformation rate. Cooling ages predicted in the model are within the range of error or variability of 16 out of 28 published cooling ages (57%). Predicted AFT ages fit seven out of 15 published samples (47%), with five of the samples not matched by the predicted ages located between 70 to 120 km from the MFT (Fig. 6). Only three of the youngest measured AFT ages (3.0-3.6 Ma) matched modeled ages in the Greater Himalayan rocks 55-65 km from the MFT, while predicted AFT ages were 3-4 Myr older than the cluster of AFT ages in the structurally higher Greater Himalaya. Predicted ZHe ages match eight out of 13 samples (62% fit), while predicted MAr ages are 2 Myr older than the oldest measured age of 14.1 Ma. Modeled MAr ages pass through the cooling window during lower Lesser Himalayan duplex formation and motion on the Shumar Thrust, which ceases activity at 15 Ma for velocity model A.

During the formation of the upper Lesser Himalayan duplex, a faster rate of deformation (37.3 mm/yr) than in the constant velocity model produces older ages across the upper Lesser Himalaya (~10 Ma) and a better match to published ZHe data (Fig. 6a). Due to the large amount of shortening accommodated by the upper Lesser Himalayan duplex, fast shortening is also required to predict the pattern of ZHe ages that do not young towards the south 15-35 km from the MFT (Fig. 5b). Across the structurally lower GH, the faster rates match one of two measured ZHe ages. Topographic uplift and increased exhumation as rocks are structurally uplifted over the décollement ramp 65 km from the MFT results in young predicted ZHe ages that match the measured ZHe ages at the ramp, but predicts ages that are notably older than the measured age north

of the ramp. Southward displacement of rocks over the ramp produces the south-to-north younging of ZHe ages 35-58 km from the MFT (Fig. 6).

Samples from the upper Lesser Himalaya cool through the AFT closure temperature after out-of-sequence motion on the KT and during rapid deceleration in deformation rate from 37 mm/yr to 6 mm/yr at 10 Ma. Both the out-of-sequence thrusting and the slower deformation rate create a prolonged timeframe for rocks to cool (Fig. 6a). Similar to ZHe ages, predicted lower Lesser Himalayan AFT ages are controlled by motion of rocks over the active MHT ramp located 65 km from the MFT. The slope of the predicted AFT ages from 30-65 km is a function of the rate of shortening from 7 Ma to present. Predicted AFT ages in the Greater Himalaya systematically increase north of the ramp, similar to the pattern observed with the constant velocity output (Fig. 6). These older predicted AFT ages located 65-85 km from the MFT cool much earlier in the deformation history when rocks were structurally uplifted over a ramp in the lower Lesser Himalaya during early stages of upper Lesser Himalayan duplexing (Fig. 3c.2).

Velocity model B uses an earlier MCT initiation at 20 Ma and a rate of upper Lesser Himalayan duplexing that is twice the rate used in velocity model A (Table 3). Despite this difference, fits to published data are remarkably similar to velocity model A, with a marginally improved fit to MAr data and upper Lesser Himalayan ZHe data (Fig. 6b). Predicted MAr ages produce a better match to published data due to a younger age for the growth of the lower Lesser Himalayan duplexing: 13.5-17 Ma with velocity B versus 15-20 Ma with velocity A. Faster and earlier upper Lesser Himalayan duplexing, which ends at 11 Ma in this scenario versus 10 Ma in velocity A, predicted slightly older and better-fitting modeled ZHe data across the upper Lesser Himalaya (10-11 Ma). Eleven out of 13 ZHe ages (85%) are reproduced within error. As in velocity model A, 7 out of 15 AFT ages are reproduced (47%). Predicted AFT ages still remain too old in the Greater Himalayan zone from 65 km northward. Although the timing and rates of deformation used in velocity model B result in a significantly better fit to published thermochronometer data than constant velocity and slightly better fit than velocity A, there is still a large discrepancy between predicted and measured AFT ages across the Greater Himalaya that cannot be resolved by changes in velocity. The sensitivity of predicted cooling ages to the age and rate of shortening is expanded on in section 5.

### 4.2.3 Effect of Topographic Estimates on Cooling Ages

We evaluate the sensitivity of the predicted thermochronometer ages to different topographic development approaches (Responsive, Static, and No Topography) using the Split KT kinematic scenario and velocity model B. The resulting predicted ages for different thermochronometer systems are shown in figures 6b and 7. The significant overlap of modeled cooling ages for the three methods of estimating topographic evolution indicates the predicted cooling ages are much less sensitive to how topography is approximated than to changes in deformation velocity, surface radiogenic heat production, or geometry (Section 4.3).

The No Topography model generated predicted ages that are remarkably similar to the Responsive Topography. In detail, No Topography yields identical or slightly older (0.5 to 3 Myr) predicted ages than the Responsive Topography, with the greatest difference in the predicted lower Lesser Himalayan AFT ages. This is in contrast to our initial expectations that the

over-eroded No Topography model would produce younger cooling ages than the other topographies because the No Topography scenarios always produced higher total exhumation; the final cross section was over-eroded by 1-2.3 km (Table 1, Supplementary Fig. 1). However, this total exhumation accumulates over the modeled history, suggesting that the incremental over-erosion of the No Topography scenario is always significantly less than the exhumation driven by structural uplift. In other words, exhumation differences due to different estimates of topography (<1 km) are significantly less than that required (2-3 km) to be recorded in thermochronometer ages (Valla et al., 2010).

Results from Static Topography versus Responsive Topography models show greater differences in predicted cooling age trends. In ZHe and MAr plots, the largest difference between the models is the spatial width of the reset cooling ages. For example, reset MAr cooling ages in the Responsive Topography model start 33 km north of the MFT, versus 36-37 km north in the Static Topography model. For ZHe ages, reset ages from the Responsive Topography model start at 10 km north of the MFT while reset ages from the Static Topography start at 20 km. There is also a high degree of scatter in the predicted AFT ages from the Static Topography model. Between 35 and 90 km, these ages range from 3-13 Ma without any pattern, except for directly over the ramp at 55-65 km from the MFT. This highly irregular cooling history is a function of the topographic modeling method not accounting for structural uplift or structural subsidence with time. Static topography that is simply spatially translated as the MHT advances southward inaccurately models burial of material where points are subsiding and modeled topography is not subsiding, and produces over-erosion of material where points experience structural uplift but modeled topography remains static. These results highlight that estimates of topographic evolution must account for areas of structural uplift and isostatic subsidence when modeling fold-thrust belt evolution.

### 4.2.4 Effect of Kinematic Variation on Cooling Ages

Changes to the prescribed kinematic order used in forward modeling the cross section were tested using flexural-kinematic models with Responsive Topography, coupled with velocity model B and surface radiogenic heat production of 2.5 μW/m$^3$ in the thermokinematic model. Because different thrust structures have different slip magnitudes, it is not possible to have precisely the same velocities with different kinematics. To most closely evaluate the effect of kinematic variations in out-of-sequence thrusting, we kept the age at which velocities change the same whenever possible. Predicted cooling age output for Early KT and Late KT kinematic scenarios are plotted in figure 8 and compared with results from the same Split KT scenario used in Section 4.2.3.

Fits of modeled MAr ages to published data are unaffected by changes to the timing of out-of-sequence thrusting; all three scenarios resulted in predicted MAr ages of ~14 Ma in the hanging wall of the MCT. This is expected because all changes to out-of-sequence thrusting occur after the formation of the lower Lesser Himalayan duplex from 13-17 Ma. In all of the modeled scenarios, the age and rate of shortening in the lower Lesser Himalayan duplex set the predicted ages for the MAr system between 40 and 60 km from the MFT.

Each of the three kinematic scenarios predicted significantly different ZHe ages across the upper Lesser Himalayan duplex, implying that there is a particular kinematic order of deformation required in the flexural-kinematic model to generate the

measured cooling ages. The pattern of predicted ZHe ages between 10 and 65 km from the MFT is controlled by age and rate of displacement of the upper Lesser Himalayan duplex, the final step of which places duplexed Baxa units over younger Gondwana rocks on a ramp in the MHT (Fig. 3d). This last step structurally elevates the entire duplex and increases local exhumation. Continued motion of the duplex over this ramp cools the rocks through the AFT system. In the Split KT

kinematic model, displacement over this ramp occurs at 11 Ma, just before the second stage of motion on the KT. In the Early KT and Late KT models, upper Lesser Himalayan duplexing is immediately followed by motion of the duplex over this ramp between 7 and 10 Ma, after a marked decrease in shortening velocity at 10-11 Ma (Table 3). The altered timing of this displacement results in young (7-10 Ma) ZHe ages and AFT ages (Fig. 8). Compared to Split KT model results (Fig. 6b), the younger ZHe ages predicted in Early KT and Late KT models are a poorer fit to published data at 10-40 km from the

MFT. The 4-5 Myr gap between published ZHe and AFT data in the upper Lesser Himalaya is only reproduced using the Split KT kinematic model. The second stage of out-of-sequence thrusting in this model postdates the development of the upper Lesser Himalayan duplex but predates motion of the duplex over the ramp of younger rocks, causing a 4-5 Myr delay between these two processes that focus exhumation in the Lesser Himalaya. In addition, none of the ZHe ages from the Greater Himalaya could be reproduced by Early KT and Late KT models using velocity model B, while the Split KT model

results fit two out of three data. To reproduce the 7.42 Ma ZHe age from the structurally higher Greater Himalaya, the absolute age of out-of-sequence thrusting would need to be at least 2 Ma younger in both models to create a mechanism of exhumation and cooling through ZHe closure. This would consequentially alter shortening rates and cooling ages both before and after out-of-sequence thrusting, producing younger modeled ZHe ages in the Lesser Himalaya, which are already too young in the Early KT and Late KT models.

The fit of predicted ages to published AFT data across the upper Lesser Himalaya 10-30 km north of the MFT is similar in all three models (Figs. 6, 8). The matching AFT curves are due to the same ages and rates of fault motion along the MBT and MFT from ~7.3 Ma to the present, when upper Lesser Himalayan rocks cool through the AFT closure isotherm in the models. Though the magnitude and timing of out-of-sequence thrusting impacts the predicted AFT ages in Greater Himalayan rocks directly south of the KT, none of these three kinematic scenarios reproduced the observed south-to-north

younging in AFT ages 70-90 km from the MFT (Figs. 6 and 8). In this area, predicted AFT ages were set during upper Lesser Himalayan duplexing, when Greater Himalayan material is carried over a ramp in the MHT. Cooling ages were subsequently modified by motion on the KT, which structurally uplifted Greater Himalayan rocks along a steep fault. The magnitude of subsidence produced in the Late KT model prevents any significant erosion from occurring in the model after out-of-sequence thrusting. Because the Split KT model applies smaller magnitudes of out-of-sequence thrusting twice,

predicted ages from the model are between the ages from Early KT and Late KT. The out-of-sequence thrusting prior to upper Lesser Himalayan duplexing in the Early KT allows for structural uplift of Greater Himalayan material after KT motion, which induces topographic uplift and erosion and predicts AFT ages 3-4 Myr younger than the Late KT scenario (Fig. 8). However, predicted cooling ages from Early KT are still 5 Myr older than the measured ages. The strong gradient from recently reset AFT ages predicted 50-70 km north of the MFT to older AFT ages 70 km to the north is a function of the

prescribed geometry of the MHT. To change this erosional history and pattern of cooling ages, which was observed in all previously presented models regardless of prescribed $A_0$, deformation rate, or kinematic order, requires a mechanism of structural and/or topographic uplift in this region.

### 4.3 Effect of Cross Section Geometric Variations on Ages

Multiple studies have shown that thrust geometry has a first-order control on cooling ages in convergent orogens (e.g., Lock and Willett, 2008; McQuarrie and Ehlers, 2015, 2017; Rak et al., 2017). Our thermokinematic modeling of the Long et al. (2011b) cross section shows that motion over a footwall cutoff in the Daling formation (Fig. 3C.2a) facilitated AFT cooling in the model from 11-13 Ma, too early to produce the measured ages of 3-6 Ma 65-90 km from the MFT. However, younger AFT ages are modeled at 50-65 km from the MFT due to more recent motion over a footwall cutoff in the Baxa and Diuri units (Fig. 6). We hypothesized that modifying the geometry of the cross section, specifically changing the locations of décollement ramps, would result in an improved model fit of the young observed AFT ages across the Greater Himalaya.

In a new, modified and re-balanced version of the Trashigang cross section, the décollement has been adjusted to partition the large ramp cutting through the Diuri and Baxa units into two separate ramps (Fig. 9). The footwall cutoff of the Diuri has remained in its same position along the décollement (65 km north of the MFT), but the footwall cutoff of the Baxa unit has been shifted ~35 km north to the present-day northern end of the lower Lesser Himalayan duplex (105 km north of the MFT). In balancing this modified geometry, an additional 35 km of slip was added to the amount of overall shortening along the section. This shortening occurs after the formation of the duplexed Baxa Group and before motion on the MBT.

Several flexural-kinematic models with this new décollement geometry were evaluated in Pecube to find the best fit to the geology exposed at the surface, dip of the décollement, and depth of the foreland basin. All models use the Split KT kinematic scenario, modified to accommodate updated magnitudes of displacement. The models varied slightly in topographic evolution (using Responsive Topography) and assigned EET. Multiple combinations of velocity and radiogenic heat production values were coupled with the updated kinematics in Pecube to evaluate the sensitivity of the predicted cooling ages to changing these parameters (Table 3).

Unlike previous models shown in this paper that used one surface radiogenic heat production value across the length of the cross section, the best fit was achieved using a higher surface radiogenic heat production of 4.0 $\mu W/m^3$ in the region of exposed Greater Himalaya rocks and a lower 2.0 $\mu W/m^3$ value for Lesser Himalaya rocks. Using different values for radiogenic heat production is consistent with previous studies that have noted the higher radiogenic heat production capacity of Greater Himalayan rocks which cluster around 4.0 $\mu W/m^3$, while Lesser Himalayan rocks have a lower average radiogenic heat production value of 2.5 $\mu W/m^3$ (Roy and Rao, 2000; Menon et al., 2003; England et al., 1992; Whipp et al., 2007; Herman et al., 2010). Because Pecube currently does not accommodate multiple $A_0$ values within a single model run, the model results shown in figures 9 to 11 merge the predicted ages using both $A_0$ 2.0 $\mu W/m^3$ and 4.0 $\mu W/m^3$ of separate model runs at the surface trace of the MCT. The full extent of each of these predicted cooling age trends across the length of the cross section is shown in supplementary figure 2.

**4.3.1 Fit of Predicted Cooling Ages Using Modified Geometry**

The modified geometry model using Responsive Topography resulted in a noticeably different and better-fitting predicted age trend in the region north of the MCT (Fig. 9b). Because changing values of $A_0$ can shift predicted ages to older or younger values as well as change the across-strike pattern of predicted ages (Fig. 5), both the original and the modified geometry are included in figure 9 using combined $A_0$ values of 2.0 $\mu W/m^3$ and 4.0 $\mu W/m^3$ for the predicted ages.

Using the new geometry combined with higher radiogenic heat production from the trace of the MCT northward, MAr and ZHe ages match measured ages from the Trashigang section. In addition, the predicted ages matched measured MAr and ZHe ages from the Kuru Chu section to the west, except for three samples in the immediate footwall of the KT (77-87 km from the MFT). Perhaps most critically, the modified geometry provides an improved fit and matches 10 out of 12 AFT data from 53 to 120 km north of the MFT (83%). The notable difference with the new geometry is that the U- shape of cooling ages in the immediate footwall of the KT is narrower (15 km across) and younger (3 Ma). The across-strike trend in predicted AFT and ZHe ages for the new geometry is a subdued pattern that becomes younger towards the south (55-80 km north of the MFT). The ages set by the formation of the lower Lesser Himalayan duplex have been modified by motion and accompanying exhumation over the two smaller ramps. Note the youngest predicted AFT ages at 65 km and 105 km from the MFT, co-located with the top of each ramp.

The new geometry produced AFT patterns in the Greater Himalaya that are noticeably different than the patterns produced by the original geometry. For the original geometry, regardless of $A_0$ value, the predicted AFT age trend is set by the motion of rocks over the large footwall ramp located 65 km from the MFT. This geometric solution continues to only match the youngest AFT ages and an additional sample at ~73 km from the MFT (Fig. 9a). On the north side of the ramp, predicted ages reflect the last exhumation event. The ZHe pattern is completely set by the formation of the lower Lesser Himalayan duplex, with ages becoming younger in the direction of ramp propagation. Using a higher $A_0$ value (4.0 $\mu W/m^3$) for the original geometry improves the fit for all three ZHe data north of the MCT. Particularly in this region north of the MCT, the comparison of the two cross section geometries and their predicted ages using the same $A_0$ values highlights the effect of geometry on the predicted cooling ages (Fig. 9).

South of the MCT, predicted thermochronometer ages from the modified geometry do not have as strong of a match to the measured ages as the previous best fit model (Responsive Topography, Split KT, velocity B, 2.5 $\mu W/m^3$; Fig. 6b). The revised geometry matches half of the measured ZHe ages in the Lesser Himalaya using the lower surface radiogenic heat production of 2.0 $\mu W/m^3$ and fits all three published AFT ages within error (Fig. 9b). When including measured ages from the Kuru Chu region, the overall fit to Lesser Himalayan data improves to 70%. The most noticeable change to the fit of the original geometry using a lower $A_0$ value is the markedly older predicted AFT ages. With lower surface radiogenic heat production, the AFT signal is not as sensitive to motion and associated exhumation over the MBT ramp located at ~ 25 km from the MFT (Fig. 9a).

**5 Discussion**

**5.1 Evaluating the sensitivity of predicted cooling ages**

Geothermal gradients and the resulting shape of isotherms in the model, which dictate the spatial and temporal changes in predicted cooling ages, are dynamic and change at each incremental time-step in our models based on 1) thermal parameters prescribed to each model in Pecube; 2) locations and magnitudes of fault displacement; 3) locations and magnitudes of erosion as controlled by structural uplift, isostatic flexure, topographic evolution, and the resulting erosion in the flexural-kinematic model; and 4) rates of deformation and exhumation which are dictated by the absolute timing assigned to each incremental deformation step. Because of this linked response of deformation, exhumation, and cooling, each component in the kinematics of a fold-thrust belt system imparts a characteristic cooling pattern to the cooling ages at the surface. Emplacement of a large thrust sheet imparts a pattern of reset cooling ages that is the oldest at the thrust tip and decreases towards the active ramp (Lock and Willett, 2008; McQuarrie and Ehlers, 2015; 2017). A southward-growing duplex will produce a pattern of cooling ages that young toward the south (Lock and Willett, 2008; McQuarrie and Ehlers, 2017). While rocks record cooling associated with every stage of structural evolution, the events that are recorded by any given thermochronometer system are dependent upon the magnitude of exhumation associated with each component of deformation and the thermal history of the rocks: duration and magnitude of burial, exhumation rate, and surface radiogenic heat production. If the magnitude of exhumation is particularly close to that necessary to reset a thermochronometer system, the predicted pattern of cooling ages can be significantly altered by small changes in modeled topography or surface radiogenic heat production. For example, minor changes to the prescribed topography or thermal parameters can shift the signal of preserved AFT ages in the immediate footwall of the KT to record the southward propagation of a duplex versus the southward displacement of material over a décollement ramp, particularly when the magnitude of exhumation associated with the décollement ramp is small (Supplementary Fig. 2). Below we discuss the effects of different topographic models, topographic evolutions, and thermal parameters on cooling ages predicted in Pecube.

**5.1.1 Sensitivity of predicted cooling ages to prescribed topographic evolution and EET**

Although our evaluation of different topographic models indicates a minor sensitivity in predicted cooling ages to how topography is estimated, modeling an evolving topography such as a topographic slope that either increases or decreases with time can significantly change the predicted pattern of cooling ages by controlling the magnitude of erosion that occurs at a given time and the exhumation event during which a thermochronometer passes through its closure temperature. In the case of AFT ages across the Greater Himalaya that become younger to the north (Fig. 9b), the trend is imparted by recent motion over a décollement ramp that must be north of the youngest age. However, this ramp through the Baxa unit spans a vertical distance of 2.5 km, half the height of other décollement ramps farther to the hinterland such as the ramp through the lower Lesser Himalaya where earlier uplift and erosion occurred (Fig. 3C2). The smaller magnitude of vertical uplift and

exhumation associated with the ramp only through the Baxa group (Fig. 9b) makes cooling ages associated with it more sensitive to changes in other parameters.

The most basic requirement to reproduce observed cooling ages is to match the timing of exhumation with the structures that are producing the across-strike exhumation pattern (McQuarrie and Ehlers, 2015, 2017). For a given flexural-kinematic model, this match is a function of the geometry, which controls the location of structural uplift, and EET and topography, both of which control the location of rocks with respect to the Earth's surface. In our best-fitting flexural-kinematic and thermal model combination (Figs. 9b and 10a), ~2.0-3.5 km of exhumation from 6 Ma to the present was required to match the AFT ages that decrease in age from 6 to3.5 Ma from 65 to 90 km north of the MFT. To simulate this magnitude of exhumation following isostatic loading and the decrease in topographic elevations south of the KT during out-of-sequence thrusting, the prescribed topographic taper angle was reduced from 2.0° to 1.5° during MBT and MFT motion, with a maximum elevation of 3 km modeled in the final cross section. The magnitude and timing of this exhumation was critical to generate cooling ages across the Greater Himalaya that recorded the signal of recent motion over the décollement ramp and fit the published data (Section 5.4).

Other flexural-kinematic models evaluated in this study did not predict cooling ages across lower Greater Himalayan rocks that matched published thermochronometer ages despite using the same geometry, kinematics, and thermal properties. The difference in predicted and observed ages were a function of both slightly different topographic evolution scenarios that control magnitudes and timing of erosion, and slightly different elastic thickness parameters that control the amount and timing of subsidence. Erosion angle and EET are features prescribed in the flexural-kinematic model before thermokinematic modeling. The flexural-kinematic model shown in figure 10b is remarkably similar to our best-fitting model when comparing foreland basin thickness, dip of décollement, and surface geology (Supplementary Figs. 1f, 1g). This model was obtained by using an initial topographic taper angle of 2° and an EET of 75 km. In comparison, the best-fitting model used an initial taper angle of 2° and an EET that increased from 60 km early in the deformation history to 85 km for the second pulse of motion on the KT and displacement on the MBT and MFT. Higher EET values early in the modeled deformation steps facilitated more erosion (0.5 to 1.5 km) between 8 and 17 Ma, resetting AFT ages at this time (Fig. 10b). In addition, the model displayed in figure 10b used a steeper topographic angle in the immediate foreland (3°) and a 2° angle in the hinterland later in the modeled deformation steps to more closely match modern topography and surface geology (Supplementary Fig. 1g). This steeper, higher topography resulted in less erosion from 8 Ma to present (~0.3-1.5 km) than the best-fitting model (Fig. 10a, Supplementary Fig. 1f); however the surface geology of both models is almost identical. The change in the exhumation history between the two models, although minor, produced a significantly different pattern of cooling ages 55-85 km from the MFT. The model in figure 10b, with 0.5 to 1.5 km of additional exhumation earlier in the model, produced cooling ages that record the age of older duplexing with southward younging of cooling ages.

Although matching the geology exposed at the surface is a critical test to evaluate the accuracy of the flexural-kinematic model, we were able to match the measured AFT data with a predicted AFT age pattern using a flexural-kinematic model that is under-eroded in the hinterland 55-85 km from the MFT (Supplementary Fig. 1h). Structurally lower Greater

Himalayan material was under-eroded by ~2-3 km in the final step of the model with 1-2 km of Tethyan material preserved at the surface. Tethyan strata in the Sakteng Klippe are found at the surface 10 km east of the Trashigang section line but have been erosionally removed along the section (Figs. 1 and 2). Similar to the best-fitting model, topography maintained a 2° taper until ~8 Ma; however, EET was 65 km. The lower EET allowed for more subsidence in the hinterland and thus less total erosion. From 6 Ma to present, EET was increased to 70 km and the topographic angle was reduced to 1.75°. The stronger EET facilitated less subsidence, particularly in the Greater Himalaya region. From 6 Ma to present, the hanging wall of the MCT underwent 2.0-2.5 km of exhumation, similar to our best-fitting model. This model produced similar predicted AFT ages and fit the pattern of observed data (Fig. 10c). In summary, modeled AFT ages can be very sensitive to the evolution of topography and small changes (0.5-1.5 km) in exhumation magnitude, as expected. They also can be sensitive to slight changes in EET. Although changing topographic taper angle from 2° to 1.5° may account for up to 0.5 km of exhumation, small, 5-km changes in EET which control the amount of subsidence have a larger effect (~1 km) on the age and magnitude of exhumation, which consequentially alter predicted thermochronometer ages (Fig. 10). Thus flexural-kinematic modeling that explicitly accounts for thrust loading and the resulting evolution of décollement and foreland basin is a critical component of linking cross section kinematics to thermal models.

**5.1.2 Synthesis of the Effect of Thermophysical Properties on Cooling Ages**

Altering the thermal history of the model by imparting a hotter or colder thermal field can also result in different cooling signals preserved at the topographic surface if the exhumation amount is close to a particular closure temperature for a thermochronometer system. For instance, the best-fitting model of the modified geometry run exclusively with $A_0 = 2.0$ $\mu W/m^3$ predicted AFT ages of ~7-11 Ma from 65-85 km north of the MFT, with a trend of younger ages toward the north 85-105 km from the MFT (Supplementary Fig. 2b). The pattern of AFT cooling ages, particularly between 75 and 90 km north of the MFT, is recording a signal of older structural uplift instead of recent motion over the décollement ramp.

Rocks at the surface in our best-fitting model (Fig. 9b) are at a critical thermal threshold where, when exhumed through a low thermal gradient, the rocks will preserve a different cooling age pattern than if that same exhumation occurred through a higher thermal gradient (Supplementary Fig. 2). In the colder model, material that is at the present-day surface passed through the AFT closure isotherm prior to the motion over the Baxa footwall ramp. In the hotter model, material at the present-day surface passed through the AFT closure isotherm during or after this structural uplift. If erosion were reduced by even a small amount in the hot model (0.5 to 1 km), the predicted ages from the model would produce a different trend in cooling ages more similar to a colder thermal model. Conversely, if erosion increased in the cold model, the signal at the surface may look similar to the warmer model. The difference in AFT cooling ages between $A_0$ values of 4.0 and 2.0 $\mu W/m^3$ highlight that the magnitude of exhumation in the Greater Himalaya in this model is around the minimum amount necessary to record these younger AFT ages at the surface.

**5.2 Using Thermochronology to Evaluate Structural Geometry**

Using traditional geologic and geophysical constraints to create balanced cross sections can often result in multiple interpretations of the subsurface geology with significant variations in proposed subsurface structures, décollement ramp locations, and total shortening estimates. While kinematic reconstructions of balanced cross sections can help in determining the viability and kinematic sequence of a cross section, thermochronometer data can offer additional insights into predicting subsurface geometry. The geometry of the subsurface and location of ramps in the décollement impart a first-order control on the thermochronologic trends present at the surface (Herman et al., 2010; Robert et al., 2011; McQuarrie et al., 2014; Coutand et al., 2014; McQuarrie and Ehlers, 2015, 2017). In this study, Pecube models for two décollement geometries of the Trashigang cross section were compared, and an additional ramp in the MHT resulted in a noticeable change in cooling ages modeled (Fig. 9). This finding is particularly evident in modeled AFT ages across the Greater Himalaya. Modeled Greater Himalayan AFT ages using the original cross section geometry reflected a cooling signal imparted by a larger ramp through the lower Lesser Himalaya that did not fit the trend of published data (Fig. 6b). Even with a higher $A_0$ value assigned to the model, the location and magnitude of this cooling signal did not significantly change (Fig. 9a, Supplementary Fig. 3a). Modeling the modified geometry with an additional décollement ramp facilitated additional erosion across the Greater Himalaya and resulted in a different pattern of predicted ages that better matched the trend of published data (Fig. 9b). Another possible structural solution to produce young AFT cooling ages preserved in Greater Himalayan rocks is an out-of-sequence fault at the modern trace of the MCT (e.g., Adlakha et al., 2013). This potential fault would need to postdate motion on the KT and have enough throw (~3-5 km) to reset AFT ages. The strongest argument against this solution is the anticipated change in topography. As highlighted by our models of out-of-sequence motion on the KT, the topographic response would be a marked increase of topography in the hanging wall and much subdued topography in the footwall. This topographic response has been used to argue for out-of-sequence faulting in Nepal (Wobus et al., 2003) and is decidedly different than the topography of Bhutan (Adams et al., 2013, 2016).

**5.3 Estimates of Timing and Rates of Deformation**

We evaluated the new geometry using a suite of velocities to test the sensitivity of predicted cooling ages to prescribed shortening rates (Fig. 11a, Table 3b). As shown in figure 5, shortening rates of 17 mm/yr or higher in the last 10 Myr result in AFT ages that are 2-3 Myr younger than measured ages. Similar to previous studies (Long et al., 2012; Coutand et al., 2014; McQuarrie and Ehlers, 2015), our modeling of the Trashigang thermochronometer data requires slow shortening velocities in eastern Bhutan (6.7-7.5 mm/yr) from 6 Ma to present to match the AFT ages observed 10-30 km from the MFT, with somewhat higher velocities (8.0-14.6) permissible between 5.3 and 8.6 Ma. The earliest permissible age for these slower velocities is 11.0 Ma, at a rate of 14.6 mm/yr from 11 Ma until 5.3 Ma (Table 3; Fig. 11).

The 40 km south-to-north extent of ~8-11 Ma ZHe ages in the Lesser Himalaya requires rapid shortening rates over this window of time with fast rates permissible as early as 13 Ma. Acceptable rates of shortening range from 45 to 70 mm/yr,

which are at or exceeding plate tectonic rates. The rate of 45 mm/yr must continue until 9 Ma or younger to fit observed data. If assigned slower rates extend back to 11 Ma, the speed of shortening must increase to 65-70 mm/yr from 11 Ma to 13 Ma. The upper age limit for these modeled fast rates is controlled by predicted cooling ages that are sensitive to the time period over which the lower Lesser Himalayan duplex forms (Fig. 11). MAr and ZHe ages located 50-70 km from the MFT in the hanging wall of the MCT dictate the permissible age for the end of the lower Lesser Himalayan duplex formation and the start of the upper Lesser Himalayan duplex. Velocity models where lower Lesser Himalayan duplexing ends and upper Lesser Himalayan duplexing begins at ~15 Ma predict the oldest MAr, ZHe, and AFT ages north of the MCT and thus the poorest fit to the measured data in this area (Fig 11). South of the MCT, the ~8-11 Ma ZHe ages located 10-25 km from the MFT set the lower age limit for rapid shortening. Continuing deformation of the upper Lesser Himalayan duplex until 6 Ma results in ZHe ages that are slightly younger than the measured ages between 10 and 25 km from the MFT (Fig. 11). The limited window of time (8-13 Ma) and high magnitude of shortening (146 km) require fast shortening rates while the upper Lesser Himalayan duplex forms. Rates depend on both time of fault activity and displacement along the fault and thus a critical question is: could shortening be reduced in the upper Lesser Himalayan duplex? The cross sections were constructed to minimize shortening while matching surface constraints (Long et al., 2011b). We have re-examined the sections and any modification to the original cross section, including changing the décollement geometry as we suggest here, will increase shortening estimates and potentially increase deformation rates.

The results from both the new cross section geometry presented in this paper and the geometry originally proposed by Long et al. (2011b) are insensitive to the age and rate of MCT displacement (Figs. 6, 11; Table 3). Due to limited MAr data available along the Trashigang section and their close proximity to the exposed trace of the MCT, the measured and predicted ages in the immediate hanging wall of the MCT are all significantly younger than the age of MCT displacement. Thus any change to the ages at which the MCT starts (20 or 23 Ma) did not affect the predicted cooling ages in the region of the MAr data. However, the modeled initiation and rate of displacement of the MCT control the predicted MAr ages between 60 and 90 km from the MFT (Figs. 6, 11) in the location of the Sakteng Klippe and Greater Himalayan synform (Figs. 1, 2). These modeled ages provide a potential direction for future research that could confirm predicted ages, shortening rates, and exhumation amounts. In most of our models, MCT motion occurred from 20 Ma until 18 Ma, at a rate of 29 mm/yr.

The measured cooling ages along the Trashigang and Kuru Chu sections are largely consistent, with the most significant deviation at 15-30 km from the MFT (Figs. 9-11). Here, the younger ZHe ages (8.5-10.0 Ma) are from the Kuru Chu and older ages (11.0-11.6 Ma) were collected along the Trashigang transect. As mentioned in section 2.2, the Kuru Chu samples are from elevations 1.0-1.4 km lower than the Trashigang samples. Our modeled elevation for this region is 0.5-0.7 km, more similar to elevations of Kuru Chu ssamples, and our preferred model more strongly matches the younger Kuru Chu ages, possibly suggesting an age-elevation dependence in this region. The location that our modeled ages deviate from the measured ages along the Kuru Chu section is between 75 and 90 km in the immediate footwall of the Kakhtang Thrust. Two measured ZHe ages and one MAr age are notably younger (3-5 Myr) than our predicted ages for these systems. Two of these samples (ZHe and MAr) are from lower Lesser Himalayan rocks in the Kuru Chu Valley and the other (ZHe) is from the

immediate hanging wall of the MCT (Long et al., 2012). All three samples would require a minimum of 4-7 km of additional exhumation to reach the exposure of the samples in the Kuru Chu region. However, similar arguments could be made for samples 65-75 km from the MFT, where a similar magnitude of exhumation difference is projected between the Trashigang and Kuru Chu sections but measured ages are markedly similar (Figs. 1, 9) (Long et al., 2012). To match the young ZHe and

MAr ages sampled along the Kuru Chu sections, possible changes to the modeled Trashigang section include smaller KT displacement to reduce footwall subsidence and/or younger lower Lesser Himalayan duplex formation (and resulting increased rates of shortening during upper Lesser Himalayan duplex formation).

We found pronounced variation in shortening rates and magnitudes of rates that are similar to those presented by McQuarrie and Ehlers (2015). However, the timeframe of rapid shortening (8-13 Ma) in this study is longer and the permissible rates

slower than the timing (8.5-11 Ma) and rates (55-75 mm/yr) proposed for the Kuru Chu section immediately to the west (McQuarrie and Ehlers, 2015). The difference in windows of rapid shortening is a result of the difference in ZHe ages and MAr ages between the two regions. In the Kuru Chu region, ZHe and MAr ages continue to become younger towards the north 70-100 km from the MFT. McQuarrie and Ehlers (2015) used the slope and age of the MAr samples to argue for the age and rate of deformation of the lower Lesser Himalayan duplex along the Kuru Chu section. They found that extending

the formation of the duplex until 11 Ma provided the best match to measured thermochronometer data. However, their best-fitting model still did not reproduce the youngest cooling ages found 80-100 km from the MFT. The very old predicted ages in this region were a result of a large footwall ramp similar to the original Trashigang geometry (Fig. 6). A potential solution to both the proposed fast rates and the misfit of predicted ages 80-100 km from the MFT may be a change in ramp geometry, similar to the modified geometry proposed here for the Trashigang section. An additional driver of exhumation across the

footwall of the KT would promote younger ages there without the need for a young age of shortening in the lower Lesser Himalayan duplex. If the lower Lesser Himalayan duplex in the Trashigang region continued to ~12 Ma with a timing and rate of deformation more similar to those proposed by McQuarrie and Ehlers (2015), including a younger age in which the MHT slows (8-9 Ma), then the predicted ages would match observed ages as well as any velocity modeled here (Fig. 11) but would suggest both sections deformed at rates of 55-75 mm/yr from 8-12 Ma. These rates are faster than plate tectonic rates

and would only be permissible with coeval extension on the Southern Tibetan Detachment (STD) as proposed by McQuarrie and Ehlers (2015). A 12.5 Ma Th-Pb monazite age from Kula Kangri (at the border of Bhutan and Tibet) and 7 Ma ZHe ages (Edwards and Harrison, 1997; Coutand et al., 2014) suggest STD activity over this time window. Even though the details of the rates may continue to evolve for both sections, general trends will remain similar such as: slow velocities between ~18-13 Ma, fast (45-65 mm/yr) velocities between ~13-8 Ma, and slow velocities from ~8 Ma to present with perhaps a more

significant decrease in the last 6 Myr. This post-6 Ma decrease in convergence is consistent with the significant decrease in erosion rate at 6 Ma in eastern Bhutan proposed by Coutand et al. (2014).

## 6 Conclusions

This study presents a successful approach for using thermochronometer data to test the viability of a proposed cross section geometry based on forward models of the kinematic, exhumational, and thermal history of an area. The cross section geometry provides a model of the horizontal and vertical component of displacement. We found that the location and magnitude of vertical displacement has the most significant control on the predicted trends of cooling ages recorded by a suite of thermochronometers. Mismatches between modeled and published thermochronometer ages provide insight into how cross sections can be modified and re-evaluated in order to create a more accurate solution to known geologic and thermochronologic constraints. We found that the addition of a ramp under the Greater Himalaya in our flexural-kinematic model resulted in more accurately modeled cooling ages across this region while preserving the modeled accuracy of other geologic and geophysical parameters.

Timing and rates of deformation in compressional settings can be quantified by coupling a high-resolution flexural-kinematic model of a balanced cross section with the thermokinematic model Pecube. Adjusting the timing of motion along structures changes the timing of corresponding exhumation and thus predicted mineral cooling ages. These changes to timing and rates of deformation control the absolute ages recorded by a thermochronometer as well as the slope of cooling ages with distance in the direction of transport. We applied a variable rate of deformation to obtain a best-fitting model of the Trashigang cross section in Bhutan. Acceptable velocities for periods of rapid shortening range from 45 to 70 mm/yr between 13 and 8 Ma. These alternate with periods of slow shortening. In particular, a significant slowing of shortening rates (6.7-7.5 mm/yr) is needed at ~8-6 Ma to present.

While geometry sets the pattern of permissible cooling ages and velocity controls the absolute ages recorded, changes to surface radiogenic heat production and topographic evolution can regulate which patterns of cooling are recorded in each chronometer. Increasing surface radiogenic heat production in our models generally produced younger cooling ages, with the pattern of predicted cooling ages critically altered in areas where rocks were close to the closure isotherm for a given system. As the timing of closure shifted in a hotter model, patterns of not just younger ages but younger structures were recorded in predicted cooling ages, such as the trend of motion over a footwall ramp versus duplex formation. Our best-fitting model combined results from hot and cold thermal models for material north and south of the MCT respectively.

Although model results were less sensitive to the exact method of estimating topography, a responsive topographic method is critical for maintaining the relationship between structural uplift and subsidence and the resulting change in topography. In addition, an evolving topographic taper angle and/or evolving EET can alter the timing of exhumation and the predicted pattern of cooling ages. We found that the timing and magnitude of erosion controls which component of deformation and associated exhumation is recorded by a given thermochronometer system. Similar to changes in surface radiogenic heat production, structural signals such as duplex formation and ramp propagation may be preserved in the cooling ages of different thermochronometer systems depending on the magnitude of exhumation. A pronounced change in the modeled pattern of cooling ages is most noticeable with lowest-temperature thermochronometers. Thus, small topographic changes

can produce significantly different results in cooling age patterns for the same cross section geometry, particularly when particles are at a temperature close to the closure temperature of a given mineral cooling system. While changes in topographic gradients over multi-million year time scales are often uncertain, we can use thermokinematic modeling coupled with flexural-kinematic models that estimate topographic evolution to better understand what is driving large- and small-scale changes in the pattern of exhumation over time and space.

This work highlights the importance of considering the distribution of cooling ages in the direction of transport, to understand their relationship to the structural and topographic evolution of a landscape. Due to the predominantly lateral transport of material in fold-thrust belts, the across-strike pattern of cooling ages from thermochronometers spanning a wide range of temperatures and spatial coverage provide the most robust constraints to the structural geometry and rate of deformation. Forward modeling cross sections and cooling ages using high-resolution spatial and temporal scales reveals which structures are responsible for a given cooling pattern, their geometry, and the rate at which they move.

## Acknowledgements

The authors acknowledge research support from the Alexander von Humboldt Foundation (McQuarrie and Eizenhöfer), the University of Pittsburgh (McQuarrie, Eizenhöfer and Gilmore), and NSF EAR 0738522 to McQuarrie. Ehlers acknowledges support by European Research Council (ERC). Midland Valley provided software support and use of the program Move. We thank Willi Kappler (University of Tübingen) for his time and programming assistance, and Peter van der Beek, Djorjie Grujic, and one anonymous reviewer for their edits and suggestions. The data for this paper are available through contacting N. McQuarrie, while the modified version of Pecube that can be coupled to 2-D *Move* restoration files is available through T.A. Ehlers.

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

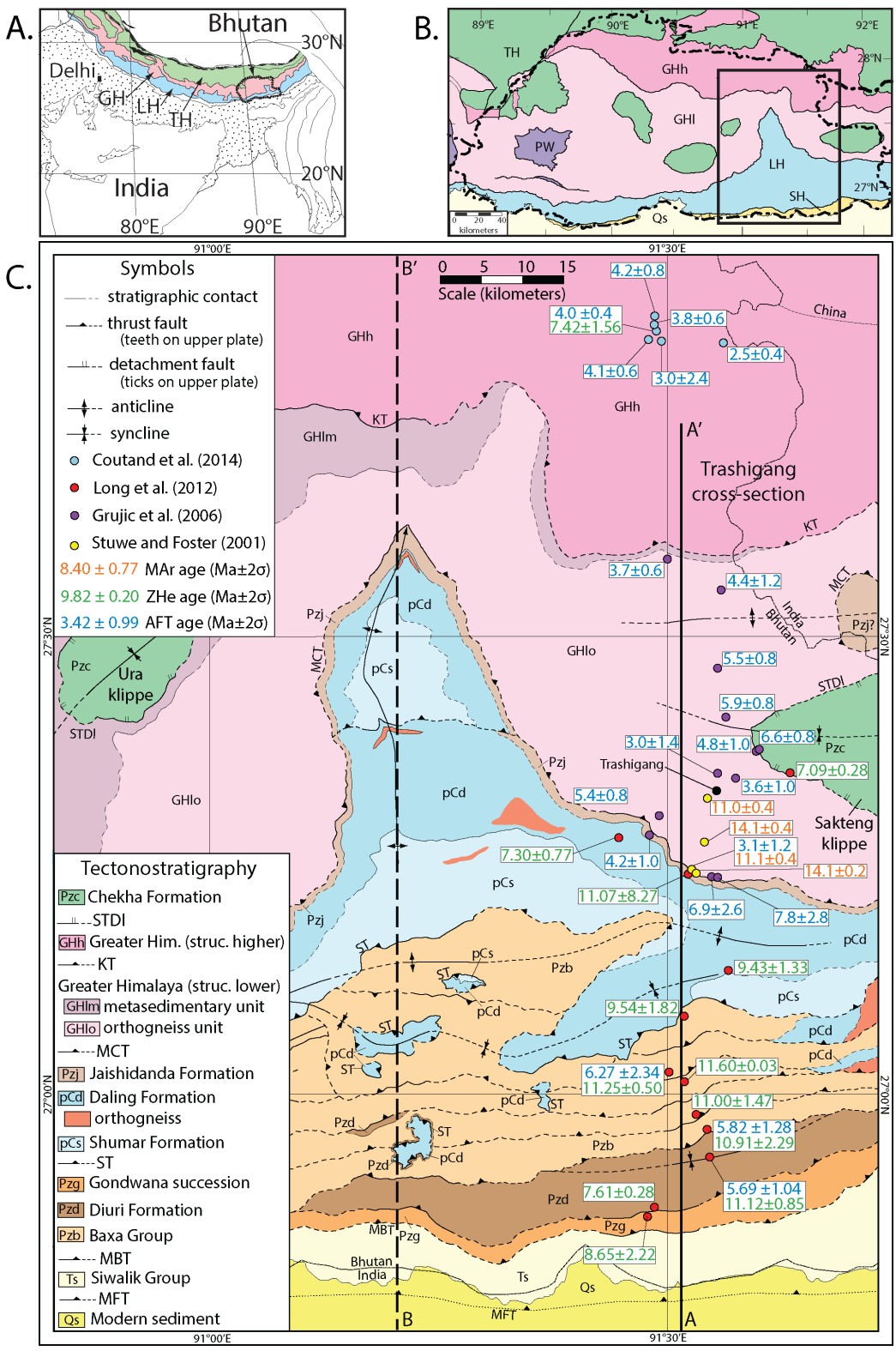

**Figure 1: (A)** Generalized geologic map of the central and eastern Himalayan orogen, modified from Gansser (1983). Abbreviations are GH: Greater Himalaya, LH: Lesser Himalaya, TH: Tethyan Himalaya.

**(B)** Simplified tectonostratigraphic map of Bhutan, modified from Long et al. (2012). The border of Bhutan is marked as a dashed and bolded line, and the area of figure 1C is outlined as a solid black rectangle. Tectonostratigraphic groups shown are TH: Tethyan Himalaya; GHh: Greater Himalaya, structurally higher; GHl: Greater Himalaya, structurally lower; PW: Paro Window; LH: Lesser Himalaya; SH: Subhimalaya; Qs: modern sediment.

**(C)** Geologic map of eastern Bhutan with Trashigang section line A-A' and reported thermochronometer data shown, modified from Long et al. (2012). Cooling ages are reported in Myr. Fault abbreviations are STDI: South Tibetan Detachment; KT: Kakhtang Thrust; MCT: Main Central Thrust; ST: Shumar Thrust; MBT: Main Boundary Thrust; MFT: Main Frontal Thrust. B-B' indicates location of the Kuru Chu cross section.

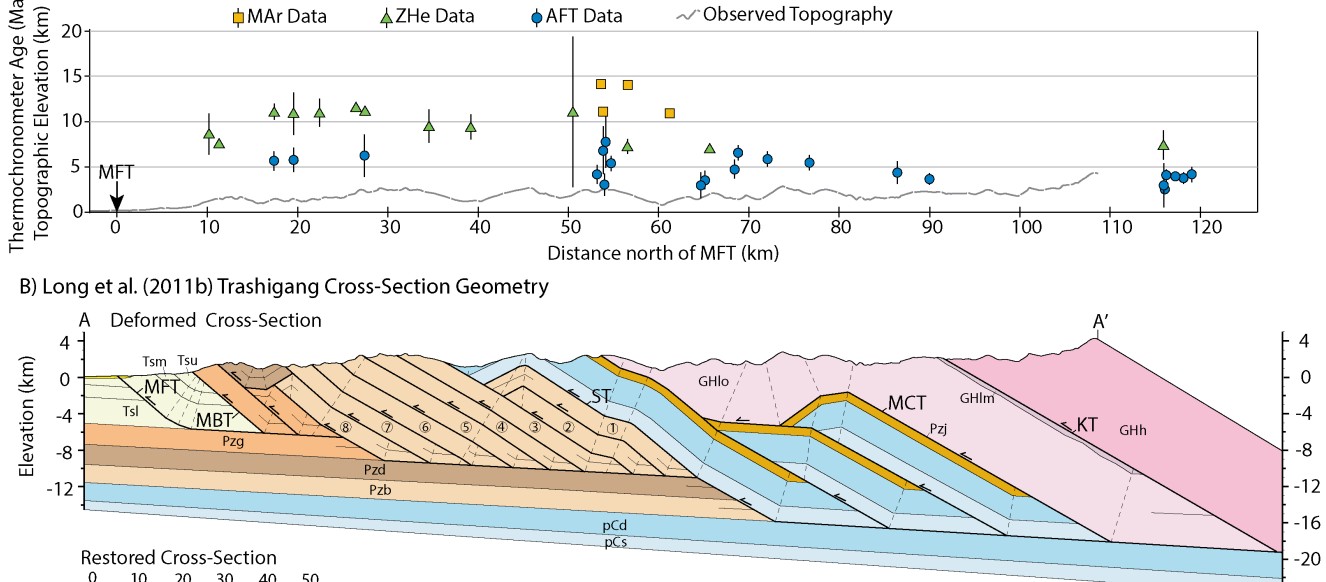

**Figure 2: (A)** Published MAr (yellow), ZHe (green), and AFT (blue) thermochronometer data plotted in the direction of transport along the Trashigang section. Topographic profile along the line of section is also shown. **(B)** Simplified balanced geologic cross section of the Trashigang region of Bhutan, modified from Long et al. (2011b). Scale of the deformed section is represented on the above graph. All unit and fault abbreviations are the same as in figure 1.

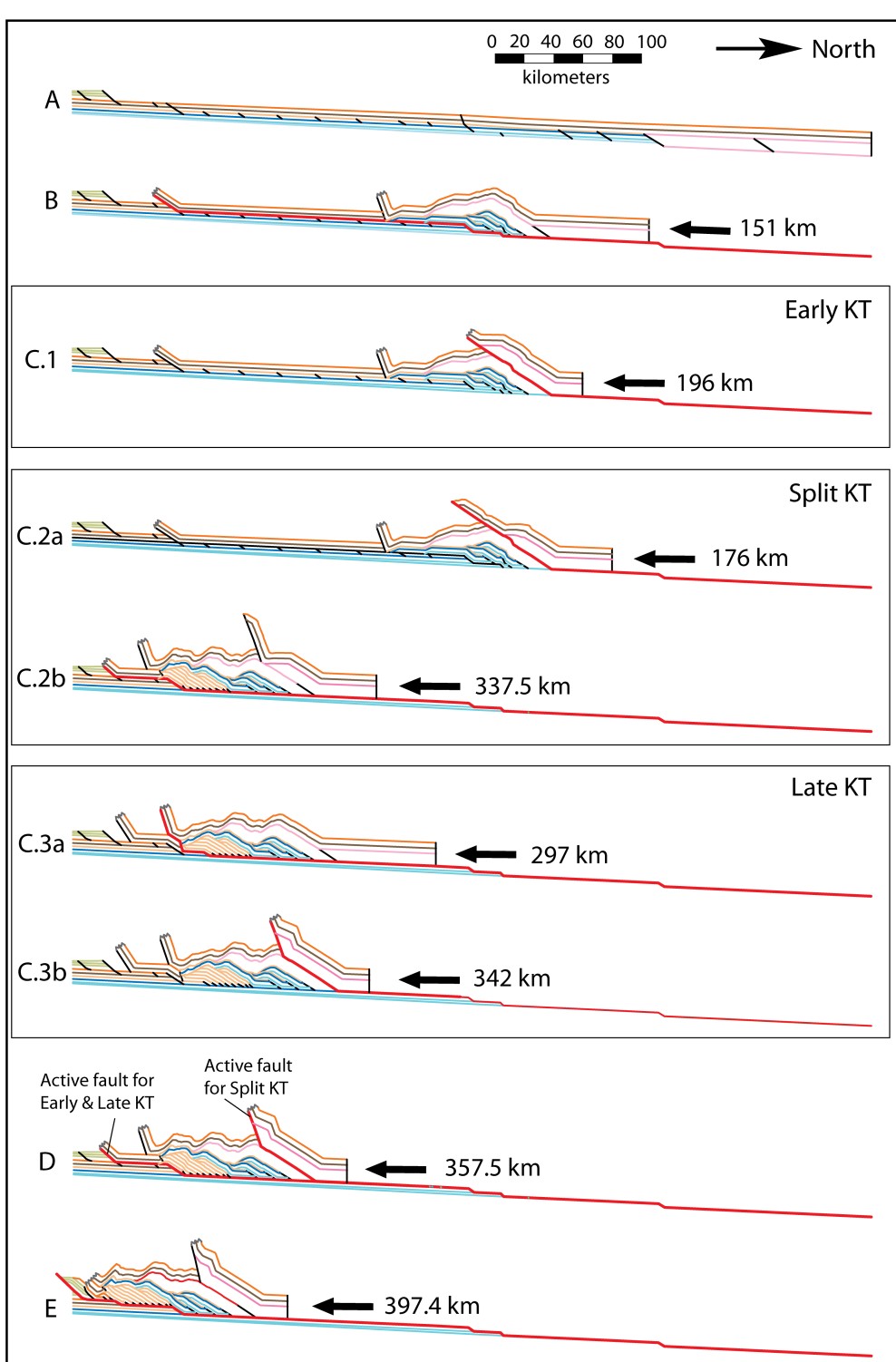

**Figure 3: Sequential kinematic reconstruction of the Trashigang cross section depicting three kinematic scenarios of out-of-sequence thrusting tested in this study. Net slip amounts are shown for each subfigure (A) the restored section used in the kinematic model; (B) deformation along the MCT and ST, including duplexing of the lower LH; (C) 1: KT motion prior to upper LH duplexing (Early KT), 2: KT motion before and after upper LH duplexing (Split KT), 3: KT motion after seven out of eight horses of upper LH duplex have been deformed (Late KT); (D) completion of out-of-sequence thrusting and Upper LH duplexing. Note that the most recent active fault in this step for Split KT varies; (E) deformation along MBT and MFT.**

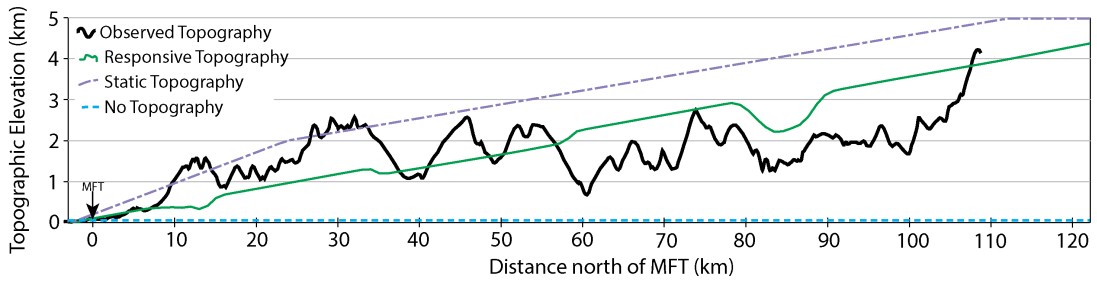

## A) Long et al. (2011b) Trashigang Cross-Section

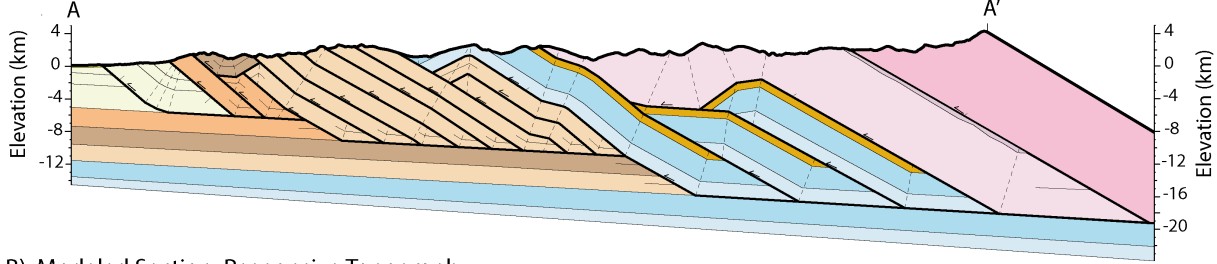

## B) Modeled Section: Responsive Topography

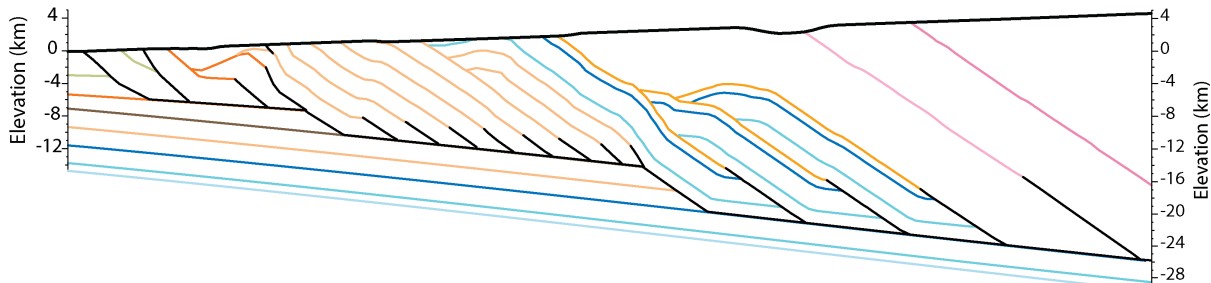

## C) Modeled Section: Static Topography

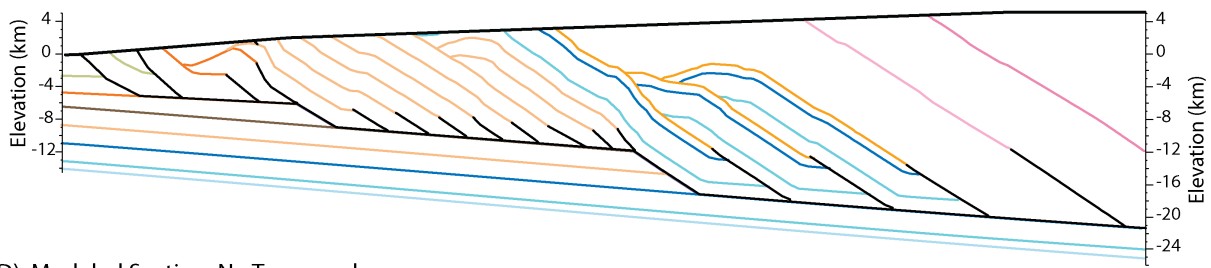

## D) Modeled Section: No Topography

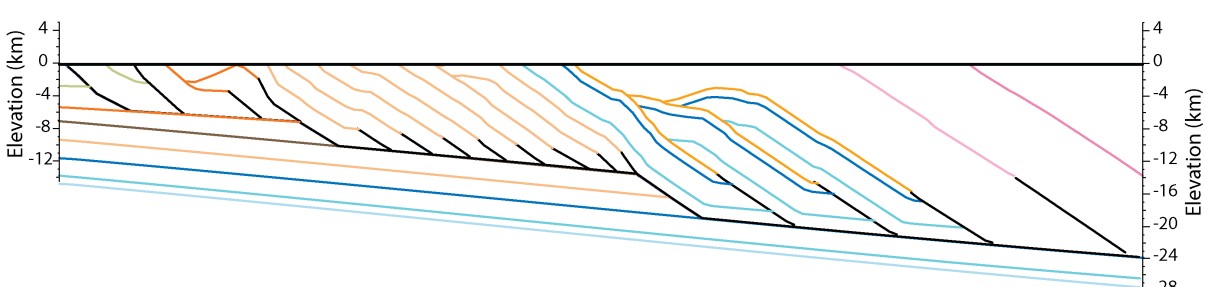

**Figure 4: Vertically exaggerated topographic model elevations compared to observed topography of the Trashigang section. (A) Long et al. (2011b) cross section is shown below the graph. Flexural-kinematic models shown were created using Split KT and (B) Responsive, (C) Static, and (D) No Topography.**

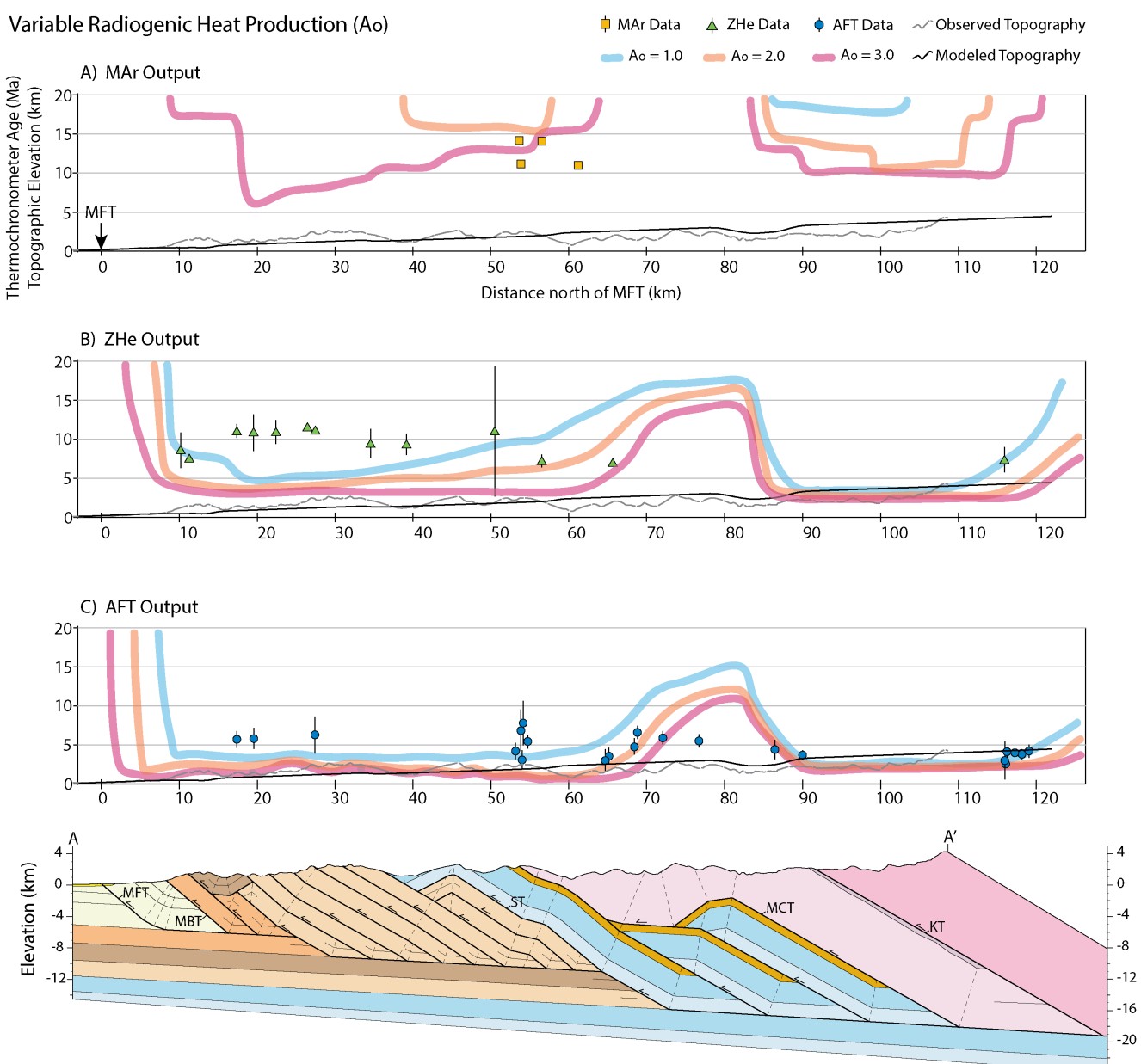

**Figure 5: Predicted (A) MAr, (B) ZHe, and (C) AFT cooling ages from Pecube using variable surface radiogenic heat production ($A_0$) values of 1.0 (blue), 2.0 (orange), and 3.0 (red) μW/m$^3$ compared to published cooling data. Other model variables are set as constant velocity, Split KT, Responsive Topography.**

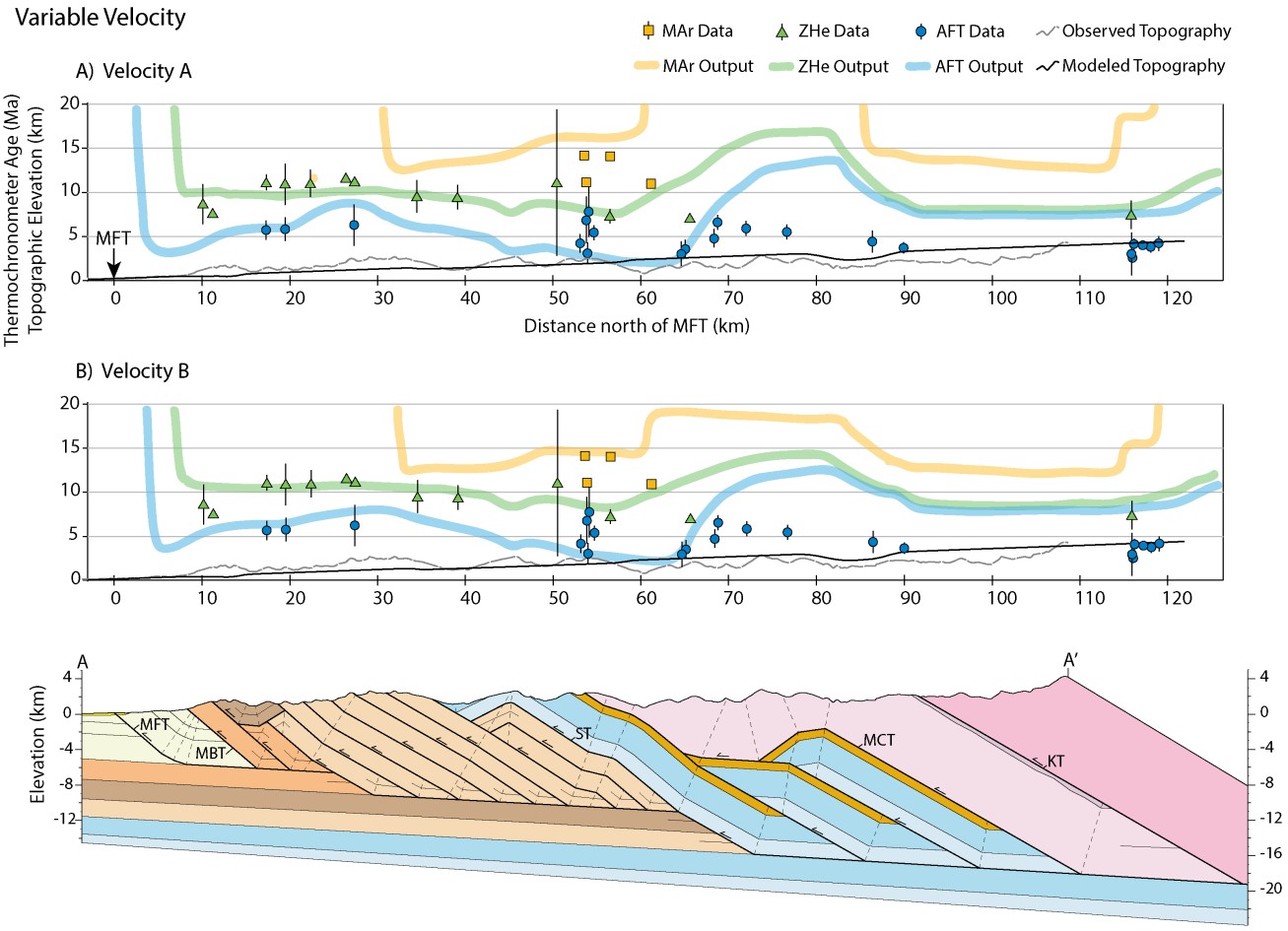

**Figure 6: Predicted MAr (yellow), ZHe (green), and AFT (blue) cooling ages using variable velocities (A) and (B) compared to published thermochronometer data. Other model variables are set as Split KT, Responsive Topography, and $A_0 = 2.5 \ \mu W/m^3$.**

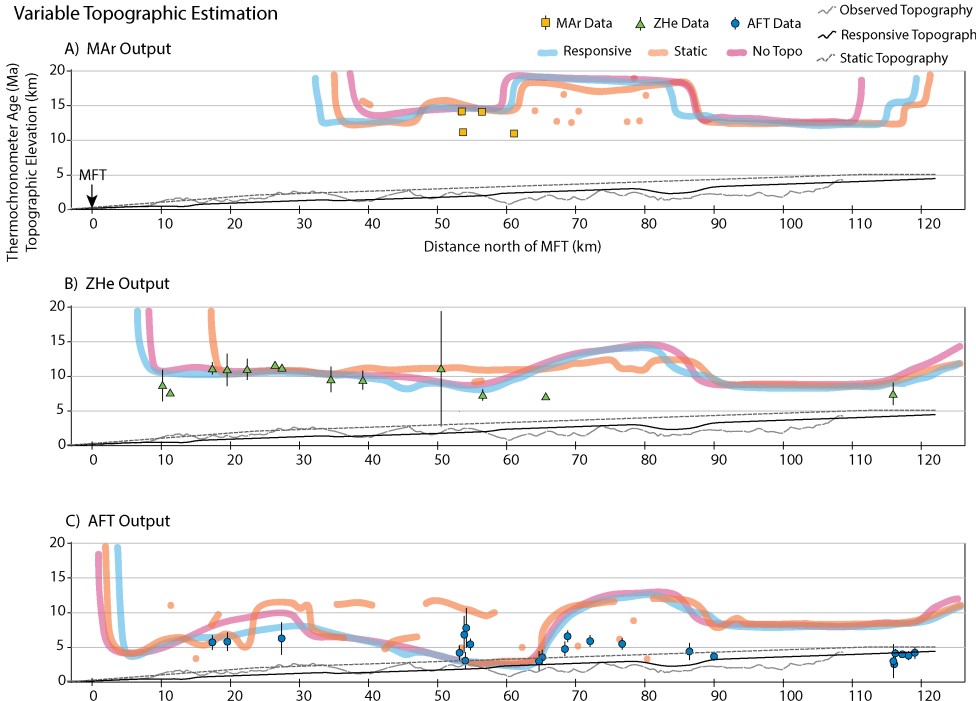

**Figure 7: Predicted (A) MAr, (B) ZHe, and (C) AFT cooling ages using Responsive (blue), Static (orange), and No Topography (red) models compared to published thermochronometer data. Other model variables are set as Split KT, Velocity B, and $A_0$ = 2.5 $\mu W/m^3$.**

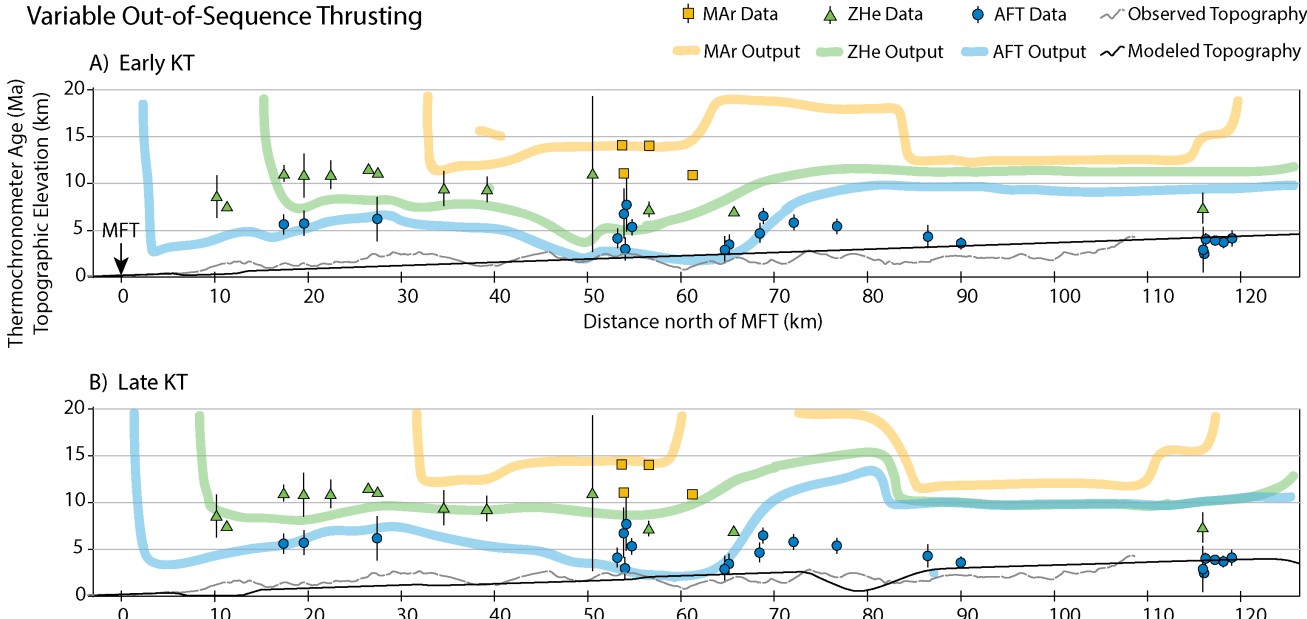

**Figure 8: Predicted MAr (yellow), ZHe (green), and AFT (blue) cooling ages using (A) Split KT, (B) Early KT, and (C) Late KT kinematic scenarios compared to published thermochronometer data. Other model variables are set as Responsive Topography, Velocity B, and $A_0 = 2.5$ µW/m$^3$.**

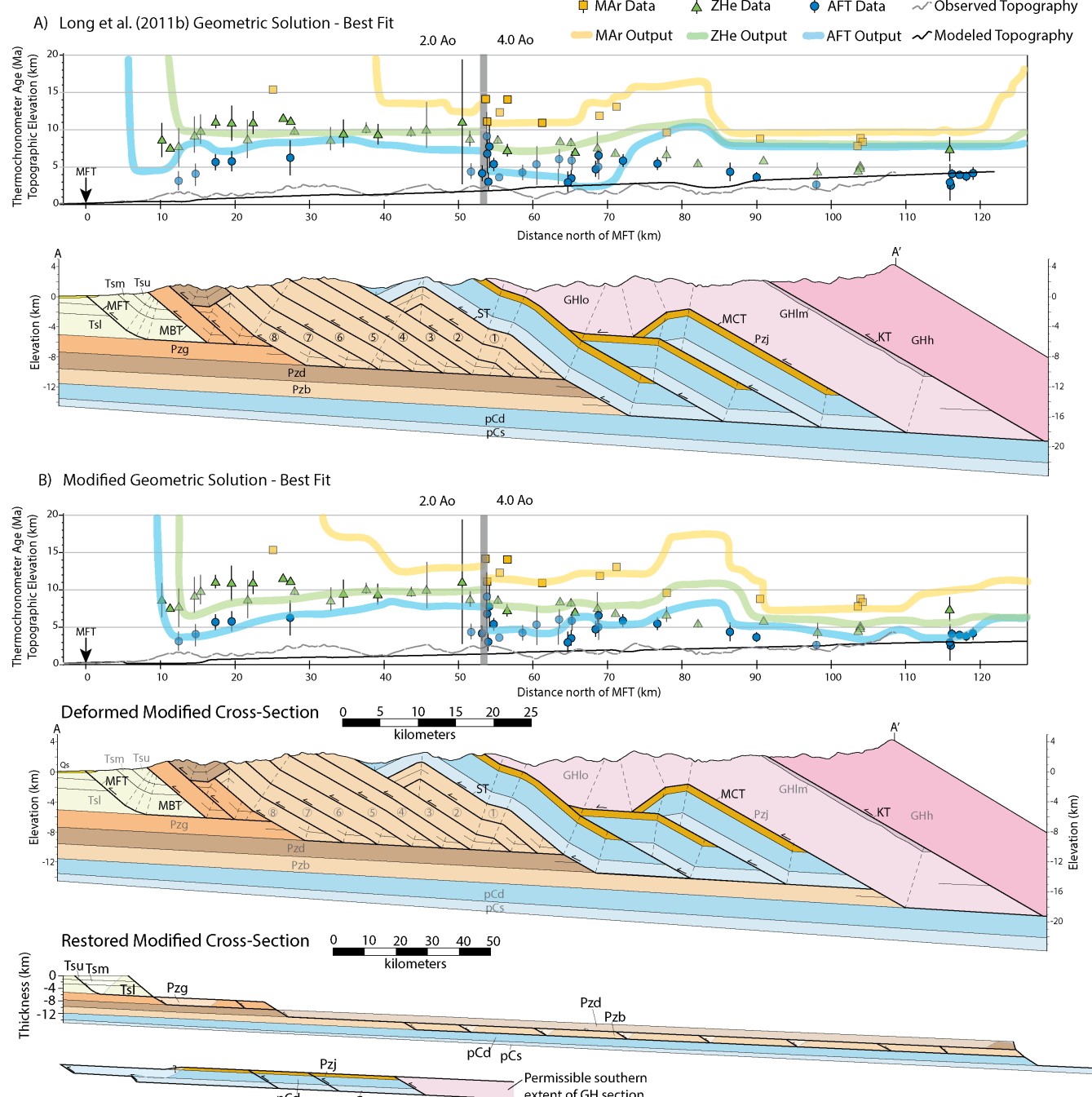

**Figure 9: Predicted MAr (yellow), ZHe (green), and AFT (blue) cooling ages using flexural-kinematic models of (A) the original geometric solution by Long et al. (2011b) with Responsive topography and velocity B, and (B) a modified geometric solution with Responsive topography and preferred velocity. The décollement ramp through the upper LH Baxa and Diuri units has been split, and the Baxa footwall ramp moved 35 km north. Published data include additional ages from the Kuru Chu line of section 30 km**

**west of the Trashigang section (Long et al., 2012) and indicated by transparent thermochronometer symbols. The grey vertical lines aligned with the location of the MCT shows the division between outputs from separate thermal models have been merged using 2.0 and 4.0 µW/m³ to the south and north of the MCT respectively. Other model variables are set as Split KT and Velocity B.**

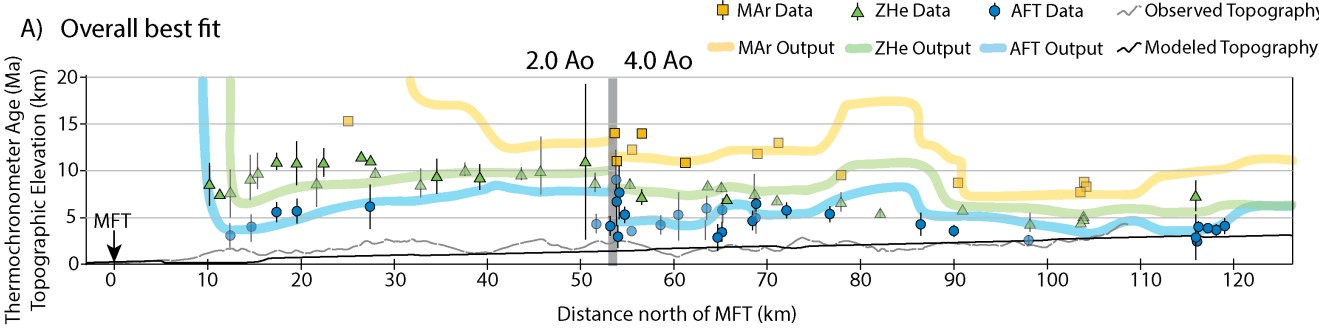

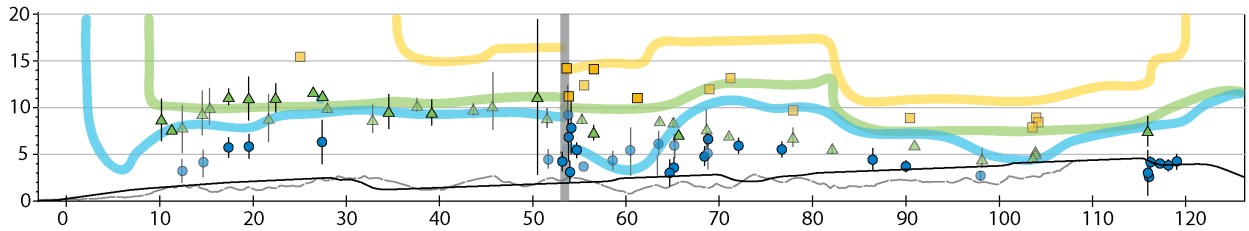

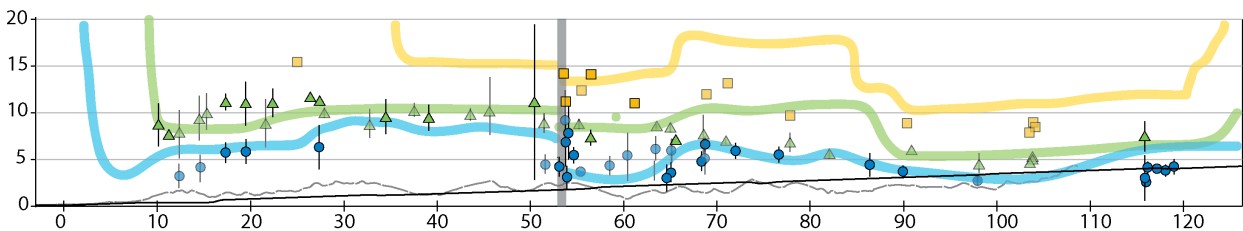

**Figure 10: Comparison of predicted MAr (yellow), ZHe (green), and AFT (blue) cooling ages between (A) the best-fitting thermokinematic model combination shown in figure 9b, (B) a well-matched flexural-kinematic model that yielded a thermal model with poorly fitting predicted ages, and (C) a poorly-matched flexural-kinematic model that produced a thermal model with well-fitting predicted ages (c). Published data include additional ages from the Kuru Chu line of section 30 km west of the Trashigang section (Long et al., 2012) and indicated by transparent thermochronometer symbols. Predicted ages are presented with combined thermal models using A$_0$ of 2.0 and 4.0 µW/m³. Flexural-kinematic models used Split KT and Responsive Topography.**

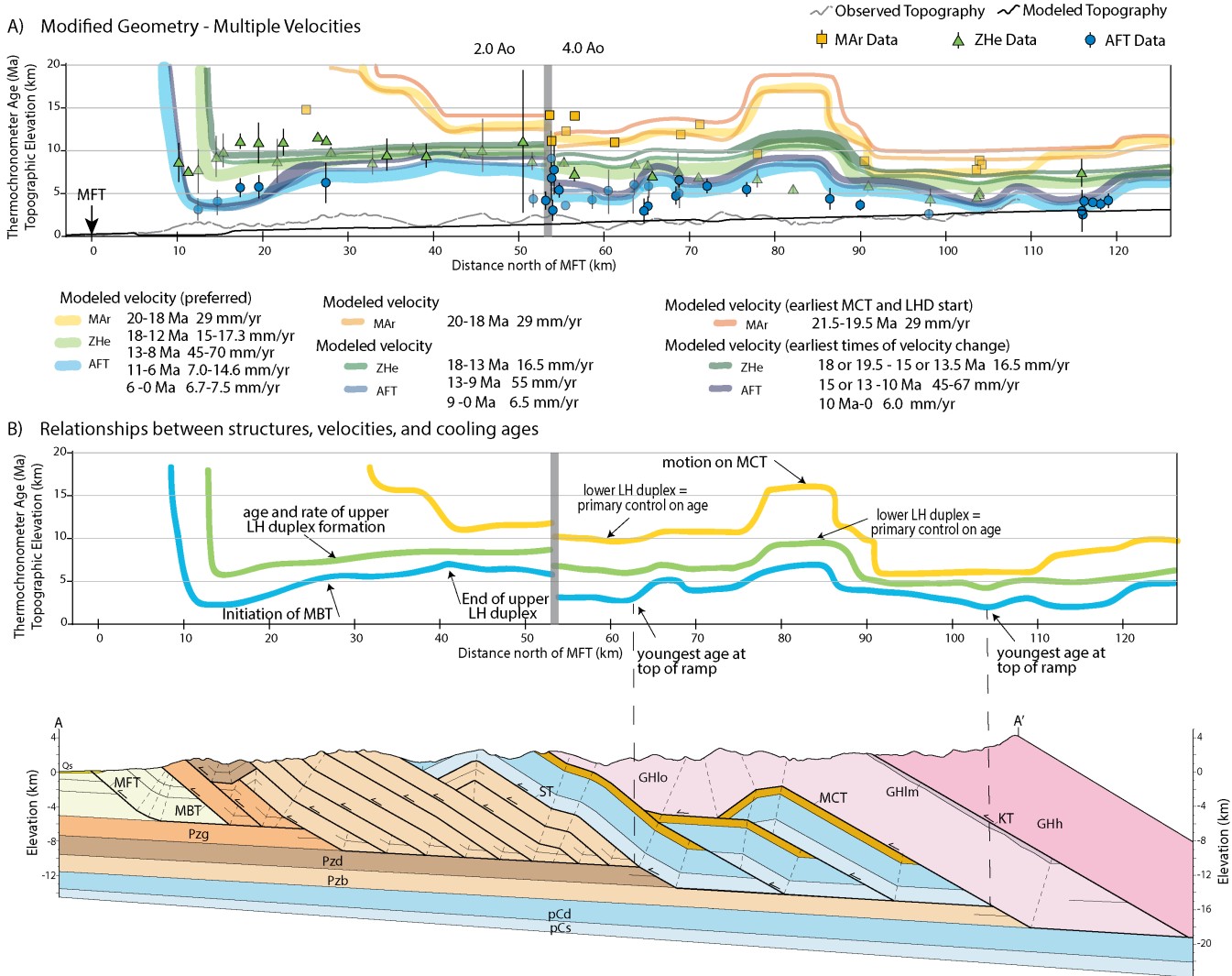

**Figure 11: A) Predicted MAr (yellow), ZHe (green), and AFT (blue) cooling ages for velocities tested using the modified Trashigang cross section geometry. Published data include additional ages from the Kuru Chu line of section, 30 km west of the Trashigang section (Long et al., 2012), which are indicated by transparent thermochronometer symbols. Predicted ages are presented with combined thermal models using $A_0$ of 2.0 and 4.0 μW/m³. Flexural-kinematic models used Split KT and Responsive Topography. B) Relationships between structures, velocities, and predicted cooling ages.**

**Flexural-Kinematic Model Output**

| Geometry & Kinematics | Topography Estimation | EET (km) | Crustal Density (g/cm³) | Foreland Basin Thickness (km) | Décollement Dip (°) | Surface Geology |
|---|---|---|---|---|---|---|
| Long et al. (2011b) | - | - | - | 5.6 | 4 | - |
| **Long et al. Geometry** | | | | | | |
| Split KT | Responsive | 65 | 2.60 | 5.2 | 5.1 | GOOD |
| Split KT | Static | 65 | 2.60 | 4.6 | 3.8 | over-eroded by 0.3 km at hanging wall of ST |
| Split KT | NoTopo | 65 | 3.20 | 5.1 | 4.6 | over-eroded by 0.9 km at Diuri Fm and by 2.3 km at hanging wall of ST |
| Early KT | Responsive | 65 | 2.60 | 4.3 | 5 | over-eroded by 0.5 km at Diuri Fm and hanging wall of ST |
| Early KT | Static | 65 | 2.60 | 4.5 | 4.4 | GOOD |
| Late KT | Responsive | 65 | 2.60 | 5.6 | 5.4 | under-eroded by 0.4 km at Diuri Fm, hanging wall of ST, and GH synform |
| Late KT | Static | 65 | 2.60 | 4.9 | 5.2 | under-eroded by 1 km at Diuri Fm |
| **Modified Geometry** | | | | | | |
| Split KT | Responsive | 70 | 2.60 | 5.7 | 4.5 | GOOD |

**Table 1: Comparison of the geologic constraints of the published Trashigang cross section to the final deformed cross section results of the flexural-kinematic models presented in this study.**

## Numerical model parameters

| Property/parameter | Model input value |
|---|---|
| **Material properties** | |
| heat production | |
| crustal volumetric heat production | 1.0 - 5.0  mW/m3 |
| e-folding depth of crustal heat prod. | 20 km |
| thermal conductivity | 2.5 W/m K |
| specific heat capacity | 800 J/kg K |
| crustal density | 2700  kg/m3 |
| mantle density | 3300  kg/m3 |
| **Numerical  properties** | |
| temperature at base | 1300°C |
| model base | 110 km |
| surface temperature at 0 km | 20 ° |
| atmospheric lapse rate | 0°/ km |
| kinematic grid spacing | 0.5 km |
| displacement increment | ~10 km |
| model domain | 730 x 110 x 5 km |
| horizontal node spacing (numerical model) | 0.5 km |
| vertical node spacing  (numerical model) | 1.0 km |
| model start time | 50 Ma |

**Table 2: Thermal and rock property parameters assigned as input for Pecube.**

**Displacement Ages & Rates**

**A. Long et al. (2011b) Geometric Solution**

**Split KT Models**

| Active Structure | Slip on Structure (km) | Total Shortening (km) | Constant Velocity Ao = 1.00, 2.00, 3.00 µW/m³ | | | Velocity A Ao = 2.25, 2.50, 2.75, 3.00 µW/m³ | | | Velocity B Ao = 2.00, 2.25, 2.50, 2.75, 3.00, 4.00 µW/m³ | | |
|---|---|---|---|---|---|---|---|---|---|---|---|
| | | | Start Age (Ma) | End Age (Ma) | Velocity (mm/yr) | Start (Ma) | End (Ma) | Velocity (mm/yr) | Start (Ma) | End (Ma) | Velocity (mm/yr) |
| MCT | 63.2 | 63.2 | 23.0 | 19.3 | 17.3 | 23.0 | 21.0 | 31.6 | 20.0 | 17.0 | 21.1 |
| Lower LH Duplex | 87.9 | 151.1 | 19.3 | 14.3 | 17.3 | 21.0 | 15.0 | 14.7 | 17.0 | 13.5 | 25.1 |
| KT | 25.0 | 176.1 | 14.3 | 12.8 | 17.3 | 15.0 | 14.3 | 37.3 | 13.5 | 13.2 | 74.6 |
| Baxa Duplex | 161.4 | 337.5 | 12.8 | 3.5 | 17.3 | 14.3 | 10.0 | 37.3 | 13.2 | 11.0 | 74.6 |
| KT | 20.0 | 357.5 | 3.5 | 2.3 | 17.3 | 10.0 | 6.7 | 6.0 | 11.0 | 7.3 | 5.4 |
| MBT | 26.5 | 384.0 | 2.3 | 0.8 | 17.3 | 6.7 | 2.2 | 6.0 | 7.3 | 2.5 | 5.4 |
| MFT | 13.4 | 397.4 | 0.8 | 0.0 | 17.3 | 2.2 | 0.0 | 6.0 | 2.5 | 0.0 | 5.4 |

**Early KT Models** — Velocity B, Ao = 2.25, 2.50, 2.75 µW/m³

| Active Structure | Slip on Structure (km) | Total Shortening (km) | Start (Ma) | End (Ma) | Velocity (mm/yr) |
|---|---|---|---|---|---|
| MCT | 63.2 | 63.2 | 20.0 | 17.0 | 21.1 |
| Lower LH Duplex | 87.9 | 151.1 | 17.0 | 13.0 | 22.0 |
| KT | 45.0 | 196.1 | 13.0 | 12.4 | 69.4 |
| Baxa Duplex (1-7) | 145.9 | 342.0 | 12.4 | 10.3 | 69.4 |
| Baxa Duplex (8) | 15.5 | 357.5 | 10.3 | 7.4 | 5.4 |
| MBT | 26.5 | 384.0 | 7.4 | 2.5 | 5.4 |
| MFT | 13.4 | 397.4 | 2.5 | 0.0 | 5.4 |

**Late KT Models** — Velocity B, Ao = 2.25, 2.50, 2.75, 3.00, 3.25, 3.50 µW/m³

| Active Structure | Slip on Structure (km) | Total Shortening (km) | Start (Ma) | End (Ma) | Velocity (mm/yr) |
|---|---|---|---|---|---|
| MCT | 63.2 | 63.2 | 20.0 | 17.0 | 21.1 |
| Lower LH Duplex | 87.9 | 151.1 | 17.0 | 13.0 | 22.0 |
| Baxa Duplex (1-7) | 145.9 | 297.0 | 13.0 | 11.0 | 73.0 |
| KT | 45.0 | 342.0 | 11.0 | 10.0 | 45.0 |
| Baxa Duplex (8) | 15.5 | 357.5 | 10.0 | 7.2 | 5.5 |
| MBT | 26.5 | 384.0 | 7.2 | 2.4 | 5.5 |
| MFT | 13.4 | 397.4 | 2.4 | 0.0 | 5.5 |

**B. Modified Geometric Solution**

| Active Structure | Slip on Structure (km) | Total Shortening (km) | Range of Velocities Tested Ao = 2.00, 2.25, 2.50, 2.75, 3.00, 3.25, 3.50, 3.75, 4.00, 5.00 µW/m³ | | | Range of Best-Fitting Velocities Ao = 2.00, 4.00 µW/m³ | | |
|---|---|---|---|---|---|---|---|---|
| | | | Start Age Range (Ma) | End Age Range (Ma) | Velocity Range (mm/yr) | Start Age Range (Ma) | End Age Range (Ma) | Velocity Range (mm/yr) |
| MCT | 58.0 | 58.0 | 21.6 - 19.9 | 19.6 - 17.9 | 29.0 | 20.4 - 19.9 | 18.4 - 17.9 | 29.0 |
| Lower LH Duplex | 78.0 | 136.0 | 19.6 - 17.9 | 14.9 - 13.2 | 15.0 - 17.5 | 18.4 - 17.9 | 13.4 - 13.2 | 15.0 - 17.3 |
| KT | 25.0 | 161.0 | 14.9 - 13.2 | 14.4 - 11.8 | 15.0 - 68.4 | 13.4 - 13.2 | 13.1 - 11.8 | 15.0 - 68.0 |
| Baxa Duplex (1-7) | 146.0 | 161.0 | 14.4 - 11.8 | 11.2 - 9.6 | 45.0 - 70.0 | 13.1 - 11.8 | 11.0 - 9.6 | 45.0 - 70.0 |
| Baxa Duplex (8) | 53.0 | 360.0 | 11.2 - 9.6 | 10.0 - 7.3 | 8.0 - 67.0 | 11.0 - 9.6 | 8.6 - 7.3 | 8.0 - 65.0 |
| KT | 20.0 | 380.0 | 10.0 - 7.3 | 6.7 - 4.7 | 6.0 - 17.0 | 8.6 - 7.3 | 6.0 - 5.3 | 7.0 - 14.6 |
| MBT | 29.0 | 409.0 | 6.7 - 4.7 | 1.8 - 1.5 | 6.0 - 7.5 | 6.0 - 5.3 | 1.6 - 1.5 | 6.7 - 7.5 |
| MFT | 11.0 | 420.0 | 1.8 - 1.5 | 0.0 | 6.0 - 7.5 | 1.6 - 1.5 | 0.0 | 6.7 - 7.5 |

**Table 3: Combinations of heat production values, deformation ages and rates tested for each flexural-kinematic model.**