# Peer review of "Testing the effects of topography, geometry and kinematics on modeled thermochronometer cooling ages in the eastern Bhutan Himalaya"

_Solid Earth, 2017_

## Referee Comment (RC1) · P.A. van der Beek (Referee) · 19 Dec 2017

Gilmore et al. present a sensitivity analysis for a recently developed modelling approach in which structural restoration is combined with forward thermal-kinematic modelling to predict thermochronometer ages in fold-thrust belts, and subsequently use these ages to constrain the timing and rate of thrust-(sheet) motion in such settings. This is a promising approach, which h is being developed by several research groups separately (e.g., Almendral et al., 2015; Erdös et al., 2014; McQuarrie and Ehlers, 2015; 2017). However it still faces challenges, in particular how to take into account the topographic evolution through time and how to handle the large degree of freedom

in the models. The present manuscript explores some of these challenges, in particular the effect of material properties (heat production rates), reconstructed geometry and kinematics, and the topographic history, which all influence the predicted thermal histories significantly but are very difficult to constrain. It is therefore a useful contribution to the still small but growing number of papers on this subject, and I would recommend publishing this in Solid Earth after moderate revisions.

I have two major comments and a number of smaller, more specific comments on this manuscript. The first major comment concerns the context of this study and what is exactly new in it. When I started reading this, this was not very clear for me. Long et al. (2012) presented the structural cross-section and thermochronology data used here, as well as similar data for the parallel more westerly Kuri Chu cross-section. McQuarrie and Ehlers (2015) modelled the data for the Kuri Chu cross-section in a similar manner to what is done here. What is new in this manuscript is the modelling of the (eastern) Trashigang cross-section. This is a valuable exercise in itself, and the comparison of the outcomes of the two modelling exercises in enlightening (see below), but I think it would be useful if the authors presented this context and the relationship of this study with previous work straight up in the introduction, so that readers are not left wondering what is new or different here with respect to previous work by the same group of authors.

My second comment concerns the inferred history of shortening rates; in particular the strong variability in these rates that the analysis suggests. I have been intrigued by this outcome since the initial paper by Long et al. (2012). I reviewed that paper at the time and already queried the authors about the robustness and implications of that finding but am still struggling to understand it. Starting from what we know (and progressing toward lesser constrained inferences): the modern convergence velocity between India and Tibet is ~20 mm/y; the total India-Asia convergence rate is about twice that. If we accept the results of Molnar & Stock (2009), India-Asia convergence rates have decreased since 20 Ma; from 54-83 mm/y before 11 Ma to 34-44 mm/y after

that, for points in the NW and NE corner of the Indian subcontinent respectively. That total India- Asia convergence rate should be distributed between far-field deformation in the Tibetan plateau and its northern borders, shortening in the Himalaya, and under-thrusting of the Indian plate beneath Tibet. It is interesting, and reassuring, to note that most of the tested models predict shortening rates in the order of 5-6 mm/y in the last ~10 Ma, which is consistent with estimated "overthrusting" rates in simpler thermo-kinematic models used to predict thermochronology ages (e.g. Brewer and Burbank, 2006; Whipp et al., 2009; Robert et al., 2009; 2011; Herman et al., 2010; Coutand et al., 2014, and others). Any increase in shortening rates up to the total India- Tibet convergence rate of ~20 mm/y could potentially be explained by temporally variable partitioning between "overthrusting" and "underthrusting"; since these concepts are re-ally defined by a particular frame of reference only (which is in my view controlled by the erosional efficiency in the Himalaya), that could be plausible and possibly linked to temporal variations in erosional efficiency. If one wants to invoke further increases up to the India-Asia convergence rate, that would only be possible by temporally transferring far-field deformation to the Himalaya, but it remains in the realm of possibilities. The inferred rates of ~70 mm/y during building of the Upper Lesser Himalayan duplex are more problematic, because – if true – they would necessarily imply north-south exten-sion in other parts of the Himalaya-Tibet system, for which there is very little evidence. The inferred reconstruction requires significant amounts of shortening to build this du-plex (at least 150 km or ~1/3 of the total shortening since 20 Ma according to Fig. 3) and I wonder whether a more conservative structural solution would not be possible to fit the surface observations for this duplex. In any case, the preferred models with variable shortening velocities pose significant questions, which should be addressed more directly. The reader is really left wondering how well resolved these shortening histories are, given the significant number of unconstrained parameters in the models. Some of the specific comments below refer to these unknowns.

Overall, the paper is fairly well written and illustrated. On a number of occasions, phrases don't run because a verb is missing or because of singular/plural confusions.

A certain number of typos also remain. All of these can be weeded out by some careful editing. The use of some internal "modelling jargon" like "Python topography", "Split KT" etc. does not add to the general understanding of the manuscript – the authors might want to find some more eloquent terms to describe these modelling settings.

Specific comments, tied to page and line number:

p. 1 l. 7-10: the first two phrases of the abstract do not really set up the problem in a very clear manner or "draw" the reader into the problem – you may want to consider rewriting these into something more clear and specific.

p. 2 l. 13-20: this first paragraph of the "Geologic background" section looks a bit lost on its own; it is not very informative (why is the onset of motion on the MCT important here?) and could easily be combined with the following "Tectonostratigraphy" section. The Daniel et al. (2003) and Tobgay et al. (2012) references are missing in the reference list.

p. 3 l. 20-21: how were the data exactly projected into the cross-section? This is a critical step, as the ages (in particular for the low-temperature systems) will be influenced by the local topography. See further comments below.

p. 3 l. 30: why do the ZHe ages require "rapid" cooling? This inference can only be drawn by comparing them to other thermochronometer data, or by assessing age-elevation profiles for instance.

p. 3 l. 32: three ZHe cooling ages north of the MCT are shown on the cross-section (but only two on the map?). Also, the cross-section of Fig. 2 gives the impression that the samples between ~57-65 km are from the lower Greater Himalayan sequence, while the map shows they are from the upper. Maybe you should sketch in some of the geology above the topography to make this clearer. This also brings us back to the question above of how these data were projected into the cross section. What was their imposed elevation? Simply plotting them on the topography in the cross-section

puts them on a much lower structural level than where they actually are!

p. 4 l. 5-9: why do you take this approach? It is easy enough to model the individual data using the combined Move/Pecube approach . . .

p. 4 l. 17-18: the question here is obviously: "how was the new topography obtained?" this is discussed further on – you may want to refer the readers to this later discussion here.

p. 4 l. 26-27: Note that a subsequent similar model by the same authors (Hammer et al., GRL 2013) comes up with much lower estimates for the elastic thickness in Bhutan (< 25 km) than in Nepal.

p. 5 l. 2: here you could reference some of the previous studies using the same approach.

p. 5 L. 27-31: a self-consistent approach would be to use a critical-taper topography in the models – it is not clear if the "Python topography" is based on such an approach, but the link between the imposed topography and a critically tapered wedge model could be outlined here.

p. 7 l. 6-17: see general comment on variable shortening rates above. More justification and discussion of these rates is needed.

p. 7 l. 16: it seems that this is the first time the Kuru Chu section is mentioned; it hasn't been introduced previously (but should be).

p. 7 l. 19 (and numerous other occurrences): why do you call the reconstructions "flexural models"? This is surprising and confusing, as flexure is only one component of these models; the structural reconstruction is at the heart of them. You could call them "kinematic models" or something like that.

p. 7 l. 30: the INDEPTH lines were shot in the Yadong rift, which overlies the Yadong cross-structure – a probably important lateral ramp in the Main Himalayan decollement.

Is the 4° dip you cite here relevant for the decollement west or east of the Yadong structure? In any case, this would be valid for western Bhutan and not necessarily for eastern Bhutan. It is not obvious that comparing the decollement dips with data that are not from the same region is very informative, given the probable lateral segmentation of the MHT.

p. 8 l. 1-4: this is counter-intuitive. The flexural response should be driven by the topographic loading, not by the kinematic scenario. Therefore, if the different kinematic models lead to differences in flexural loading profiles, it must be because the (imposed) topographic response to the kinematics is different between these models.

p. 8 l. 15-19: why do you not simply use the present-day topography as the final topography in the model? This is a known entity, and at least that would help in comparing kinematic and thermal histories at the right structural and topographic levels for the data points.

p. 8 l. 22-24: this phrase is hard to read and also appears counter-intuitive. In the critical-wedge model, the surface topography ($\alpha$) and decollement dip ($\beta$) are linked through the critical taper angle (which itself depends, among other things, on $\beta$). Therefore, it might be more self-consistent to try to find a surface topography angle that corresponds to the critical taper for each time step (and degree of topographic loading). This would be an iterative approach, but I'm sure it can be done. See comment on p. 5 l. 27-31 above.

p. 9 l. 11: you may have modified your version of Pecube, but in the "standard" model, heat production is constant with depth, so that "surface heat production" is a bit of a confusing term in this context.

p. 9 l. 15: this seems a fairly obvious result, since the kinematics of the models do not change, only the thermal field. The samples have the same "normalized" thermal histories; the temperatures are simply somewhat higher throughout for the models with higher heat production.

p. 9 l. 19: "ages" not "rocks", I think.

p. 10 l. 3-5: a bit of a rambling phrase that is difficult to read/understand.

p. 10 l. 24: "later" not "earlier" I think?

p. 10 l. 32-33: there are many free parameters in these models: not only an infinite number of shortening-rate histories, but also significant degrees of freedom in the imposed structure and the topographic evolution. I fully understand and appreciate the difficulties in exploring this complex parameter space, but how robust are the inferred rates really? This is not obvious, and given the important implications of the shortening-rate history, this should be discussed. An alternative approach would be to not allow shortening rates that are greater than the plate-scale convergence rates at any time (i.e. use the plate-convergence rates as a constraint) and try to find models that can explain the data using this constraint.

p. 11 l. 10-11: why is this your expectation? The erosional history would depend on the topographic history through time, rather than the final topography. In the no-topography scenario, if I understand well, there is no topographic change through time. If in the other topographic scenarios topography diminishes locally in the final timesteps, this will predict younger ages.

p. 11 l. 14-15: a list of 6 adjectives ("Python topography model fully reset Mar ages") followed by another of 4 . . . Maybe rewrite?

p. 11 l. 20-23: this is an important point but it also seems fairly obvious. It clearly points to the need of a self-consistent treatment of topographic evolution. The best way forward may be to combine these models with simple surface-process models to erode the topography through time.

p. 12 l. 2: I think you are discussing MAr ages specifically here? May be useful to state this.

p. 12 l. 30: "older ages" seems more correct than "earlier ages" in this context.

p. 12 l. 31: you have been calling this the MHT throughout the manuscript. Better stick to this acronym so as not to confuse the readers.

p. 13 l. 1-2: another somewhat rambling phrase . . .

p. 13 l. 5: this ramp is rather located at ∼90 km in the present-day geometry (Fig. 2)?

p. 13 l. 12-15: is the cross-section of Fig. 9 still balanced? There is all of a sudden 35 km more Baxa group in this cross-section, while the rest of it has not been modified. Could these additional 35 km be found by reducing shortening in the upper LHS duplex? In that manner you might also be able to reduce the problematic shortening rates necessary to produce this (and the associated ZHe ages).

p. 13 l. 25-29: this is problematic. First of all, you change two major inputs to the model (structural geometry and heat production) at the same time here, while previously you have carefully only changed one parameter at a time. Second, you introduce spatially variable heat production here, which you did not do previously and which could have led to better fits in the previous models. This is a large change in the thermal structure and it should be justified. Although I am sympathetic to the fact that heat production could be significantly higher in the GHS than in the LHS, to really model this properly you should ascribe heat-production values to the different units, and advect these with the units.

p. 13 l. 31-32: OK, but we are left wondering how much of this improved fit can be ascribed to the new structure and how much to the increased heat production. p. 14 l. 5-6: following up on the previous comment; can the data really tell the difference between the improved structural geometry and the increased heat production? There is very little data in the "bump" region. You use a simple visual comparison of predicted and observed ages; it would be useful to provide a more objective and quantitative comparison to back up inferences such as this.

p. 16 l. 10-12: this is introducing yet another unconstrained parameter. I am not sure it

is the best strategy to further complexify the models to improve the fit; this seems like a bit of a "flight forward". A more complete sensitivity and resolution analysis might be a more productive way forward.

p. 17 l. 9-10: "the amount of exhumation in this model is just at the amount necessary to reset AFT ages" is strange and apparently incorrect. The ages record cooling through the closure temperature at a certain time in the past. The thermal structure is going to affect that time, but the total amount of exhumation is much larger than the AFT closure depth it would seem.

p. 18 l. 10-15: A bunch of hard-to-read phrases that are in need of a few commas. Also, "after 13 Ma" would be better than "longer than" and replace the colloquial "till" by "until".

p. 18 l. 15-20: another potential issue that is not discussed concerns the diffusion kinetics of He in zircon. Recent work has shown that the effective closure temperature of the ZHe system can vary from as low as ~120 °C to as high as ~240 °C as a complex function of the degree of $\alpha$-damage (e.g. Guenthner et al., 2013). If you have underestimated the ZHe closure temperature (I suppose you are using the "standard" ZHe diffusion parameters built into Pecube) you could significantly underestimate the duration of shortening on the upper LH duplex, and thereby overestimate the shortening rates.

p. 18 l. 25-28: the first part of this argument is somewhat circular, since the McQuarrie and Ehlers (2015) scenario was input in the models here, without extensively testing all other potential scenarios. So the fact that the model predicts these variations in rates should not come as a surprise. In contrast, the dissimilar timing between the two sections that are only ~25 km apart should be worrying. How can the same structure be active at time intervals that are several million years different between two adjacent locations? Again, the reader is left wondering how much of this difference could be due to variable diffusion kinetics?

p. 19 l. 2: given the numerous unexplored degrees of freedom in the models, it appears risky to assess the validity of the data based on the modelling outcomes.

p. 20 l. 1: not sure what is meant with this phrase; what is "the spatial nature of thermochronometry"?

Figures

Fig. 1: the inset geological map of Bhutan (panel B) is very small and not very readable. You should either increase its size or decrease the amount of detail on it. Also, in the legend of the main panel (C), the Chekha Formation should be above the Greater Himalaya to keep all units in their structural order. Finally, it would help the reader if the colours used for the different thermochronometers were consistent between this figure and the following.

Figs. 5-10: much more data appears to be plotted in these figures than in Figs. 1 and 2. What do the lighter-coloured data points refer to? For clarity it would be better to take them out. In Fig. 7, why does the "template topography" model not predict AFT ages everywhere?
* * *

---

## Referee Comment (RC2) · Anonymous Referee #2 · 3 Jan 2018

Review of the manuscript entitled "Testing the effects of topography, geometry and kinematics on modeled thermochronometer cooling ages in the eastern Bhutan Himalaya" by Gilmore et al.

This manuscript analyzes the impact of variable radiogenic heat production, convergence rate, topographic estimates and out-of-sequence thrusting in determining the pattern of previously published thermochronologic ages along a transect across the Bhutan Himalaya. The authors utilize their results to validate a revised cross-section geometry of the study region.

The manuscript is generally well written. The topic is of potential interest for a broad

international audience. However, it would benefit from a more comprehensive discussion of the whole range of geologic processes that may have an impact on the thermochronologic record of the study area.

The modelling approach utilized in this work is based on flexural and thermal-kinematic models. The authors sequentially deform the study cross section, and apply flexural loading and erosional unloading at each step to develop a high-resolution evolution of deformation, erosion, and burial over time. In other words, their approach only considers relatively shallow geologic processes. Deeper tectonic processes (e.g., channel flow exhumation and slab breakoff) that may also affect the thermochronologic record, especially higher temperature systems such as Ar-Ar on mica, are not discussed. This may puzzle part of the potential readership. I suggest to improve on the discussion, and possibly the modelling, in order to include these issues.

The dataset of previously published thermochronologic ages, which is utilized as a benchmark for modelling, is not homogeneous. AFT and ZHe data are available in most of the transect, but Ar-Ar data are not. This would suggest more caution in the conclusions based on modelling results. Moreover, these ages are invariably interpreted as cooling ages during exhumation across the closure temperature of the Ar-Ar system. Petrologic studies demonstrate that micas in metamorphic rocks often preserve disequilibrium textures, and their Ar-Ar age may thus record fluid-induced recrystallization below the closure temperature, rather than monotonic cooling (e.g., Villa 1998 - Terra Nova). Why mica Ar-Ar ages are so different in samples that are so close each other? What is the potential role of recrystallization during deformation? These issues should be discussed in the revised main text.

Some of the findings of the authors are not surprising for an active orogenic belt such as the Himalaya, notably the minor effect of radiogenic heat production and topography compared to tectonics. Nevertheless, the authors' conclusion should be supported by more robust thermochronologic data. The addition of a new ramp under the Greater Himalaya does better explain available thermochronologic ages. However, this is just

one of the possibilities, given the degree of freedom of the models.

Is the stratigraphy predicted by modelling consistent with the geologic record? This may provide independent constraints to the reconstructions illustrated in this work, that are prone to remain otherwise speculative. I suggest to describe in more detail the stratigraphic evolution of the foreland basin, as well as all of the other geologic evidence that may be useful to support the authors' conclusions.

The abstract should be improved. The first two sentences are not relevant to introduce the focus of the manuscript. The Introduction and section 2.1 are biased by excessive self-referencing.

I will be happy to read a revised version of this potentially interesting manuscript.

---

## Referee Comment (RC3) · D. Grujic (Referee) · 11 Jan 2018

Dear Colleagues In this manuscript the authors present results of sensitivity of predicted thermochronological age distribution on several parameters: prescribed topographic evolution, geometry of the basal detachment and kinematics of the related fold-and-thrust belt and crustal heat production. The authors conclude that "this study presents a successful approach for using thermochronometer data to test the viability of a proposed cross section geometry based on forward models of the kinematic, exhumational, and thermal history of an area". I fully agree with this statement but have several comments that could help authors improve the manuscript and help reader

better evaluate the contributions. I concur with the comments by referee #1 and try not to repeat them here. I apologise for several self-citations, but my research group has been working in the area and applying similar research techniques since couple of decades.

General Comments 1. The general limitation of the kinematic models is that the geometry and kinematics is prescribed—therefore despite their best efforts dependent on authors' interpretation. I agree that this is still the best approach to interpret the spatial pattern of thermochronological data, and couple of authors of this manuscript have made significant progress with their previous publications (McQuarrie and Ehlers, 2015) in reducing these limitations. Unfortunately, the additional problem with the Pecube is that it cannot generate simultaneous movement on faults with opposite sense of slip. In the Himalaya, and in particular for the GHS, the cooling and exhumation were affected by the simultaneously motion along the MHT at the base and the South Tibetan Detachment (STD) at the top. The STD in the eastern Himalaya was active as a ductile shear zone until 11 Ma, which is half of the period of the here presented experiments. Could the "tectonic denudation" affect the cooling pattern of the northern part of the section?

2. The shape of isotherms and their effect on the cooling rates. Himalaya are an active contractional orogen, therefore, the isotherms are deformed and the geothermal gradient is not constant in space and time. Was this accounted for in the experiments when calculating the eroded material or when calculating the exhumation rates? For example the same rock uplift rate, minus same surface erosion rate will not yield the same cooling rate. Therefore because the exhumation rates are based on thermochronology, i.e., cooling rates, thermochronological data cannot be simply converted into exhumation rates based on an assumed geothermal gradient. The exhumation rates will depend on local instantaneous geothermal gradient at different times. This is not discussed in the manuscript. 3. The authors write that they have performed a sensitivity analysis. However they have performed a limited number of experiments changing one or two

parameters at the time (I concur with the related comments by referee #1). However it would have been better to perform a systematic search through the parameter "space" by providing the ranges of variables and searching for the most optimal value – the lowest misfit. I agree that this is a very time consuming approach, which requires tens of thousands of experiments. However this is the only approach that can provide a statistically relevant evaluation of any of the parameters. Pecube produces posterior probability density functions (PPDFs) for each model parameter, (Braun, J., P. Van Der Beek, P. Valla, X. Robert, F. Herman, C. Glotzbach, V. Pedersen, C. Perry, T. Simon-Labric, and C. Prigent (2012), Quantifying rates of landscape evolution and tectonic processes by thermochronology and numerical modeling of crustal heat transport using PECUBE, Tectonophysics, 524-525, 1–28, doi:10.1016/j.tecto.2011.12.035. ) I admit that I do not know if this can be implemented by the technique presented here (combination of Pecube thermokinematic modeling and Move kinematic modeling). 4. The GHC is not a trust sheet-the rocks in this lithotectonic units were affected by pervasive and heterogeneous ductile deformation. Similarly the MCT is not a fault but a several kilometers thick ductile shear zone with mylonites derived both from footwall block rocks and the hanging wall block rocks. All these rocks deformed as visco-elasto-plastic thermally activated materials and ought to be modeled as such not as Mohr-Coulomb materials. I do not question the applicability of cross section balancing and thermokinematic modeling for the rocks and structures that were dominantly deformed as the latter mechanisms. Therefore the particle displacement paths were not as simple as implemented by thermal-kinematic models. In conclusion, these models are applicable for the period after the cessation of pervasive ductile deformation. This is regardless weather the lithotectonic unit was emplaced according to the channel flow tectonic mode or to the classical fold nappe mode. In either case the pervasive ductile deformation occurred before the thermochronological record used here. Finally, the thermochronological data presented here and available in general cannot constrain the tectonic processes that occurred before them. 5. The authors analyse and discuss the effect of the thermophysical properties of the rocks on the spatial pattern of cooling

ages. However only the values of heat production were changed (2 and 4 $\mu$W/m3). However the thermal properties control the Péclet number, which dictates how strongly are the isotherms deflected because of the thrusting. This furthermore implies that the thermal properties have to include the study of sensitivity on thermal conductivity, heat capacity and density of the rocks.

Specific Comments 1. Valla et al. [2010] have shown that relief development must be 2–3 times faster than the background exhumation/erosion rate to be recorded and quantitatively extracted from thermochronological data. Valla, P., F. Herman, P. A. van der Beek, and J. Braun (2010), Inversion of thermochronological age-elevation profiles to extract independent estimates of denudation and relief history—I: Theory and conceptual model, Earth Planet. Sci. Lett., 295, 511–522. Please comment in your manuscript in the relevant places. 2. What is the evidence in the field (i.e., petrological) for the burial by Kakhtang thrust? Kakhtang thrust appears very steep therefore the burial rate might not be high. In addition the KT emplaced some of the hottest rocks in the Himalaya therefore the isotherms might have been disturbed during its activity, in other words heating and cooling does not need to imply burial and exhumation. 3.

Technical corrections a) Vertical uplift and vertical exhumation. Both rock and surface uplift and exhumation concern the vertical component of the particle displacement (in three different reference frames). Therefore word vertical is superfluous. However one must make difference between rock uplift and surface uplift, in particular in an article like this one where both processes are discussed. Please adhere strictly to the definitions by England and Molnar, 1990. Surface uplift, uplift of rocks, and exhumation of rocks. Geology, 18(12), pp.1173-1177. b) There is no process named "surface radiogenic heat production". Please correct the wording accordingly in the entire document.

All the above comments and further technical comments are in the annotated file.

Please also note the supplement to this comment:
https://www.solid-earth-discuss.net/se-2017-117/se-2017-117-RC3-supplement.pdf

[Figure]

**Supplement:**

[revised manuscript text omitted]

---

## Author Comment (AC1) · 11 Feb 2018

We appreciate the time and energy that reviewer 1 put into the evaluation of our manuscript. The comments and questions were insightful and addressing them has improved the quality and the clarity of the presented science. We have arranged our response by 1) reiterating the comments of the reviewers 2) providing our response and clarifying where the comment was addressed in the revised manuscript.

RC1 Gilmore et al. present a sensitivity analysis for a recently developed modelling approach in which structural restoration is combined with forward thermal-kinematic modelling to predict thermochronometer ages in fold-thrust belts, and subsequently

use these ages to constrain the timing and rate of thrust-(sheet) motion in such settings. This is a promising approach, which h is being developed by several research groups separately (e.g., Almendral et al., 2015; Erdös et al., 2014; McQuarrie and Ehlers, 2015; 2017). However it still faces challenges, in particular how to take into account the topographic evolution through time and how to handle the large degree of freedom in the models. The present manuscript explores some of these challenges, in particular the effect of material properties (heat production rates), reconstructed geometry and kinematics, and the topographic history, which all influence the predicted thermal histories significantly but are very difficult to constrain. It is therefore a useful contribution to the still small but growing number of papers on this subject, and I would recommend publishing this in Solid Earth after moderate revisions. I have two major comments and a number of smaller, more specific comments on this manuscript. The first major comment concerns the context of this study and what is exactly new in it. When I started reading this, this was not very clear for me. Long et al. (2012) presented the structural cross-section and thermochronology data used here, as well as similar data for the parallel more westerly Kuri Chu cross-section. McQuarrie and Ehlers (2015) modelled the data for the Kuri Chu cross-section in a similar manner to what is done here. What is new in this manuscript is the modelling of the (eastern) Trashigang cross-section. This is a valuable exercise in itself, and the comparison of the outcomes of the two modelling exercises in enlightening (see below), but I think it would be useful if the authors presented this context and the relationship of this study with previous work straight up in the introduction, so that readers are not left wondering what is new or different here with respect to previous work by the same group of authors.

[Reply] Introduction was revised to highlight new contributions and improve context with previous work. In particular we describe what is new in the introduction; p. 2 lines 13-21.

My second comment concerns the inferred history of shortening rates; in particular

the strong variability in these rates that the analysis suggests. I have been intrigued by this outcome since the initial paper by Long et al. (2012). I reviewed that paper at the time and already queried the authors about the robustness and implications of that finding but am still struggling to understand it. Starting from what we know (and progressing toward lesser constrained inferences): the modern convergence velocity between India and Tibet is ∼20 mm/y; the total India-Asia convergence rate is about twice that. If we accept the results of Molnar & Stock (2009), India-Asia convergence rates have decreased since 20 Ma; from 54-83 mm/y before 11 Ma to 34-44 mm/y after that, for points in the NW and NE corner of the Indian subcontinent respectively. That total India- Asia convergence rate should be distributed between far-field deformation in the Tibetan plateau and its northern borders, shortening in the Himalaya, and under-thrusting of the Indian plate beneath Tibet. It is interesting, and reassuring, to note that most of the tested models predict shortening rates in the order of 5-6 mm/y in the last ∼10 Ma, which is consistent with estimated "overthrusting" rates in simpler thermokine-matic models used to predict thermochronology ages (e.g. Brewer and Burbank, 2006; Whipp et al., 2009; Robert et al., 2009; 2011; Herman et al., 2010; Coutand et al., 2014, and others). Any increase in shortening rates up to the total India- Tibet convergence rate of ∼20 mm/y could potentially be explained by temporally variable partitioning between "overthrusting" and "underthrusting"; since these concepts are really defined by a particular frame of reference only (which is in my view controlled by the erosional efficiency in the Himalaya), that could be plausible and possibly linked to temporal variations in erosional efficiency. If one wants to invoke further increases up to the India-Asia convergence rate, that would only be possible by temporally transferring far-field deformation to the Himalaya, but it remains in the realm of possibilities. The inferred rates of ∼70 mm/y during building of the Upper Lesser Himalayan duplex are more problematic, because – if true – they would necessarily imply north-south extension in other parts of the Himalaya-Tibet system, for which there is very little evidence. The inferred reconstruction requires significant amounts of shortening to build this duplex (at least 150 km or ∼1/3 of the total shortening since 20 Ma according to Fig. 3)

and I wonder whether a more conservative structural solution would not be possible to fit the surface observations for this duplex. In any case, the preferred models with variable shortening velocities pose significant questions, which should be addressed more directly. The reader is really left wondering how well resolved these shortening histories are, given the significant number of unconstrained parameters in the models. Some of the specific comments below refer to these unknowns.

[Reply] We agree that the fast rate from $\sim$ 13- 8 Ma are unexpected, and yet this is a robust part of the model and is a function of the suite of ZHe ages that are all 8.5- 10 Ma in the Kuru Chu area and 9.5 to 11 Ma in the Trashigang area. These rocks cool through the ZHe closure temperature as the Baxa duplex forms, and accommodates 155-165 km of shortening. 160 km in 2 Myr is 80 mm/yr. That is essentially the problem. We appreciate the suggestion for a more conservative structural solution to reduce the shortening expressed by the Baxa Duplex. However, this is a region where the shortening amount is remarkably well constrained. Shortening magnitude in its simplest sense identifies an area (a box), and calculates the length of a unit with thickness X necessary to fill that box. The Kuru Chu and Trashigang sections from Long et al. (2011b) show how well-constrained this box is. Unlike sections in Nepal, the Baxa duplex in this area is almost entirely exposed and fault bedding plane relationships show the hanging wall cut-offs for the Baxa faults have (almost all) been eroded (implying more shortening possible). Yet, there are erosional remnants of the Paleoproterozoic Shumar/ Daling rocks carried by the Shumar Thrust exposed in fault klippe almost all of the way to the MBT (Long et al., 2011). These fault klippe define the top of the box as being essentially immediately above the erosion surface. There is just enough space to erode the hanging-wall cut offs of the Baxa faults. The base of the box is defined by the décollement. The décollement depth for the Long et al., (2011) cross sections in the region of the Baxa duplex is directly between the 2 permissible depths estimated by Coutand et al., (2014) and matches geophysical constraints in the region (Mitra et al., 2005; Singer et al., 2017). If anything, estimates of the décollment depth are deeper (Coutand et al., 2014) which would just exacerbate the problem. The only variable left

is the thickness of the Baxa, which can be observed in the field, as can the faults that repeat it. Field observations provide several thickness estimates that all fall between 2.1 and 2.5 km and well constrained shortening estimates of 150-165 km. Thus in both the Ehlers and McQuarrie (2015) and in this manuscript, we have tried to figure out what is an acceptable age range that does not violate the data. Shortening rates can viably increase up to the India-Asia convergence rate of 40-45 mm/yr. The expectation is that during that window of time the Himalayas are taking up the entire magnitude of convergence. Shortening rates above that (45-70 mm/yr do require coeval extension to be viable. We have conducted more simulations looking at the sensitivity of shortening rates, particularly using the new geometry. Due to limited measured cooling ages between 70 -100 km from the MFT there is more flexibility in the Trashigang section than the Kuru Chu and rates as low as 45 mm/yr (at plate tectonic rates) are permissible. Our new thoughts are that a revised geometry for the Kuru Chu section (two ramps) may facilitate more exhumation in this region and thus lessen the need for excessively fast rates (55-75 mm/yr) for that section. Intriguing enough (because I (McQuarrie) have never been a huge fan of extrusion or channel flow) the age of this rapid shortening in the Baxa duplex overlaps with the age of the STD in this portion of the Himalaya –12.5 Ma (Th-Pb monazite age from Kula Kangri at the border of Bhutan and Tibet) and 7 Ma (ZHe ages)( Edwards and Harrison, 1997; Coutand et al., 2014). Although shortening rates should not be faster than plate convergence rates, it is permissible if it is accompanied by fault parallel normal faulting, such as is postulated by channel flow models. To me (McQuarrie), one of the strongest arguments for channel flow/ extrusion like behavior is thrust faulting rates above plate tectonic rates. The observed thrusting rate would be the shortening rate plus the extension (extrusion) rate. Numerous changes were made to the manuscript in section 5.3 to address this comment and our new simulations. (1) Without relating all of the justifications for the cross-section, we have included the statement that the cross section itself is a minimum shortening estimate and that any change to the cross-section will increase the shortening. We referred again to the Long et al. (2011b) paper where the details are laid out. (p. 22

~l. 30) (2) The manuscript includes revisions to discussion evaluating the permissible ranges of deformation ages and rates based on our simulations (~p. 23 l. 5-8) (3) To present these new simulations in the paper, a new figure 11 has been created and introduced in this section, and table 3 updated. (4) Comparison of the Trashigang section to the Kuru Chu section, thermochrometer data available along the sections, and reasons for differences between rates proposed by this study and by McQuarrie and Ehlers (2015) are included. (p. 23 ~l. 19-35) (5) Because so many of the permissible shortening rates are above plate tectonic rates we have also expanded on our discussion of modeled rates to include their relationship to convergence rates in section 5.3. (p. 24 , l. 15-25)

Overall, the paper is fairly well written and illustrated. On a number of occasions, phrases don't run because a verb is missing or because of singular/plural confusions. A certain number of typos also remain. All of these can be weeded out by some careful editing. The use of some internal "modelling jargon" like "Python topography", "Split KT" etc. does not add to the general understanding of the manuscript – the authors might want to find some more eloquent terms to describe these modelling settings.

[Reply] The manuscript has been edited to correct typos and clarify wording in areas that are currently mistyped or confusing. We changed the topographic estimations from Python Topography and Template Topography to Responsive Topography and Static Topography respectively. Since "Split KT" refers to Kakhtang Thrust motion at two different periods of time (versus all early or all late), we could not find a word that was more descriptive or more accurate and that would improve the readability of the paper. If you have a suggestion, we would be more than willing to incorporate it.

Specific comments, tied to page and line number: p. 1 l. 7-10: the first two phrases of the abstract do not really set up the problem in a very clear manner or "draw" the reader into the problem – you may want to consider rewriting these into something more clear and specific.

[Reply] Abstract was revised. Comments about the first two sentences of the abstract were raised by multiple referees.

p. 2 l. 13-20: this first paragraph of the "Geologic background" section looks a bit lost on its own; it is not very informative (why is the onset of motion on the MCT important here?) and could easily be combined with the following "Tectonostratigraphy" section. The Daniel et al. (2003) and Tobgay et al. (2012) references are missing in the reference list.

[Reply] The geologic background was removed and the critical information was included in section 2.1 on tectonostratigraphy. Daniel et al. (2003) and Tobgay et al. (2012) was added to the reference list.

p. 3 l. 20-21: how were the data exactly projected into the cross-section? This is a critical step, as the ages (in particular for the low-temperature systems) will be influenced by the local topography. See further comments below. [Reply] In order to maintain structural context along the cross section, all of the data (including data from Coutand et al., 2014) were projected onto the cross-section along-structure (i.e. in the direction of the trend of fault while maintaining distance from structures as possible). The exceptions to this in the original manuscript were minor and have been corrected. We have corrected all figures data projected along the section to be consistent with the along-structure projection method, and text in section 2 describes this projection. Since most samples were not taken exactly along the line of section, the elevations of most samples vary from the elevations at these projected location. However, our models do not use present-day elevation in the models either (discussed below in response to RC1 comment on p. 8 l. 15-19). We have plotted all of the data with respect to elevation and limited age elevation trends emerge strongly suggesting the ages are controlled by structural uplift and minimally modified by topography – this is clarified in the manuscript in section 2.2.

p. 3 l. 30: why do the ZHe ages require "rapid" cooling? This inference can only

be drawn by comparing them to other thermochronometer data, or by assessing age-elevation profiles for instance.

[Reply] There is no a priori reason to indicate rapid due to the age and the adjective has been removed.

p. 3 l. 32: three ZHe cooling ages north of the MCT are shown on the cross-section (but only two on the map?). Also, the cross-section of Fig. 2 gives the impression that the samples between ~57-65 km are from the lower Greater Himalayan sequence, while the map shows they are from the upper. Maybe you should sketch in some of the geology above the topography to make this clearer. This also brings us back to the question above of how these data were projected into the cross section. What was their imposed elevation? Simply plotting them on the topography in the cross-section puts them on a much lower structural level than where they actually are!

[Reply] As explained above, this was a plotting error and has been corrected in figures and in text where fit has changed because of the re-projection. Overall results are not impacted by this revision. Since the modeled ages are all predicted at the surface, projecting the samples in the air would have limited applicability to match modeled results. Where discrepancies between modeled and measured ages exist we do examine both the structural and topographic elevations that the samples are from. For example see new text in section 5.3, p. 23 l. 5-10

p. 4 l. 5-9: why do you take this approach? It is easy enough to model the individual data using the combined Move/Pecube approach . . .

[Reply] At the scale we are evaluating predicted versus measured ages and what is controlling the change in ages, these samples plot basically on top of each other, particularly when projected into the cross section. In the version of Pecube we use, the ages plot as the age trend shown on figures 5, 6, etc. In our view, they represent a true variability in sample age and can be considered a clustered datum rather than several data for our purposes. The one minor caveat to this is the cluster of AFT age in structurally higher Greater Himalayan rocks. As expanded on in our responses to reviewer 3, (and included in the text at the end of section 2) there is a modest age elevation trend here. However the exhumation rate given by the age-elevation differences is 0.4 mm/yr while an average 3.5 Ma AFT age suggests more of a 1-1.7 mm/yr exhumation rate. Additional details of possible age elevation relationship are mentioned at the end of section 2.2.

p. 4 l. 17-18: the question here is obviously: "how was the new topography obtained?" this is discussed further on – you may want to refer the readers to this later discussion here.

[Reply] We mention where the approach is discussed further in the first paragraph of section 3.1.

p. 4 l. 26-27: Note that a subsequent similar model by the same authors (Hammer et al., GRL 2013) comes up with much lower estimates for the elastic thickness in Bhutan (< 25 km) than in Nepal. [Reply] Yes, the very low values (in Hammer et al., and in Berthet et al., 2013) are in part a function of their approach for estimating EET that varies spatially (something that we are unable to mimic using the flexural algorithms in Move). In addition, the solution is for modern EET, which for Bhutan is strongly depending on the narrow width between the MFT and Shillong Plateau. 1) Our EET is a much longer-term average and, 2) is not meant to be viewed as a calculation of the EET in the area. However we can state with confidence that using low (25-40 km) EET values in the flexural-kinematic model will not reproduce the foreland basin thickness, the modern dip of the décollement or the geology exposed at the surface today. This section has been modified appropriately.

p. 5 l. 2: here you could reference some of the previous studies using the same approach.

[Reply] Although there has been a suite of groups moving forward with linking cross-sections to advection diffusion models the details of the kinematic model are not always

clear particularly if or how flexural loading and erosional unloading were accounted for. A good example of the potential influence is ErdÅŚs et al. (2014). They noted that a cooler crustal thermal structure was needed to match the measured high-temperature cooling data (than the lower temperature data) in the Pyrenees. Alternatively, their model could be restoring the rocks to a position that is too deep (thus becoming too warm) because thrust-related isostasy was not taken into account, or perhaps accurately accounted for as the section was retro-deformed backwards in time. What this paper highlights is that accounting for flexure (and erosion) in the kinematic model is a critical and necessary component. We added text addressing this in section 3.1 as well as 3.1.1. In both sections, references to work using this approach were added. We added more detail in section 3.1 to discuss the kinematic modeling process, in particular how different groups account for flexure, erosion, and thus paleodepths, because these decisions are going to control the estimated temperature histories and ages

p. 5 L. 27-31: a self-consistent approach would be to use a critical-taper topography in the models – it is not clear if the "Python topography" is based on such an approach, but the link between the imposed topography and a critically tapered wedge model could be outlined here.

[Reply] The "Python Topography" (now Responsive Topography) may be viewed as a simplified critical taper approach, with the first order angle of topography estimated from modern topographic angles in the Himalayas. A key difference is that we do not systematically vary the topography angle based on the décollement angle. Please see further discussion response to p. 8 l. 22-24 comment below.

p. 7 l. 6-17: see general comment on variable shortening rates above. More justification and discussion of these rates is needed.

[Reply] As mentioned in the general comment above, a whole range of velocities were tested and we acknowledge that a full suite of parameters tested (including velocities) was not reflected in the previous version of this manuscript. We have addressed this in

section 3.2.2 and Table 3. This is also more fully addressed in the discussion section 5.3.

p. 7 l. 16: it seems that this is the first time the Kuru Chu section is mentioned; it hasn't been introduced previously (but should be).

[Reply] The Kuru Chu section and corresponding studies are now mentioned in section 2.2, (multiple locations), and earlier in section 3.2.2, and quite a lot in section 5.3.

p. 7 l. 19 (and numerous other occurrences): why do you call the reconstructions "flexural models"? This is surprising and confusing, as flexure is only one component of these models; the structural reconstruction is at the heart of them. You could call them "kinematic models" or something like that. [Reply] The decision to call the models "flexural models" stems from the multi-step process of achieving a viable "kinematic model" in Move – and from our suspicion that the flexural component is missing from most thermo-kinematic modeling approaches that use cross section kinematics (clarified in the revised end of section 3.1). Without accounting for flexure in the kinematic solution, the evolution of the décollement cannot be determined and thus the estimated depth history (and resulting thermal history) of a given rock becomes a complete guess. Thus a forward model taking into account flexure is critical. We are weighting the flexural component with the term 'flexural'. The work flow for any given kinematic model is to first find a pure kinematic solution (the "kinematic model") with only fault motion accounted for, the second round of iterations is the flexural component that requires an evolution of topography, erosion, foreland basin development, and décollement flexure. We have revised the name to include both adjectives, Flexural-kinematic model, to make the model name more intuitively descriptive. We also clarify the reasoning for this in the revised end of section 3.1.

p. 7 l. 30: the INDEPTH lines were shot in the Yadong rift, which overlies the Yadong cross-structure – a probably important lateral ramp in the Main Himalayan décollement. Is the 4° dip you cite here relevant for the décollement west or east of the Yadong

structure? In any case, this would be valid for western Bhutan and not necessarily for eastern Bhutan. It is not obvious that comparing the décollement dips with data that are not from the same region is very informative, given the probable lateral segmentation of the MHT.

[Reply] We have removed this reference and added Singer et al. (2017), which has estimates for both the décollement and Moho for this region of eastern Bhutan.

p. 8 l. 1-4: this is counter-intuitive. The flexural response should be driven by the topographic loading, not by the kinematic scenario. Therefore, if the different kinematic models lead to differences in flexural loading profiles, it must be because the (imposed) topographic response to the kinematics is different between these models.

[Reply] Yes, this is correct. We have rephrased this to make it much more clear and more accurate. See revised section 4.1

p. 8 l. 15-19: why do you not simply use the present-day topography as the final topography in the model? This is a known entity, and at least that would help in comparing kinematic and thermal histories at the right structural and topographic levels for the data points.

[Reply] While at first impression it seems that using present-day topography as the final topography would improve the integrity of the models, that is only true of a model that can 'predict' a topographic evolution where the next to final topography is almost identical to the modern topography. If there is significant discrepancy between the penultimate predicted topography and the present-day topography (if inputted as the final step) the result would be unrealistic "deposition" of material in areas that are modeled in the prior step with a lower topographic elevations than actual topography. Simultaneously, in areas that have lower actual topography than modeled, using present-day topography could simulate several km of unexplainable erosion. We recognize that topography of the Earth's surface is altered by more processes than are accounted for in our simplified, first-order estimation of topography such as river incision, the geometry

of interfluves, and the effect of axial or transverse drainages. Our approach to modeling topography is outlined in McQuarrie and Ehlers (2017): "the more simplified critical taper model that responds to regions of uplift or subsidence will account for the longest-wavelength, and most significant, topographic effect (i.e., valley and ridge topography) in the thermal calculation." Each kinematic scenario prescribes a different evolution of topography because as Reviewer 1 stated in the p.8, l. 1-4 comment, "topographic response to the kinematics is different between these models." Our goal is to determine if the estimation of modern topography using the python script can successfully replicate the first-order patterns of present-day topography. This is why we compare where and how the modeled topography deviates from the actual topography.

p. 8 l. 22-24: this phrase is hard to read and also appears counter-intuitive. In the critical-wedge model, the surface topography ($\alpha$) and décollement dip ($\beta$) are linked through the critical taper angle (which itself depends, among other things, on $\beta$). Therefore, it might be more self-consistent to try to find a surface topography angle that corresponds to the critical taper for each time step (and degree of topographic loading). This would be an iterative approach, but I'm sure it can be done. See comment on p. 5 l. 27-31 above.

[Reply] This is an intriguing point and one that we have thought about. As elaborated on in our reply to Reviewer 3, Move is a purely kinematic model and thus not governed by mechanical responses. Critical taper is a mechanical response that is dependent on a ratio of internal rock strength to décollement strength (i.e. resistance to sliding) (Dahlen 1990; Suppe, 2007). Thus assuming constant critical angle (one in which the topography angle becomes smaller over time as the décollement angle becomes steeper) would most likely misrepresent the topography evolution of the fold-thrust belt because décollement strength changes as lithologies change. As pointed out by Stockmal et al. (2007), pure critical wedge solutions become more limited when evaluating the effect of material differences, particularly ones with original horizontal geometries, and the ways in which those initial planes of weakness impact the internal structural

geometry, strain history patterns, etc. This non-uniform behavior alters the predicted erosional response. An example may be the front of the fold-thrust belt dramatically propagating forward (on a weak décollement) before the development of a duplex system. The jumping forward would dramatically reduce the taper angle and the duplex response would be to increase structural and topographic elevation to regain "critical" taper (so the system can move forward). Using a constant (say 2° topography angle) in a model suggests that the taper angle is increasing through time. A true self-similar response would argue that the initial topography angle of the cross sections presented here would be 2.5°- 3.5° with an initial décollement angle of 1.5° to produce a final critical taper of 6°- 7° (broadly similar to the modern 4-5° décollement and a 2° topographic slope). What we do know is the geology that is at the surface today, the modern dip of the décollement, and the cooling ages of a suite of minerals. What we can test is a topographic evolution that best matches all of those constraints because the ability of the model to predict older and deeper thermochronometer ages reflects its ability to accurately estimate the relationship of those rocks to the evolving surface of the earth. Critical taper theory gives us broad bounds for what may be a realistic topographic evolution though time. And that is an evolution that can get tested (using a range of permissible topographic angles) to see how accurately it reproduces the first order features in the modern topography. Regardless, a taper angle is topography plus décollement, and defines an area that is filled with folded and faulted rocks. If the area does not change, (because the taper angle des not change, then a lower topographic angle would require a steeper décollement. We have rephrased this in section 4.1 to make this clearer.

p. 9 l. 11: you may have modified your version of Pecube, but in the "standard" model, heat production is constant with depth, so that "surface heat production" is a bit of a confusing term in this context.

[Reply] Following the approach and rationale summarized in McQuarrie and Ehlers (2017), we prescribe an exponential decrease in heat production with depth, as opposed to assuming a constant crustal heat production. An exponential decrease in heat production with depth requires definition of a surface heat production (Ao) and an e-folding depth. One caveat of this approach is that material properties are not exhumed during the simulations to modify the surface heat production value. However, an exponential decrease in heat production with depth has the advantage of honoring observations that heat production diminishes with depth through the crust and that this decline is not monotonic (Chapman, 1986; Ketcham, 1996; Brady et al. 2006). This approach not only allows honoring measured surface values of heat production in the Himalaya (e.g. see Whipp et al. 2007), but also produces reasonable mid and lower crustal temperatures that would not produce partial melts. This text has been added to section 3.2.1.

p. 9 l. 15: this seems a fairly obvious result, since the kinematics of the models do not change, only the thermal field. The samples have the same "normalized" thermal histories; the temperatures are simply somewhat higher throughout for the models with higher heat production.

[Reply] Yes, we agree. We have added this phrase when we first talk about the differences in the predicted ages. i.e. "The most apparent trend among all three thermochronometer systems is that predicted cooling ages become younger as the radiogenic heat production increases from 1.0 to 3.0 $\mu$W/m3 due to the higher temperatures throughout the model." In addition we now talk about how changing values of heat production effects the three thermochronometer systems differently. Specifying the changes in predicted cooling ages as Ao values change is necessary to fully address the concern raised in p. 13 l. 31-32, when we altered both heat production AND geometry , Reviewer 1 was left wondering "OK, but how much of this improved fit can be ascribed to the new structure and how much to the increased heat production." The background we have expanded upon here is needed to emphasize what signals are a function of changing geometry and what signals are a function of changing heat production when both change later in the manuscript (sections 4.3.1, 5.1.2 and 5.2).

p. 9 l. 19: "ages" not "rocks", I think.

Corrected

p. 10 l. 3-5: a bit of a rambling phrase that is difficult to read/understand. We revised this paragraph to make it easier to read.

p. 10 l. 24: "later" not "earlier" I think?

Revised to "more recent"

p. 10 l. 32-33: there are many free parameters in these models: not only an infinite number of shortening-rate histories, but also significant degrees of freedom in the imposed structure and the topographic evolution. I fully understand and appreciate the difficulties in exploring this complex parameter space, but how robust are the inferred rates really? This is not obvious, and given the important implications of the shortening-rate history, this should be discussed. An alternative approach would be to not allow shortening rates that are greater than the plate-scale convergence rates at any time (i.e. use the plate-convergence rates as a constraint) and try to find models that can explain the data using this constraint.

[Reply] We agree that the sensitivity of the model to the prescribed rates needs to be more fully discussed. The questions that Reviewer 1 raises on how-well constrained shortening magnitudes are, helps to elucidate what additional information is needed. To address this comment, we removed much of the last paragraph in section 4.2.2 that emphasized the variations in shortening rates. Instead we ended with the very important observation that even with dramatic changes in shortening rate, the model still can not accurately predict cooling ages through the greater Himalayan section. We return to the discussion of shortening rates and the sensitivity of the predicted ages to these rates in section 5.3. We discuss the sensitivity of the model to rates that are at plate convergence rates ($\sim$45 mm/yr) versus faster than plate convergence rates when we present the revised geometry. In the end, there is limited usefulness in evaluating

rates with a geometry that will never reproduce the measured ages.

p. 11 l. 10-11: why is this your expectation? The erosional history would depend on the topographic history through time, rather than the final topography. In the no-topography scenario, if I understand well, there is no topographic change through time. If in the other topographic scenarios topography diminishes locally in the final timesteps, this will predict younger ages.

[Reply] Yes, a topographic scenario where topography diminished with time would produce younger ages, and the expected exhumation difference would be approximately the change in topographic elevation (maximum 2-3 km). Our expectation that the No Topography scenario would produce younger ages is because these models always produced higher total exhumation where the final cross section was over eroded by 1-2.3 km. The age in which this exhumation happens is a function of the age that a given structural relief was being generated. As an example, some component of over-erosion happened as the upper Lesser Himalayan duplex moved up and over the pronounced ramp at 65 km. Thus our expectation is that predicted AFT ages that show this exhumation would be younger. The conclusion is that since the magnitude of erosion that happens during this displacement in each topographic scenario is significant, the additional 1-1.5 km of extra erosion in the No Topography scenario is not significant – particularly when viewed incrementally (e.g. Valla et al., 2010).

p. 11 l. 14-15: a list of 6 adjectives ("Python topography model fully reset Mar ages") followed by another of 4 . . . Maybe rewrite?

This was revised.

p. 11 l. 20-23: this is an important point but it also seems fairly obvious. It clearly points to the need of a self-consistent treatment of topographic evolution. The best way forward may be to combine these models with simple surface-process models to erode the topography through time.

[Reply] We agree, a self-consistent treatment of topographic evolution where the modeled topography is a function of the deformation is a key result from this work. Although this seems like an obvious result, it is also a common approach to use a DEM of modern topography in models and assume that topography is in steady state and not changing – this result highlights that assumption is not valid either (and may also cause burial of material where particle points are subsiding and topography is not, and produces over-erosion of material where rock uplift occurs but topography remains static. Also, while it is obvious to Reviewer 1, how topography is estimated particularly over long time windows is still a rather new item of discussion and application for thermokinematic modeling in compressional orogens. As outlined in the introduction, several other studies that have used Pecube have not used a method of applying topographic evolution that account for localized structural uplift and isostatic subsidence. Rather, they apply a muted topography similar to present-day elevations, infer topographic changes that seem appropriate or increase/decrease topographic slope over time. Yes, a self-consistent way to estimate topography is critical. While we see the value of using other surface-process models such as Cascade to erode topography over time, the Python code (or an equivalent Matlab code) we use in this study, which approximates the first order topographic slope and specifically accounts for increasing topography in regions of active uplift and subsiding topography, provides a critical first step for estimating topographic change particularly in the isostasy calculations in Move. What may not have come through in the paper was the iterative process of finding a flexural solution (which is why we referred to it as flexural modeling). The kinematic displacements are known, and we are searching for a solution where the sequential kinematic restoration in Move (using flexure) can reproduce the depth of the foreland basin, geology at the surface and the dip of the décollement. This may take 20+ iterations to achieve using 20 km shortening increments. Thus whatever mechanism is being used to generate an initial topographic estimate needs to evaluate the magnitude of topography change and predict a new topography in <1 minute to be viable in the iterative process. In addition, 1D erosion models require an estimate of time

(which would have to be approximated for the initial reconstructions). Of course if the velocity were to change then the flexural-kinematic reconstruction in move would need to be redone. 1D erosion models also do not account for sedimentation (in a growing foreland basin). Our thought process is that the thrust loading and erosional unloading are much more sensitive to the first-order component of topography and thus using the responsive topographic taper approach is the best approach for Move. Once the displacement field has been determined (and then the resulting velocity fields), Pecube can run in conjunction with Cascade, to predict a more realistic and variable topography. As a double check –this Cascade Topography can be imported again in Move – just to make sure the resulting isostatic load is the same. We are currently working on fully integrating our modified version of Pecube and Cascade.

p. 12 l. 2: I think you are discussing MAr ages specifically here? May be useful to state this.

Revised

p. 12 l. 30: "older ages" seems more correct than "earlier ages" in this context.

Corrected

p. 12 l. 31: you have been calling this the MHT throughout the manuscript. Better stick to this acronym so as not to confuse the readers.

Changed

p. 13 l. 1-2: another somewhat rambling phrase . . .

This has been revised

p. 13 l. 5: this ramp is rather located at _90 km in the present-day geometry (Fig. 2)?

[Reply] No, this early ramp is no longer visible in the cross section. See figure 3 C.2a for ramp location. We have clarified this in the Manuscript by revising the first 2 paragraphs of section 4.3 and referring to the appropriate figure location and ramp locations in the

text.

p. 13 l. 12-15: is the cross-section of Fig. 9 still balanced? There is all of a sudden 35 km more Baxa group in this cross-section, while the rest of it has not been modified. Could these additional 35 km be found by reducing shortening in the upper LHS duplex? In that manner you might also be able to reduce the problematic shortening rates necessary to produce this (and the associated ZHe ages).

[Reply] The new cross-section in figure 9 is balanced. Forward modeling the kinematics of a cross section ensures that it is balanced. But Reviewer 1 is correct in that the distribution of shortening has changed. All ramps north of the new Baxa footwall cutoff were shifted 35 km north, and thus 35 km of shortening was added. Yes, we agree this does not reduce the problem of the fast rates (it can make the rates higher). Our modeling (and others) have highlighted the strong relationship between ramps and young cooling ages. We can use this relationship and what is required by the geology to figure out how far south we can place the southern ramp (through the Diuri) initially this was placed at its location because the pervasive northward dips in structurally higher units (the northward dipping boundary of the Shumar-Daling on Baxa, GH on Shumar-Daling and the northward limb of the STD all suggest a northward dipping ramp $\sim$ in the location shown on both cross section). What we did was turn this large ramp in Long et al.'s (2011b) original section into two ramps to better match the cooling signal. We know that the Baxa formation *has* to be under the anticline of Shumar-Daling because of the along strike relationship shown in the Kuru Chu section of the map (figure 1 – the anticline shown in the cross section is underlain by the Baxa Group rocks repeated by faults). So, even though we could move this ramp farther south to 50 km (location of the youngest AFT age in this region – figure 9), we can't remove either of the Baxa horses. In addition, moving the ramp farther south would make each of these horses longer, adding more shortening back into the geometry. The cross sections were constructed to minimize shortening while matching surface constraints – thus any modification to the cross section that also matches surface constraints will

tend to increase shortening estimates. This last point was added in the discussion section on rates, and we have included the restored modified cross-section below the deformed section in figure 9b.

p. 13 l. 25-29: this is problematic. First of all, you change two major inputs to the model (structural geometry and heat production) at the same time here, while previously you have carefully only changed one parameter at a time. Second, you introduce spatially variable heat production here, which you did not do previously and which could have led to better fits in the previous models. This is a large change in the thermal structure and it should be justified. Although I am sympathetic to the fact that heat production could be significantly higher in the GHS than in the LHS, to really model this properly you should ascribe heat-production values to the different units, and advect these with the units.

[Reply] Yes, we agree that the jump was too large to independently see the effect of both, but our goal was to show the best fit and a reasonable number of models and iterations. Numerous kinematic and thermal model iterations were performed in addition to the specific model results presented in this paper. Most of these iterations were performed with the goal of obtaining an improved fit using the cross-section geometry published by Long et al. (2011b). Several new models (changing flexural and topographic parameters were run using the new geometry to produce several models with slightly different exhumational histories to test the sensitivity of the model results to changing these different parameters. All models run in Pecube were evaluated using heat production values ranging from 2.0 to 5.0. (in steps of 0.5) and a range of different velocity combinations. In all, nearly 100 forward modeling combinations of the Long et al. (2011b) geometry were run for this study, and over 100 for the new geometry. None of the models from the original Long et al., (2011) cross section could reproduce the AFT age trend seen across the GH (younger ages farther north), even with significantly higher heat production values in Pecube. This unsuccessful result of not being able to match the cooling ages with the original section led to the decision

to strategically explore new geometry options, beginning with the replacement of the Baxa footwall cutoff. After evaluating a range of velocities and heat production values, we concluded that it would be best to ascribe different heat production values for different units in the model – even though we agree the most accurate approach would be to characterize each unit with distinct heat-production values in a single model. However we were limited by the current capabilities of our model. Thus the simplest way forward was to combine the results of the two models at the surface location of the MCT. Using Supplementary Figure 2 and 3, one can infer the range of potential cooling ages that would be predicted if it were possible to implement unit-prescribed heat production in Pecube. This seems most important for units in the immediate hanging wall and footwall of the MCT (∼52 km north of MFT) where GH rocks that are known to be hotter with higher radiogenic heat production are spatially juxtaposed with the cooler Daling-Shumar units. This area is also where the greatest amount of cooling data are available. We have worked to make this as transparent as possible in the revised manuscript. These include figure revisions to figure 9 and supplementary figure 2 and 3, and clarifications throughout sections 4 and 5. Examples include (1) the final paragraph of section 4.2.4 which highlights what is controlling predicted AFT ages in the immediate footwall of the KT, (2) the fourth paragraph of 4.3 detailing the rationale and method for combining models with different Ao values, and (3) section 5.2 which re-emphasizes the point that, even with higher heat production, fit of AFT ages remain poor in the immediate footwall of the KT using the original geometry proposed by Long et al. (2011b).

p. 13 l. 31-32: OK, but we are left wondering how much of this improved fit can be ascribed to the new structure and how much to the increased heat production.

[Reply] Supplementary figure 2 graphically presents the best results from the using the updated cross-section geometry with 2.0 and 4.0 $\mu$W/m3 heat production values applied along the entire line of section. New supplementary figure 3 does the same with the original geometry. As mentioned in the response to the comment directly

before this one, we have clarified this in the text.

p. 14 l. 5-6: following up on the previous comment; can the data really tell the difference between the improved structural geometry and the increased heat production? There is very little data in the "bump" region. You use a simple visual comparison of predicted and observed ages; it would be useful to provide a more objective and quantitative comparison to back up inferences such as this.

[Reply] We have added discussion in the manuscript that quantitatively compares the predicted AFT ages from the Long et al. (2011b) geometry to the predicted AFT ages from new geometry presented in this paper in order to support our conclusions of improved fit.

p. 16 l. 10-12: this is introducing yet another unconstrained parameter. I am not sure it is the best strategy to further complexify the models to improve the fit; this seems like a bit of a "flight forward". A more complete sensitivity and resolution analysis might be a more productive way forward.

[Reply] There are no new parameters. The parameters being discussed are EET and topography (section 3.1.1 and 3.1.3), and any given solution presented in this manuscript is a function of both parameters that combine to affect the exhumation of rocks. Is the added complexity you mention changing the value of EET or topography with time? There are strong arguments that can be made that both may have changed with time- and reflects your point made previously (p. 11, l 10-11). A forward model where multiple parameters have to be evaluated, and it is impossible to see if the model is a match to present day conditions until the last step, will always be a "flight (fight) forward". Not all questions or problems can be addressed through inverse solutions. The reality (which is why this section is important) is that subtle changes in EET have a larger effect on the modeled cooling ages than subtle changes in topography (such as using a process based estimation of topography or a simplified critical taper relationship). The reason why, is that a 5 to 10 km change in EET can impart a

1-3 km difference in magnitude of exhumation. Unfortunately, the flexural response to fault motion and associated topographic displacement (solved in the kinematic model) is something that is not included in many models attempting to link cross section to thermokinematic models, yet it has a significantly larger control on the predicted cooling ages than topographic estimations. We have clarified this section to emphasize this point.

p. 17 l. 9-10: "the amount of exhumation in this model is just at the amount necessary to reset AFT ages" is strange and apparently incorrect. The ages record cooling through the closure temperature at a certain time in the past. The thermal structure is going to affect that time, but the total amount of exhumation is much larger than the AFT closure depth it would seem.

We have rewritten and clarified this point in section 5.1.2

p. 18 l. 10-15: A bunch of hard-to-read phrases that are in need of a few commas. Also, "after 13 Ma" would be better than "longer than" and replace the colloquial "till" by "until".

We have edited this text for clarity and grammar.

p. 18 l. 15-20: another potential issue that is not discussed concerns the diffusion kinetics of He in zircon. Recent work has shown that the effective closure temperature of the ZHe system can vary from as low as _120 _C to as high as _240 _C as a complex function of the degree of _-damage (e.g. Guenthner et al., 2013). If you have underestimated the ZHe closure temperature (I suppose you are using the "standard" ZHe diffusion parameters built into Pecube) you could significantly underestimate the duration of shortening on the upper LH duplex, and thereby overestimate the shortening rates.

[Reply] The reviewer raises a very good point, and we have modified the manuscript to state this as a potential caveat, although we do not think this is important for our

samples because of the high cooling rate. The text now added in Section 2.2 is as follows: The predicted ZHe ages in this study do not account for the effects of radiation damage on the closure temperature (e.g. Guenthner et al., 2013). The potential effect of this could be to underestimate the ZHe closure temperature. However, the effects of radiation damage on ZHe (or AHe) closure temperatures are most pronounced for long durations at relatively low ($\sim$220°C) temperatures (Guenthner et al., 2013). The Lesser Himalayan samples evaluated here experienced temperatures greater than 350° (Long et al., 2011c, Long et al., 2012), have young ages (typically $\sim$7-11 Ma), highly reproducible ages (for individual samples) and underwent extremely rapid cooling (e.g., or around 16.3-22.5 C /Myr cooling rate since closure at $\sim$180 C), thereby leading us to infer that radiation damage effects are minimal.

p. 18 l. 25-28: the first part of this argument is somewhat circular, since the McQuarrie and Ehlers (2015) scenario was input in the models here, without extensively testing all other potential scenarios. So the fact that the model predicts these variations in rates should not come as a surprise. In contrast, the dissimilar timing between the two sections that are only _25 km apart should be worrying. How can the same structure be active at time intervals that are several million years different between two adjacent locations? Again, the reader is left wondering how much of this difference could be due to variable diffusion kinetics?

[Reply] We agree that many more rates need to be evaluated and presented, and we have clarified that in the updated version of the manuscript (see section 5.3, figure 11 and Table 3). We do not think that variable diffusion kinetics play a significant role (see response to previous comment) but elevation differences might. In addition, a revised geometry for the Kuru Chu section (two ramp scenario) may allow for an older age of transition from lower to upper LH duplexing which would decrease the fast rates.

p. 19 l. 2: given the numerous unexplored degrees of freedom in the models, it appears risky to assess the validity of the data based on the modelling outcomes.

[Figure]

That was not quite our point—thus we have revised and removed this sentence.

p. 20 l. 1: not sure what is meant with this phrase; what is "the spatial nature of thermochronometry"?

[Reply] Wording was edited to clarify this point. The second part of the sentence is the important part: "the importance of considering the aerial distribution of cooling ages in the direction of transport and their relationship to the structural evolution of a landscape."

Figures Fig. 1: the inset geological map of Bhutan (panel B) is very small and not very readable. You should either increase its size or decrease the amount of detail on it. Also, in the legend of the main panel (C), the Chekha Formation should be above the Greater Himalaya to keep all units in their structural order. Finally, it would help the reader if the colours used for the different thermochronometers were consistent between this figure and the following.

[Reply] Figure 1 has been revised. The colors of data points on the map are assigned based on the original studies due to overlap in sampling (e.g. ZHe and AFT data collected at same location). Colors used to label ages from thermochronometers at each sampling location do match colors used in subsequent figures.

Figs. 5-10: much more data appears to be plotted in these figures than in Figs. 1 and 2. What do the lighter-coloured data points refer to? For clarity it would be better to take them out. In Fig. 7, why does the "template topography" model not predict AFT ages everywhere?

[Reply] Figures 9, 10, and 11 include data from the Kuru Chu region (50% transparent) as well to help evaluate similarities and differences between the two sections. This has been clarified on the figure captions and expanded on in the text. Published data are presented in Figures 1 and 2. Are plotted on figure 5-8. Template Topography in Figure 7 does predict ages along the cross-section as completely as the other

two models' output shown. In some areas, there is significant overlap with the other modeling results. In the AFT output plot, the Template Topography output lines are discontinuous because predicted ages were more scattered.

Please also note the supplement to this comment:
https://www.solid-earth-discuss.net/se-2017-117/se-2017-117-AC1-supplement.pdf

---

## Author Comment (AC2) · 11 Feb 2018

We appreciate the time and energy that the reviewer put into the evaluation of our manuscript. The comments and questions were insightful and addressing them has improved the quality and the clarity of the presented science. We have arranged our response by 1) reiterating the comments of the reviewers 2) providing our response and clarifying where the comment was addressed in the revised manuscript.

RC2 This manuscript analyzes the impact of variable radiogenic heat production, convergence rate, topographic estimates and out-of-sequence thrusting in determining the pattern of previously published thermochronologic ages along a transect across the

Bhutan Himalaya. The authors utilize their results to validate a revised cross-section geometry of the study region. The manuscript is generally well written. The topic is of potential interest for a broad international audience. However, it would benefit from a more comprehensive discussion of the whole range of geologic processes that may have an impact on the thermochronologic record of the study area. The modelling approach utilized in this work is based on flexural and thermal-kinematic models. The authors sequentially deform the study cross section, and apply flexural loading and erosional unloading at each step to develop a high-resolution evolution of deformation, erosion, and burial over time. In other words, their approach only considers relatively shallow geologic processes. Deeper tectonic processes (e.g., channel flow exhumation and slab breakoff) that may also affect the thermochronologic record, especially higher temperature systems such as Ar-Ar on mica, are not discussed. This may puzzle part of the potential readership. I suggest to improve on the discussion, and possibly the modelling, in order to include these issues.

[reply] A discussion of more ductile processes on the higher temperature thermochronometer systems was raised by Reviewer 2 and Reviewer 3. The flexural and thermokinematic model looks at the evolution of rocks from 30 km depth and $\sim$ 600-700 °C (peak temperature produced in Greater Himalayan rocks in the thermokinematic model, Pecube). As mentioned in the reply to Reviewer 3, the kinematic model will not capture all of the deformation processes, but it can evaluate if the cooling through the closure temperature of the MAr system was simply a function of shallower fold-thrusts belt processes — or if deeper processes (such as channel flow or slab break off) are needed to explain the data. Also, channel flow (if active) is interpreted to be reflected in the much higher temperature monazite data, which is not modeled in this study. What is key to note is that the kinematics described here can reproduce the peak temperatures and cooling history recorded in the rocks. We have made minor revisions in multiple sections of the manuscript to incorporate this discussion raised in RC2 and RC3: 1) 2.1 Tectonostratigraphy states that the Greater Himalaya was deformed through ductile processes, and that MCT shear is pervasive above and below the fault, 2) 3.2 Thermal Model includes clarification on the depth and temperature range of the model as well as how isotherms are advected by motion along faults, 3) The discussion section clarifies permissible processes to reproduce the measured ages (including MAr).

The dataset of previously published thermochronologic ages, which is utilized as a benchmark for modelling, is not homogeneous. AFT and ZHe data are available in most of the transect, but Ar-Ar data are not. This would suggest more caution in the conclusions based on modelling results. Moreover, these ages are invariably interpreted as cooling ages during exhumation across the closure temperature of the Ar-Ar system. Petrologic studies demonstrate that micas in metamorphic rocks often preserve disequilibrium textures, and their Ar-Ar age may thus record fluid-induced recrystallization below the closure temperature, rather than monotonic cooling (e.g., Villa 1998 - Terra Nova). Why mica Ar-Ar ages are so different in samples that are so close each other? What is the potential role of recrystallization during deformation? These issues should be discussed in the revised main text.

[reply] The available MAr data for this transect are very limited and were previously published by Stüwe and Foster (2001). The 40Ar-39Ar age spectra show relatively flat but slightly discordant age spectra that were interpreted to represent cooling ages for all 4 samples. The two sets of 11 Ma and 14 Ma ages were interpreted to record the same cooling signal that had been repeated by a fault. Our interpretation is broader and proposed that the 11-14 Ma ages represents a permissible age range in which rocks have passed through their closure temperature due to the short spatial scales between samples. Recent work from Sikkim Himalaya across the same Lesser Himalaya to Greater Himalaya transition highlights natural variability in MAr ages due to both the thermal conditions experienced by micas and the residence time at those temperatures. They measured both single grain ages (for 5-11 grains) as well as more traditional plateau age (Mottram et al., 2015) across a transect that spanned a temperature gradient over $\sim$ 5 km. They found a significant spread in the single grain ages (2-5 Ma not including errors) and that the spread decreased (to 1.5-2 Ma) with higher

temperatures and longer predicted residence times at those temperatures, suggesting that the duration of metamorphism and the temperatures reached affected the loss of Ar from mica. In each case the MAr plateau ages spanned over a much narrower age range (13- 13.4 M) with significantly more precise error bars (0.05-0.2 Ma) than the single grain ages. The $\sim$ 5 km transect crossed temperatures that ranged from 580°c to 650°C, while the maximum temperature range for the MAr samples presented here were between 600° and 700°C (Daniels et al., 2003). Their study also showed that a dispersion of +/- 2 Ma would be expected due to diffusive differences caused by grain size variations. We do not have access to the samples to go back and examine the textures of the mica that produced the cooling ages. However we have looked at many similar rocks from almost the exact same area and have found no textures indicative of fluid flow or alteration. While this does not rule out an age spread from post-cooling fluid flow or recrystallization during deformation, we are confident that the 11-14 Ma age range encompasses the actual cooling age of these rocks because of strong similarities in age to data available directly to the east near the Kuru Chu section ( $\sim$12 Ma, Long et a., 2012; Figure 9 in this manuscript), as well as the range in ages measured by Mottram et al. (2015 in Sikkim (12-16 Ma). These ages are all younger than the youngest age for south-directed shear in GH rocks, 16-18 Ma (Grujic et al., 2002; Daniel et al., 2003; Kellett et al., 2009). In our model, the age and rate of deformation in the northern duplex of lower Lesser Himalaya most prominently control the predicted MAr ages modeled in this area of the Greater Himalaya. Text was revised to address this point in sections 2.1, 2.2, 3.2, and 5.3. New citations are also included, i.e.: Mottram, C. M., Warren, C. J., Halton, A. M., Kelley, S. P., and Harris, N. B. W.: Argon behaviour in an inverted Barrovian sequence, Sikkim Himalaya: The consequences of temperature and timescale on 40Ar/39Ar mica geochronology, Lithos, 238, 37–51, doi: 10.1016/j.lithos.2015.08.01, 2015.

Some of the findings of the authors are not surprising for an active orogenic belt such as the Himalaya, notably the minor effect of radiogenic heat production and topography compared to tectonics. Nevertheless, the authors' conclusion should be supported by

more robust thermochronologic data. The addition of a new ramp under the Greater Himalaya does better explain available thermochronologic ages. However, this is just one of the possibilities, given the degree of freedom of the models.

[reply] Compared to other regions, even in the Himalaya, the dataset shown in this paper is rich, especially when including the data immediately east along the Kuru Chu transect as shown in Figures 9-11. MAr and AFT data are more limited than ZHe data due to cost and appropriate samples respectively. The reviewer raises an important point and that is, the models highlight regions where the predicted thermochronologic ages are very sensitive to the geometry or radiogenic heat production or velocity. Knowing these areas prior to collecting thermochronology samples would strongly influence where sampling would be the most useful for delineating geometry. Regrettably many of the gaps in the AFT data are a function of the apatite-poor lithology. Resampling and additional analyses are beyond the scope of this paper. However, the model process we present is useful for directing future thermochronologic work in the Himalaya and other mountain ranges. Although many geoscientists model data following the collection of samples, this work suggests that initial thermokinematic modeling of an area prior to collecting data can direct and inform sampling strategies. We are not sure what other possibilities the reviewer envisioned for changes to the cross-section to also explain the published dataset. We chose to highlight an obvious additional structural solution that was proposed to the east in Arunachal Pradesh: an out-of-sequence fault at the trace of the MCT (Adlakha, V. A., Lang, K. A., Patel, R. C., Lal, N., and Huntington, K. W.: Rapid long-term erosion in the rain shadow of the Shillong Plateau, Eastern Himalaya, Tectonophysics, 582, 76–83, doi: 10.1016/j.tecto.2012.09.022, 2013.). As expanded on in section 5.2, Using Thermochronology to Evaluate Structural Geometry, we evaluate whether an out-of-sequence fault can explain all of the observations. While it may be able to address the younger cooling ages, having a second, more southern out-of-sequence fault that post-dates the Kakhtang Thrust would have a pronounced effect on the topography (as highlighted in our response to reviewer 3, specific comment 2), that is not seen in the model topography or geomorphic metrics of active/

recent uplift. In addition, see response to RC1 for further comments on systematic approach to structural and thermal modeling. We have revised this manuscript to clarify these points in sections 5 (Discussion) and 6 (Conclusions).

Is the stratigraphy predicted by modelling consistent with the geologic record? This may provide independent constraints to the reconstructions illustrated in this work, that are prone to remain otherwise speculative. I suggest to describe in more detail the stratigraphic evolution of the foreland basin, as well as all of the other geologic evidence that may be useful to support the authors' conclusions.

[reply] One of the key parameters that we match through this process is the depth of the foreland basin. The modeling process also makes strong predictions regarding the detrital sedimentary signal recorded in the basin and the potential detrital thermochronologic record. Most of this research was accomplished as another research group was examining the details of the detrital climate, provenance, and sediment accumulation signal in the Siwaliks of Bhutan (e.g. Coutand, I., Barrier, L., Govin, G., Grujic, D., Dupont-Nivet, G., Najman, Y., and Hoorn, C.: Late Miocene-Pleistocene evolution of India-Eurasia convergence partitioning between the Bhutan Himalaya and the Shillong plateau: New evidences from foreland basin deposits along the Dungsam Chu section, Eastern Bhutan, Tectonics, 35, 2963–2994, doi:10.1002/2016TC004258, 2016. and, Govin, G., Najman, Y., Copely, A., Millar, I., van der Beek, P., Huyghe, P., Grujic., D., and Davenport, J.: Timing and mechanism of the rise of the Shillong Plateau in the Himalayan foreland, Geology, doi:10.1130/G39864.1, 2018). As with any provenance or stratigraphy study, most information is gained when there is a unique signal that enters the foreland basins, and these papers highlight that much of that signal is associated with the rise of the Shillong Plateau or ages that have a Tibetan origin. The paper by Govin et al. (2018) highlights that at 6.35 Ma, there is significant input of Lower LH detritus into the foreland basin. Our models show both the age (6.35 Ma) and the signal (lower LH detritus), and the depth of the basin at this time (2.75 km), are all consistent. We agree with Reviewer 2 that matching the predicted foreland basin with the

measured foreland basin is a powerful tool for evaluating the flexural-kinematic modeling and rates of deformation. We are currently working on a fully-integrated detrital provenance and thermochronologic cooling set for the Siwalik basin, but a detailed description of the stratigraphic evolution of the foreland with respect to detrital provenance cooling signals and rates is well beyond the scope of this paper to do it properly.

The abstract should be improved. The first two sentences are not relevant to introduce the focus of the manuscript. The Introduction and section 2.1 are biased by excessive self-referencing.

[reply] Abstract issues were raised by multiple referees and have been addressed. Introduction and section 2.1 have been revised to include more references to other research groups as available. In general, 26 new references (not self-citing) have been added to the manuscript.

Please also note the supplement to this comment:
https://www.solid-earth-discuss.net/se-2017-117/se-2017-117-AC2-supplement.pdf

---

## Author Comment (AC3) · 11 Feb 2018

We appreciate the time and energy that the reviewer put into the evaluation of our manuscript. The comments and questions were insightful and addressing them has improved the quality and the clarity of the presented science. We have arranged our response by 1) reiterating the comments of the reviewers 2) providing our response and clarifying where the comment was addressed in the revised manuscript.

RC3 Dear Colleagues In this manuscript the authors present results of sensitivity of predicted thermochronological age distribution on several parameters: prescribed topographic evolution, geometry of the basal detachment and kinematics of the related

fold-and-thrust belt and crustal heat production. The authors conclude that "this study presents a successful approach for using thermochronometer data to test the viability of a proposed cross section geometry based on forward models of the kinematic, exhumational, and thermal history of an area". I fully agree with this statement but have several comments that could help authors improve the manuscript and help reader better evaluate the contributions. I concur with the comments by referee #1 and try not to repeat them here. I apologise for several self-citations, but my research group has been working in the area and applying similar research techniques since couple of decades. General Comments: 1. The general limitation of the kinematic models is that the geometry and kinematics is prescribed – Therefore despite their best efforts dependent on authors' interpretation.

[reply] This is true, the geometry and kinematics are both prescribed, but they are also testable. Following this approach, a cross-section can be invalidated by not matching available cooling data –which is an important step forward. Although this approach seems limiting, it has the potential to refine the geometry of the active décollement in addition or as a compliment to inverse methods. The determination of a décollement through searching a parameter space (see response to general comment 3 below) provides low broad posterior probability density functions (PPDFs) that may have a permissible range in depth of 3-5 km (Coutand et al., 2014). Within that range we can test a specific geometry and require it to match additional known constraints such as the surface geology. See additional comments to RC1 and RC2 on cross-section solutions. Multiple sections of the manuscript reflect the ability to test different geometries and as well as different (albeit prescribed) kinematics using the approach of this study. The revised manuscript retains this emphasis.

1, continued. I agree that this is still the best approach to interpret the spatial pattern of thermochronological data, and couple of authors of this manuscript have made significant progress with their previous publications (McQuarrie and Ehlers, 2015) in reducing these limitations. Unfortunately, the additional problem with the Pecube is

that it cannot generate simultaneous movement on faults with opposite sense of slip. In the Himalaya, and in particular for the GHS, the cooling and exhumation were affected by the simultaneously motion along the MHT at the base and the South Tibetan Detachment (STD) at the top. The STD in the eastern Himalaya was active as a ductile shear zone until 11 Ma, which is half of the period of the here presented experiments. Could the "tectonic denudation" affect the cooling pattern of the northern part of the section?

[reply]There are two important points here. 1) Using the modified version of Pecube as presented in this paper, we actually can generate simultaneous motion on both the MCT and the STD. This can be done in Move by first applying 10 km of motion to the MCT, then 10 km of motion to the STD, and finally accounting for the flexural load and resulting change in topography. The resulting displacement field would show pure extrusion of Greater Himalayan rock in prescribed 10-km increments (or increment value of choice). 2) Although we could, we did not include simultaneous motion of the STD. This choice was made for a variety of reasons; 1) early STD magnitude is largely unconstrained and predates the ages preserved in the thermochronometers systems used in this manuscript, 2) not including STD motion (i.e. potentially more recent activity) allows us to evaluate what component of the low-temperature (ZHe or AFT) exhumation required extensional exhumation from 10-0 Ma. Tectonic denudation could absolutely affect the cooling in the northern part of the cross section. However, based on the match between our best-fitting models and measured cooling ages, we argue that any recent (7-0 Ma) tectonic denudation is minimal. The critical dataset needed in the north would be MAr ages. These data should record the earlier ($\sim$ 11 Ma ??) cooling signal of the STD. We do suggest potential links between the periods of rapid shortening and STD activity in section 5.3 of the resubmitted manuscript.

2. The shape of isotherms and their effect on the cooling rates. Himalaya are an active contractional orogen, therefore, the isotherms are deformed and the geothermal gradient is not constant in space and time. Was this accounted for in the experiments when

calculating the eroded material or when calculating the exhumation rates? For example the same rock uplift rate, minus same surface erosion rate will not yield the same cooling rate. Therefore because the exhumation rates are based on thermochronology, i.e., cooling rates, thermochronological data cannot be simply converted into exhumation rates based on an assumed geothermal gradient. The exhumation rates will depend on local instantaneous geothermal gradient at different times. This is not discussed in the manuscript.

[reply] Geothermal gradients and the resulting shape of isotherms in the model are dynamic and change at each incremental time-step based on 1) thermal parameters prescribed to each model in Pecube; 2) locations and magnitudes of fault displacement; 3) locations and magnitudes of erosion as dictated by structural uplift, isostatic flexure, topographic evolution, and erosion in the flexural-kinematic model; and 4) the rates of deformation and exhumation which are dictated by the absolute timing of each step which we assigned as input in Pecube. We reproduce the same inverted thermal gradients at the MCT (when active) and KT (when active) that have been both proposed and modeled for these structures. Reviewer 3 is correct in that this point should be explicitly stated in the manuscript for clarity. Revisions were made in sections 3.2 and 5.1.

3. The authors write that they have performed a sensitivity analysis. However they have performed a limited number of experiments changing one or two parameters at the time (I concur with the related comments by referee #1). However it would have been better to perform a systematic search through the parameter "space" by providing the ranges of variables and searching for the most optimal value – the lowest misfit. I agree that this is a very time consuming approach, which requires tens of thousands of experiments. However this is the only approach that can provide a statistically relevant evaluation of any of the parameters. Pecube produces posterior probability density functions (PPDFs) for each model parameter, (Braun, J., P. Van Der Beek, P. Valla, X. Robert, F. Herman, C. Glotzbach, V. Pedersen, C. Perry, T. Simon-Labric, and C.

Prigent (2012), Quantifying rates of landscape evolution and tectonic processes by thermochronology and numerical modeling of crustal heat transport using PECUBE, Tectonophysics, 524-525, 1–28, doi:10.1016/j.tecto.2011.12.035.âËŸA'l) I admit that I do not know if this can be implemented by the technique presented here (combination of Pecube thermokinematic modeling and Move kinematic modeling).

[reply] The variables that are assigned in Pecube (in particular heat production properties) can be determined by a systematic search through parameter space. However, the much more interesting and debated properties such as cross section geometry, kinematics, velocity, and topography are all a function of the flexural-kinematic model generated in Move. For these models, designing a parameter search or graphically representing a parameter search is much more complicated. For example, supplementary data Figure 1 shows 9 different models where topography, kinematics, or geometry were varied. These 9 models all produced a foreland basin, dip of the decollement and surface geology that were all considered acceptable (within 1 km of modern thickness; +1/ -0.5° of modern dip; and 1 km of modern surface geology). Over 50 other flexural models were tested that did not match these criteria. Of the 9 models that are presented in this study, each model was run using 4-7 different velocities to 1) see predominant trends on the predicted cooling ages and 2) determine which combination of velocities resulted in predicted cooling ages that best matched the measured data. For all of the different velocities and the different kinematics and geometries we examined a range of thermal properties, specifically Ao (surface radiogenic heat production – see response to comments from supplementary document (annotated manuscript) p. 9, l. 8 below), which has a large, known range of measured values (i.e. Ray and Rao, 2000; Menon et al., 2003; England et al., 1992; Whipp et al., 2007; Herman et al., 2010 – all references in manuscript). Ao was varied in 0.25 to 0.5 $\mu$W/m3 increments. Note that we do not test the effect of basal heat production, but rather hold that fixed at 1300°C at the asthenosphere $\sim$110 km (Table 2). While the number of variations we tested is not an infinite number (or 10's of thousands) it is respectably above 500 simulations (in Pecube). The challenge is of course visually showing that range. We understand,

based on comments by reviewer 1 and reviewer 3, that the full range of parameters tested was not clear and we have rectified this in the new version of the manuscript, particularly in section 3.

4. The GHC is not a thrust sheet-the rocks in this lithotectonic units were affected by pervasive and heterogeneous ductile deformation. Similarly the MCT is not a fault but a several kilometers thick ductile shear zone with mylonites derived both from footwall block rocks and the hanging wall block rocks. All these rocks deformed as viscoelasto-plastic thermally activated materials and ought to be modeled as such not as Mohr-Coulomb materials. I do not question the applicability of cross section balancing and thermokinematic modeling for the rocks and structures that were dominantly deformed as the latter mechanisms. Therefore the particle displacement paths were not as simple as implemented by thermal-kinematic models. In conclusion, these models are applicable for the period after the cessation of pervasive ductile deformation. This is regardless weather the lithotectonic unit was emplaced according to the channel flow tectonic mode or to the classical fold nappe mode. In either case the pervasive ductile deformation occurred before the thermochronological record used here. Finally, the thermochronological data presented here and available in general cannot constrain the tectonic processes that occurred before them.

[reply] The short answer is that we completely agree with Reviewer 3 that "the thermochronological data presented here and available in general cannot constrain the tectonic processes that occurred before them." The available cooling age data we are evaluating are all younger (MAr ages of 11-14 Ma) than the MCT emplacement (23-16 Ma). However, what is interesting about the model process is that some of the models (such as our best fit model between 80-90 km from the MFT) predicts MAr ages that are a result of the proposed age and rate of the MCT. We do not have data in this region, so it would be interesting to see if they are in fact as old as what is predicted. With respect to the MCT as a fault or a shear zone, the boundary between uniquely Greater Himalayan rocks (by provenance) and Lesser Himalayan rocks (again by provenance)

is actually quite discrete (« 1 km). However, we absolutely agree with Reviewer 3 that the shear imparted to the rocks above and below this zone is pervasive and hetero-geneous and that the emplacement of the MCT on the LH rocks occurred while both lithotectonic packages were not behaving in a purely elastic or brittle fashion. We also agree that the AFT, ZHe, and MAr cooling ages reproduced accurately by the model all reflect cooling after predominantly ductile deformation in these rocks. It is important to note that the models used (Move and Pecube) do not attribute any mechanical behavior to the rocks. They only describe kinematics, or the motion of material. The kinematics invoked here are just as discrete as the kinematics used in Coutand et al. (2014) at 600°-700°C at 20-30 km depth or Herman et al. (2010) at 600-700 °C and 20-30km depth. The kinematics modeled in Move do not differentiate how ductile or plastically the rocks are deforming internally. The emplacement of the Greater Himalayan rocks above Lesser Himalayan rocks is critical for the heating and cooling of the Lesser Hi-malayan rocks and thus needs to be in the model. If the data we were evaluating were sensitive to the compressed temperature gradient (~450-700°) across the MCT zone (~ 1 km below and above in Bhutan; e.g. Long et al., 2016), trying to replicate the magnitude of fault-parallel shear would be more critical. As stated in response to General Comment 1, with respect to sensitivity of the cooling ages to the STD, the model predicts a cooling history and exhumation age and rate for GH rocks that can be compared to measured histories to assess how close just the simple (albeit possibly ductile) thrust emplacement model can account for the measured temperatures before attempting to incorporate a much more complex process. The manuscript was revised to clarify these points in Sections 2 and 3.

5. The authors analyze and discuss the effect of the thermophysical properties of the rocks on the spatial pattern of cooling ages. However only the values of heat production were changed (2 and 4 $\mu$W/m3). However the thermal properties control the Péclet number, which dictates how strongly are the isotherms deflected because of the thrusting. This furthermore implies that the thermal properties have to include the study of sensitivity on thermal conductivity, heat capacity and density of the rocks.

[reply] We agree with Reviewer 3 that crustal thermal fields are sensitive to thermal conductivity, heat capacity, and density. In this study (and most exhumation studies), crustal thermal properties are assumed constant because most thermal models, including Pecube, solve the advection-diffusion equation on an Eulerian grid which is not capable of tracking moving material properties. This makes implementation of variable thermal conductivity in a highly deformed thrust belt impossible. Lagrangian grids circumvent this problem, but have shortcoming for exhumation studies. They cannot accommodate large amounts of deformation, such as in the Himalaya, without becoming unstable, and require frequent re-meshing and interpolation of model parameters and properties, thereby progressively introducing numerical uncertainty into the model. Although hybrid Eulerian-Lagrangian techniques exist, these are not commonly used and difficult to implement. Given these limitations, we (like most other studies) use average upper crustal thermophysical properties and assume they remain constant through time. However, please note that the thermophysical properties we do use are based on observations (largely from Whipp et al., 2007 and references therein). To accommodate this reviewer's concern, we have modified the manuscript in the following ways: 1. We more clearly state in the model setup section 3.2.1 that we are using observed thermal physical properties for the lithologies present in this region (see Whipp et al. 2007, and Ehlers, 2005). 2. We add a caveat statement in the same section to say: "Although thermophysical properties such as thermal conductivity, heat capacity, and density vary between different lithologies within a fold and thrust belt, the implementation of variable material properties in areas of large deformation is not possible in programs such as Pecube which solve the advection diffusion equation on an Eulerian grid. Thus, we address this potential issue by using the best available average measurements of thermophysical properties for the lithologies in this region." In addition, see response to RC1 (p. 9 l. 11) for surface radiogenic heat production.

Specific Comments: 1. Valla et al. [2010] have shown that relief development must be 2–3 times faster than the background exhumation/erosion rate to be recorded and quantitatively extracted from thermochronological data. Valla, P., F. Herman, P. A. van

der Beek, and J. Braun (2010), Inversion of thermochronological age-elevation profiles to extract independent estimates of denudation and relief history I: Theory and conceptual model, Earth Planet. Sci. Lett., 295, 511–522. Please comment in your manuscript in the relevant places.

[reply] The approach and conclusions of Valla et al. [2010] is included in Introduction and Discussion.

2. What is the evidence in the field (i.e., petrological) for the burial by Kakhtang thrust? Kakhtang thrust appears very steep therefore the burial rate might not be high. In addition the KT emplaced some of the hottest rocks in the Himalaya therefore the isotherms might have been disturbed during its activity, in other words heating and cooling does not need to imply burial and exhumation.

[reply] We agree with the reviewer's comments. However, the flexural response of motion on the steep Kakhtang Thrust is subsidence in the footwall. Modeled isostatic accommodation of this thrusting dramatically lowered topography in the footwall of the thrust and reduced erosion rates south of the thrust. In some models, enough subsidence occurred during out-of-sequence thrusting that sedimentation occurred in the immediate footwall. A potential relict of this footwall subsidence is the enigmatic low-relief surface preserved in the Bhutan Himalaya (Duncan et al., 2003; Grujic et al., 2006). This low-relief landscape contains hundreds of meters of sediment infilling of paleo-relief and is now out of equilibrium with respect to where it was formed (Adams et al., 2016). In eastern Bhutan, the infilled sediment is derived from the structurally higher GH; conglomerate is common, thus making it easy to associate the clasts with rocks carried by the KT. As published by reviewer 3, the low-relief surface is in the immediate footwall of the Kakhtang Thrust (Grujic et al., 2006). Our flexural modeling of this region and others (e.g. McQuarrie and Ehlers, 2015, 2017; Rak et al., 2017) has highlighted the ubiquitous response of footwall subsidence and the development of low relief in the footwall region of out-of-sequence faults, thrusts, etc. We do not think that the spatial relationship between the low-relief surface and the Kaktang thrust is coincidental. Not only do slower erosion rates in the Kakhtang Thrust footwall alter thermal gradient, but the reduced topography limits the magnitude and rate of future erosion. The current model does take into account the deflection of isotherms as the KT moves and advects deep, hot material upward during fault motion. The amount of disturbance to isotherms in each time-step is related to the geometry of the fault and magnitude and rate of motion assigned in the time-step. Section 3.1.2 has been revised to explain these observations in the footwall of the KT and our rationale for modeling different timings of out-of-sequence thrusting. Section 4 has also been revised based on these comments.

3. Technical corrections a) Vertical uplift and vertical exhumation. Both rock and surface uplift and exhumation concern the vertical component of the particle displacement (in three different reference frames). Therefore word vertical is superfluous. However one must make difference between rock uplift and surface uplift, in particular in an article like this one where both processes are discussed. Please adhere strictly to the definitions by England and Molnar, 1990. Surface uplift, uplift of rocks, and exhumation of rocks. Geology, 18(12), pp.1173-1177. b) There is no process named "surface radiogenic heat production". Please correct the wording accordingly in the entire document.

[reply] Language was corrected to reflect England and Molnar definitions.

RC3: All the above comments and further technical comments are in the annotated file. Please also note the supplement to this comment: https://www.solid-earth-discuss.net/se-2017-117/se-2017-117-RC3-supplement.pdf

[reply] The corrections suggested in the supplementary document were all addressed. Specific comments raised in the Supplementary PDF that are not addressed above are included below.

p.3, l. 30-31 furthermore as indicated by thermo-kinematic experiments, like this study, the cooling rates were not steady in time and space.

[reply] We agree completely with this statement but feel the point is best addressed in the results and discussion sections. We have removed interpretations of "rapid cooling" and or processes and simply describe the age data.

p.4 line 5 Is there any effect of sample elevation? This is important since you are testing the model for the sensitivity on surface processes.

[reply] There are very limited/ modest age elevation relationships as discussed in McQuarrie and Ehlers, 2015. The relationships that are present include 1) the samples from Coutand et al. (2014) that have an age-elevation relationship, and if used to determine an exhumation rate, suggest a very modest rate of 0.4 mm/yr; and 2) the southern ZHe samples when combining the data from the Kuru Chu and Trashigang transects, located ∼30 km from the MFT. The younger ages (8.5 to 10 Ma) at lower elevations (0.5 to 1 km) in the Kuru Chu and older ages (11 to 11.6 Ma) at higher elevations (1.6 to 2.4 km) along the Trashigang transect suggests differential exhumation of 0.7 mm/yr. These are expanded on in sections 2.2 and in 5.3.

p. 4, l. 20 since it is an active convergent orogen the isotherms are deformed and the geothermal gradient is not constant in space and time. Was this accounted for in the model when calculating the eroded material?

[reply] In this section we mention that advection-diffusion thermal models are used to calculate the evolving subsurface temperatures (i.e. modified isotherms and geothermal gradient) but discuss this more explicitly in section 3.2.

p. 7, l.30 What is the argument that the geometry along distant section is applicable to the study area? How do these values compare to the estimates by Coutand et al. (2014) and by Singer, J., Obermann, A., Kissling, E., Fang, H., Hetényi, G., Grujic, D. (2017) Along-strike variations in the Himalayan orogenic wedge structure in Bhutan from ambient seismic noise tomography. Geochemistry, Geophysics, Geosystems, 18, 4, 1483-1498. DOI: 10.1002/2016GC006742.

[reply] We have removed the INDEPTH reference. We found that Singer et al. (2017, JGR Solid Earth) had the specific data on to the dip of the décollement and the dip of the Moho. The relationship between the décollement geometry of Long et al. (2011b) and Coutand et al. (2014) is no different than that described in McQuarrie and Ehlers (2015; figure 3C). However the modified décollement geometry is much closer to EB1 in Coutand et al. (2014), with more discrete ramp steps.

p. 9, l. 8 ["surface radiogenic heat production"] –There is no such physical process. Please correct the wording accordingly in the entire document.

[reply] Author Response: With due respect, yes, there is. Please see detailed response to comment: p. 9 l. 11 from Reviewer 1: Radiogenic heat production at the surface can vary spatially by large amounts (e.g., Mareschal and Jaupart, 2013) and is a function of the concentration of heat-producing elements in the crust. Systematic sampling of crustal rocks now exposed at the surface indicates that heat production diminishes with depth through the crust and that this decline is not monotonic (Ketcham, 1996; Brady et al., 2006). Thus we prescribe an exponential decrease in heat production with depth, as opposed to assuming a constant crustal heat production. An exponential decrease in heat production with depth requires definition of a surface radiogenic heat production (Ao) and an e-folding depth.

p. 13, l. 25 there is no such a process. Do you mean surface heat flow or (radiogenic) heat production

[reply] Neither, please see response to p. 9, l. 8.

p. 15, l.2 The GHC is not a thrust sheet as the MCT is not a fault but a several kilometers thick ductile shear zone. Therefore the particle displacement paths were as simple as predicted by thermal-kinematic models. Therefore these models are applicable for the period after the cessation of pervasive ductile deformation.

[reply] In the end we may need to continue to agree to disagree with Reviewer 3 on this

point. Our definition of a thrust sheet is not quite that rigid (literally and figuratively). We have added "ductile" in front of "thrust" because we agree with Reviewer 3 that the fault that places Greater Himalayan rocks on Lesser Himalayan rocks is part of a much broader shear zone with pervasive shear both above and below the tectonostratigraphic boundary between the two units.

p. 15, l.25-29 Do you consider also that the isotherms are less deflected above this ramp than above the major ramp? This influences how closely spaced are the isotherms and therefore even with the sample particle displacement vector and surface denudation rate, the cooling rate will be different.

[reply] Yes, absolutely. The version of Pecube that we are using calculates the evolving thermal field including the deflection (or lack thereof) of isotherms with ramps.

p. 16, l. 28 Is there a justification to increase the number of significant digits (i.e. topography angle of 1.75°)

[reply] Yes, small changes to small angles (0.25° is 12% of a 2° angle) have a large effect when applied over hundreds of kilometers.

p. 18, l. 5 [The sensitivity of the model to the age of MCT] –However this depends also on how is the MCT treated. As a single fault with a displacement of a slab above it or as it is in the field as broad ductile shear zone with pervasive deformation in the hanging wall block.

[reply] While we agree that how deformation in the broad MCT zone is treated may have an effect on the temperatures within a few kilometers above and below the thrust fault (within the shear zone), this deformation is not captured by any of the available cooling age data we are evaluating. All of the thermochronometers available are younger (11-14 Ma) than the MCT emplacement (23-16 Ma). Available geochronologic data for this region support ductile shearing on the lower STD between circa 23 and 16 Ma, and shearing and associated exhumation of GH rocks in the MCT sheet after circa 23 Ma

and continuing until circa 18–16 Ma [Grujic et al., 2002; Daniel et al., 2003; Kellett et al., 2009, 2010; Chambers et al., 2011]. Thus our statement that we cannot evaluate the MCT emplacement age or rate is correct.

Figure 2 and Figure 4: [pointing to northern part of cross-section]–Something is missing here, both the topography and the geology.

[reply] The geology and topography in the northernmost portion of the cross section was never a part of the original geologic cross section (Long et al., 2011b). This is why it is blank in this figure as well.

Please also note the supplement to this comment:
https://www.solid-earth-discuss.net/se-2017-117/se-2017-117-AC3-supplement.pdf

---

## Author Comment (AC4) · 11 Feb 2018

[revised manuscript text omitted]

---

## Author Comment (AC5) · 11 Feb 2018

[Figure]

**Supplementary Figure 1: Flexural-kinematic modelling output of the present-day Trashigang cross-section for all models presented in sections 4 and 5.**

**Supplementary Figure 2: Predicted MAr (yellow), ZHe (green), and AFT (blue) cooling ages using a flexural-kinematic model of the modified Trashigang cross-section geometry proposed in this study (Supplementary Figure 1f), preferred shortening rates (Table 3b), and surface radiogenic heat production values of Velocity B, and surface radiogenic heat production values of (A) 4.0 μW/m³ and (B) 2.0 μW/m³. The Baxa footwall ramp in the décollement has been shifted 35 km north. Published data include additional ages from the Kuru Chu line of section west of the Trashigang section (Long et al., 2012), shown in the graphs with lighter-colored symbols. The flexural-kinematic model used a Split KT kinematic scenario and Responsive topography.**

**Supplementary Figure 3: Predicted MAr (yellow), ZHe (green), and AFT (blue) cooling ages using the Trashigang cross-section geometry originally proposed by Long et al. (2011a), Velocity B, and surface radiogenic heat production values of (A) 4.0 µW/m³ and (B) 2.0 µW/m³ . Published data include additional ages from the Kuru Chu line of section west of the Trashigang section (Long et al., 2012), shown in the graphs with lighter-colored symbols. The flexural-kinematic model used a Split KT kinematic scenario and Responsive topography (Supplementary Figure 1a).**

| Study | Sample | Unit | Elevation (m) | Longitude (°E) | Latitude (°N) | Central AFT Age (Ma) | 2σ Analytical Error (Ma) | Mean ZHe Age (Ma) | 2σ Variability Range (Ma) | Reported MAr Age (Ma) | 2σ Analytical Error (Ma) |
|---|---|---|---|---|---|---|---|---|---|---|---|
| Stuwe & Foster (2001)* | 8 | GHlo | 2540 | 91.53157 | 27.24116 | - | - | - | - | 14.1 | 0.2 |
| Stuwe & Foster (2001)* | 9 | GHlo | 2480 | 91.52660 | 27.24549 | 3.1 | 1.2 | - | - | 11.1 | 0.4 |
| Stuwe & Foster (2001)* | 11 | GHlo | 1750 | 91.54075 | 27.27539 | - | - | - | - | 14.1 | 0.4 |
| Stuwe & Foster (2001)* | 12 | GHlo | 1060 | 91.54378 | 27.32288 | - | - | - | - | 11.0 | 0.4 |
| Grujic et al. (2006) | BH53 | GHlo | 2405 | 91.548083 | 27.237361 | 6.9 | 2.6 | - | - | - | - |
| Grujic et al. (2006) | BH52 | GHlo | 2350 | 91.554667 | 27.236056 | 7.8 | 2.8 | - | - | - | - |
| Grujic et al. (2006) | BH60 | Pzj | 795 | 91.480667 | 27.282361 | 4.2 | 1.0 | - | - | - | - |
| Grujic et al. (2006) | BH61 | GHlo | 780 | 91.491000 | 27.303417 | 5.4 | 0.8 | - | - | - | - |
| Grujic et al. (2006) | BH90 | GHlo | 910 | 91.574528 | 27.344972 | 3.6 | 1.0 | - | - | - | - |
| Grujic et al. (2006) | BH64 | GHlo | 825 | 91.554472 | 27.350056 | 3.0 | 1.4 | - | - | - | - |
| Grujic et al. (2006) | BH324 | GHlo | 1995 | 91.59683 | 27.374361 | 4.8 | 1.0 | - | - | - | - |
| Grujic et al. (2006) | BH94 | Pzc | 2050 | 91.599833 | 27.375333 | 6.6 | 0.8 | - | - | - | - |
| Grujic et al. (2006) | BH100 | GHlo | 905 | 91.563722 | 27.411389 | 5.9 | 0.8 | - | - | - | - |
| Grujic et al. (2006) | BH72 | GHlo | 1420 | 91.554722 | 27.465000 | 5.5 | 0.8 | - | - | - | - |
| Grujic et al. (2006) | BH66 | GHlo | 930 | 91.561139 | 27.551361 | 4.4 | 1.2 | - | - | - | - |
| Grujic et al. (2006) | BH70 | GHlo | 1760 | 91.499528 | 27.584167 | 3.7 | 0.6 | - | - | - | - |
| Long et al. (2012) | BU07-53 | Pzg | 655 | 91.48011 | 26.86572 | - | - | 8.65 | 2.22 | - | - |
| Long et al. (2012) | BU07-54 | Pzd | 700 | 91.48028 | 26.87497 | - | - | 7.61 | 0.28 | - | - |
| Long et al. (2012) | BU07-33 | Pzd | 1710 | 91.54794 | 26.93311 | 5.69 | 1.04 | 11.12 | 0.85 | - | - |
| Long et al. (2012) | BU07-35 | Pzb | 1580 | 91.54761 | 26.95992 | 5.82 | 1.28 | 10.91 | 2.29 | - | - |
| Long et al. (2012) | BU07-36 | Pzb | 1785 | 91.53083 | 26.97442 | - | - | 11.00 | 1.47 | - | - |
| Long et al. (2012) | NBH-18 | Pzb | 1815 | 91.52072 | 27.01200 | - | - | 11.60 | 0.03 | - | - |
| Long et al. (2012) | BU07-37 | Pzb | 2385 | 91.50142 | 27.02675 | 6.27 | 2.34 | 11.25 | 0.50 | - | - |
| Long et al. (2012) | BU07-42 | Pzb | 2165 | 91.52089 | 27.08486 | - | - | 9.54 | 1.82 | - | - |
| Long et al. (2012) | BU07-43B | Pcd | 2315 | 91.56708 | 27.13450 | - | - | 9.43 | 1.33 | - | - |
| Long et al. (2012) | BU07-55 | Pzj | 2350 | 91.52122 | 27.24222 | - | - | 11.07 | 8.27 | - | - |
| Long et al. (2012) | BH-57 | Pcd | 605 | 91.44656 | 27.27869 | - | - | 7.30 | 0.77 | - | - |
| Long et al. (2012) | BH-78 | Pzc | 1000 | 91.63897 | 27.35144 | - | - | 7.09 | 0.28 | - | - |
| Coutand et al. (2014) | BH-363 | GHh | 3610 | 91.37263 | 27.96956 | 2.5 | 0.4 | - | - | - | - |
| Coutand et al. (2014) | BH-351 | GHh | 3870 | 91.30357 | 27.97318 | 3.0 | 2.4 | - | - | - | - |
| Coutand et al. (2014) | BH-352 | GHh | 3880 | 91.29016 | 27.97416 | 4.1 | 0.6 | - | - | - | - |
| Coutand et al. (2014) | BH-357 | GHh | 4085 | 91.29827 | 27.98563 | 4.0 | 0.4 | 7.42 | 1.56 | - | - |
| Coutand et al. (2014) | BH-355 | GHh | 4275 | 91.2987 | 27.99005 | 3.8 | 0.6 | - | - | - | - |
| Coutand et al. (2014) | BH-362 | GHh | 4300 | 91.29901 | 27.99750 | 4.2 | 0.8 | - | - | - | - |

**Supplementary Table 1: Thermochronometer sample locations and reported cooling ages used in this study. Reported AFT and MAr data include 2σ analytical error. ZHe ages are based on the mean reported age among all aliquots for each sample; 2σ range shown for ZHe includes variability among aliquots. *Latitude and longitude of samples from Stüwe and Foster**

5   **(2001) were estimated using ESRI ArcMap WGS84 datum.**

---

## Author Response (AR2)

Reviewer 1:  Peter van der Beek review 2

This revised version of the manuscript by Gilmore et al. addresses the comments expressed by myself and other reviewers in a mostly quite satisfactory manner and has become much easier to read and follow than the initial discussion paper. I had two major comments on the initial version of this manuscript, concerning (1) the context of the study and (2) the history of shortening rates. The context is now clearly laid out and I have no further queries on that. I am still not completely convinced that the inferred history of shortening and shortening rates is the only one allowable by the data, but this should not stand in the way of publication of this manuscript. This methodology is clearly still being developed, and therefore it is logical that some of the results remain slightly equivocal at this stage.

So overall I think this manuscript is pretty much ready to go. I have a few minor specific comments, indicated below by page/line number. It could do with a final critical editorial read-through as well, as there are still some minor glitches in the English.

> Three authors have read over the revised manuscript and caught numerous small English and typing errors.

p. 2/l. 7: This is not quite what Erdös et al. (2014) did. Instead, they argued that, since some (admittedly disputable) paleo-elevation estimates argue for similar elevations of the Pyrenees to the present since Eocene times, they could use steady-state topography in their models. They then added a transient fill of the valley bottoms in some of their models, but this only affected the post-orogenic phase (see Fillon and van der Beek, Basin Res. 2012 for a detailed description of what was done and why).

> We agree, in trying to keep the sentence simple, we have misrepresented the approach. We have rephrased this to state "…as a steady state topography that matches modern topography (e.g. Coutand et al., 2014; Erdös et al., 2014; Herman et al., 2010; Whipp et al., 2007) or as an evolving topography where relief increases or decreases with time as indicated by geologic datasets (e.g., Erdös et al., 2014)." We hope by adding the Erdos reference to both and clarifying the reason for imposing an evolving topography addresses the valid critique without going into the details of evolving topographic theories in the Pyrenees.

p. 2/l. 11: Valla, 2010 should be Valla et al., 2010.

> Changed.

p. 2/l. 17-18 "We do this by … in the Bhutan Himalaya": this phrase is incomplete and unclear – adjacent to what?

Adjacent to the section evaluated by McQuarrie and Ehlers (2015).  This was added to clarify differences to the McQuarrie and Ehlers (2015) study.  However we agree that the phrasing was not clear.  We have edited this section to read:

"We expand on the approach taken by McQuarrie and Ehlers (2015) and assess the control that cross section geometry, kinematics, shortening rates, and topographic assumptions have on modeled cooling ages by systematically changing these features. These parameters are evaluated using a balanced geologic cross section and associated thermochronometer data from the Bhutan Himalaya on section that is adjacent (30 km east) to the one examined in McQuarrie and Ehlers (2015)."

p. 3/l. 21-24: I suppose this was added to discuss the comment that the Move restorations cannot model ductile deformation. This is indeed an issue that needs to be discussed, but I am not sure that the point made here is very clear.

We have made this clearer by adding the section in italics to the text:  "Both Greater and Lesser Himalayan rocks preserve pervasive ductile deformation above and below the MCT (Grujic et al., 1996; Long et al., 2011c; Long et al., 2016) *that cannot be replicated with kinematics that only account for fault displacement.* However, during initial emplacement of the MCT and active displacement on the MHT, ductile processes at depth transition to brittle processes as thrust and shear systems approach the surface, with a transition temperature of ~350 °C (Avouac, 2007).

p. 6/l. 24: Jordan & Watts, 2005 is not (yet) in the reference list.

Added.

p. 6/l. 25-28: Similarly, this was added to justify the choice of EET values, which appear higher than those inferred from flexure/gravity studies in the eastern Himalaya (as acknowledged in the previous phrase). However, I am not sure I understand the justification; it is not clear how the timescale come into the argument? One could clearly argue that the geophysical studies really only constrain the modern EET and not what it could have been in the past, which should in theory be possible to constrain with the kinematic-flexural modelling performed here. However, this should still end up with the same modern-day EET.

Yes, geophysical studies really only constrain the modern EET and not what it could have been in the past. We agree that the modern value *is* the modern value (and in an ideal response we should end up with it). However the last step of the model is a 1.5 – 2 Myr increment (i.e. still not modern). In addition, changing the last step has almost negligible impact on the final decollement geometry, erosion level, or foreland basin because each step has a small incremental response that adds up to the final solution. We address this in the paper by adding "Keeping all other parameters the same, a change in EET at the last model step to reflect modern

conditions established between ~ 2 Ma to present would increase the depth of the decollement and decrease the resulting magnitude of erosion at the surface by 250 m." However, it would also be possible to change the preceding EET and density so that the final basin geometry has no change to that modeled. This is what we mean by "EET and density values are not unique" (p. 7 / l. 1-2). There are many aspects that control measured values of EET that are not included in a basic line or rectangular load flexure model, which is why it is worth discussing how sensitive the model results are to small changes in EET (see section 5.1.1 and comments to p. 19 / l. 31-32 below).

P 13/l. 13: this choice of (relatively high) surface heat-production value does not follow logically from the preceding section (which concluded that relatively low values predicted the thermochronology data better) and should therefore be justified here.

The surface heat-production value, as well as the other parameters mentioned (responsive topography and split Kakhtang thrust kinematics), produced the best fit of modeled to measured ages for the original cross section geometry. This is specified in the revised manuscript p. 13 / l. 19-20. The reason why is that the absolute ages are much more sensitive to velocity changes than Ao.

p. 19/l. 26: shouldn't this be "farther to the foreland" instead of "hinterland"?

No—farther to the hinterland is correct, i.e. the height of ramps that these rocks have experienced structural uplift and associated erosion *previously.* We clarify this in the text by adding "where earlier uplift and erosion occurred (Fig. 3C2)" to p. 19 / l.31.

p. 19/l. 31-32: how does the EET "control the location of rocks with respect to the mantle"? This is unclear and somewhat mysterious to me.

Reviewer 1 is correct, that was a poor way to describe the response. It has been re-written to state "
[revised manuscript text omitted]

Michelle Gilmore 4/9/2018 5:42 PM

Michelle Gilmore 4/9/2018 5:43 PM

Eizenhoefer, Paul R. 4/10/2018 9:32 AM

Eizenhoefer, Paul R. 4/10/2018 9:33 AM

Nadine McQuarrie 4/10/2018 1:56 PM

Michelle Gilmore 4/9/2018 5:55 PM

Michelle Gilmore 4/9/2018 5:57 PM

Michelle Gilmore 4/9/2018 5:58 PM

Nadine McQuarrie 4/9/2018 2:44 PM

Nadine McQuarrie 4/9/2018 2:45 PM

Michelle Gilmore 4/9/2018 5:58 PM

Nadine McQuarrie 4/9/2018 2:47 PM

Nadine McQuarrie 4/9/2018 2:46 PM

Eizenhoefer, Paul R. 4/10/2018 9:35 AM

Nadine McQuarrie 4/10/2018 1:57 PM

Eizenhoefer, Paul R. 4/10/2018 9:37 AM

Nadine McQuarrie 4/9/2018 2:48 PM

Eizenhoefer, Paul R. 4/10/2018 9:37 AM

Eizenhoefer, Paul R. 4/10/2018 9:37 AM

[revised manuscript text omitted]

Nadine  McQuarrie 4/9/2018 3:40 PM

Michelle Gilmore 4/9/2018 9:06 PM

Eizenhoefer, Paul R. 4/10/2018 11:26 AM

Eizenhoefer, Paul R. 4/10/2018 11:27 AM

Michelle Gilmore 4/9/2018 9:08 PM

Nadine  McQuarrie 4/9/2018 3:42 PM

Nadine  McQuarrie 4/9/2018 2:15 PM

Michelle Gilmore 4/9/2018 9:08 PM

Nadine  McQuarrie 4/9/2018 2:15 PM

Nadine  McQuarrie 4/9/2018 3:43 PM

Nadine  McQuarrie 4/9/2018 2:16 PM

Nadine  McQuarrie 4/9/2018 3:43 PM

Nadine  McQuarrie 4/9/2018 3:43 PM

Eizenhoefer, Paul R. 4/10/2018 11:28 AM

Eizenhoefer, Paul R. 4/10/2018 11:28 AM

Michelle Gilmore 4/9/2018 9:10 PM

Michelle Gilmore 4/9/2018 9:12 PM

Nadine  McQuarrie 4/10/2018 3:42 PM

Nadine  McQuarrie 4/10/2018 3:42 PM

Nadine  McQuarrie 4/10/2018 3:43 PM

Michelle Gilmore 4/9/2018 9:12 PM

Michelle Gilmore 4/9/2018 9:12 PM

[revised manuscript text omitted]

Eizenhoefer, Paul R. 4/10/2018 11:49 AM